# Stage-wise Distortion–Perception Traversal in Zero-shot Inverse Problems with Diffusion Models

**Jiawei Zhang** [1]  **Ziyuan Liu** [1]  **Leon Yan** [1]  **Zhenyu Xiao** [1]  **Yuantao Gu** [2][1]

## Abstract

The distortion–perception (D–P) tradeoff is a fundamental phenomenon of Bayesian inverse problems, which characterizes the inherent tension between distortion performance and perceptual quality. Enabling flexible traversal of the D-P tradeoff at inference time is crucial for practical applications. Despite the recent success of diffusion models in zero-shot inverse problem solving, efficient and principled strategies for D-P traversal in diffusion-based inverse algorithms remain inadequately characterized. In this paper, we propose a stage-wise framework for realizing D-P traversal using a single diffusion model in zero-shot inverse problems. Our proposed method, termed MAP-RPS, starts with an MAP estimation stage that approximates the MMSE solution and provides a low-distortion initialization, followed by a re-noised posterior sampling stage that progressively improves perceptual quality. We provide theoretical analyses for both stages, establishing the validity and effectiveness of the proposed design. Furthermore, we extend MAP-RPS to the latent space, yielding LMAP-RPS, which enjoys broader applicability by leveraging large-scale pre-trained latent diffusion backbones. Extensive experiments demonstrate that MAP-RPS and LMAP-RPS enable more effective D-P traversal on various tasks, while also exhibiting strong performance as efficient solvers for real-world inverse problems.

## 1. Introduction

Bayesian inverse problems are fundamental to a wide range of scientific and engineering applications, where the goal is to faithfully reconstruct original signals from degraded observations. In recent years, diffusion models (Sohl-Dickstein et al., 2015; Ho et al., 2020; Kingma et al., 2021) have emerged as a powerful class of generative methods. Their remarkable capability to capture complex data priors enables zero-shot solutions to various inverse problems without task-specific retraining (Kawar et al., 2022; Chung et al., 2023; Song et al., 2023; Rout et al., 2023; Song et al., 2024; Rout et al., 2024; Mardani et al., 2024; Zhang et al., 2024a; 2025a;b; Alkhouri et al., 2025). Consequently, diffusion-based inverse algorithms have achieved state-of-the-art (SOTA) performance across diverse applications, including image restoration (Lugmayr et al., 2022), signal separation (Mariani et al., 2024), medical imaging (Song et al., 2022), and compressed sensing (Elata et al., 2024).

Solving inverse problems is intrinsically constrained by the well-known distortion–perception (D-P) tradeoff (Blau & Michaeli, 2018; Freirich et al., 2021). Distortion performance measures sample-wise reconstruction fidelity under full-reference metrics, which is typically quantified by distances in Euclidean space. In contrast, perception performance reflects the visual or perceptual quality of reconstructions and is commonly characterized by discrepancies between the distributions of reconstructed and real signals. It has been theoretically established that distortion and perception objectives are inherently in tension, implying that improving one often necessitates a degradation of the other.

Motivated by these developments, there has been a growing interest in adjusting the D–P tradeoff at inference time beyond merely designing diffusion-based inverse algorithms in order to accommodate diverse practical requirements. Several studies have empirically observed that adjusting the number of sampling steps (Whang et al., 2022; Delbracio & Milanfar, 2023; Ren et al., 2023; Yue et al., 2024; Luo et al., 2024; Zhussip et al., 2024; Cohen et al., 2025), performing sample averaging (Whang et al., 2022; Li et al., 2024a), or tuning certain algorithmic hyperparameters (Mardani et al., 2024; Zhang et al., 2024a) can partially control the tradeoff. More recently, Wang et al. (2025) propose a variance-scaled posterior sampling strategy that modulates the injected noise level together with careful hyperparameter calibration, and establish theoretical guarantees under Gaussian posterior assumptions. However, despite these advances, principled

[1]Department of Electronic Engineering, Tsinghua University, Beijing, China [2]Shenzhen International Graduate School, Tsinghua University, Shenzhen, China. Correspondence to: Yuantao Gu <gyt@tsinghua.edu.cn>.

*Proceedings of the 43rd International Conference on Machine Learning*, Seoul, South Korea. PMLR 306, 2026. Copyright 2026 by the author(s).

and computationally efficient mechanisms for traversing the D-P tradeoff remain inadequately characterized in diffusion-based zero-shot inverse algorithms.

To tackle this challenge, we propose a stage-wise framework to realize D-P traversal with a single pretrained diffusion model. Specifically, the first stage employs a maximum a posteriori (MAP) estimator to approximate the minimum mean squared error (MMSE) solution, serving as a low-distortion starting point. Then the second stage progressively reduces the perception error via a re-noised posterior sampling strategy. We term this two-stage framework MAP-RPS, and provide theoretical analyses for both stages, demonstrating its effectiveness in enabling D-P traversal. Furthermore, we extend MAP-RPS to latent-space diffusion models, yielding an algorithm termed LMAP-RPS, which enjoys broader applicability by leveraging powerful pretrained latent diffusion backbones.

We compare MAP-RPS and LMAP-RPS with existing D-P traversal methods across various image inverse problems. The results demonstrate that our approaches more closely approximate the ideal D-P curve, providing a novel perspective for realizing D-P traversal in zero-shot inverse problems with pretrained diffusion models. Moreover, the proposed framework can also serve as a practical inverse algorithm. We further evaluate LMAP-RPS on near–real-world inverse problems using the MS-COCO dataset. Compared with existing latent diffusion–based inverse algorithms, LMAP-RPS achieves superior performance with even lower computational complexity, highlighting its strong practical potential and providing new insights into the design of efficient diffusion-based inverse algorithms in the future. Codes for reproducing our experiments are available at https://github.com/weigerzan/MAP_RPS.

# 2. Backgrounds

## 2.1. Diffusion models

Diffusion models (Sohl-Dickstein et al., 2015; Ho et al., 2020; Kingma et al., 2021) consist of a forward diffusion process and a corresponding reverse process. The forward process gradually perturbs data drawn from a target distribution $p_{X_0} = p_X$ into standard Gaussian noise $\mathcal{N}(\mathbf{0}, \mathbf{I})$, while the reverse process aims to invert this corruption by transforming noise back into data samples. In this work, we focus on variance-preserving (VP) diffusion models, where the forward process is described by the following stochastic differential equation (SDE):

$$d\mathbf{x}_t = f(t)\mathbf{x}_t dt + g(t)d\mathbf{w}_t, \quad t \in [0, T], \ \mathbf{x}_0 \sim p_{X_0}, \quad (1)$$

where $f(t) = \frac{d \log \sqrt{\overline{\alpha}_t}}{dt}$ and

$$g^2(t) = \frac{d(1 - \overline{\alpha}_t)}{dt} - 2\frac{d \log \sqrt{\overline{\alpha}_t}}{dt}(1 - \overline{\alpha}_t).$$

Here $\mathbf{w}_t$ denotes the standard Wiener process, and $\overline{\alpha}_t$ are monotonically decreasing hyperparameters satisfying $\overline{\alpha}_0 = 1$ and $\overline{\alpha}_T = 0$.

The forward SDE admits the following reverse-time SDE with identical joint distribution (Song et al., 2021b):

$$d\mathbf{x}_t = \left(f(t)\mathbf{x}_t - g^2(t)\nabla_{\mathbf{x}_t} \log p_t(\mathbf{x}_t)\right) dt + g(t)d\overline{\mathbf{w}}_t, \quad (2)$$

where $\mathbf{x}_T \sim \mathcal{N}(\mathbf{0}, \mathbf{I})$, $\overline{\mathbf{w}}_t$ is the reverse-time Wiener process, and $\nabla_{\mathbf{x}_t} \log p_t(\mathbf{x}_t)$ denotes the (Stein) score function (Chwialkowski et al., 2016; Liu et al., 2016). In practice, the score function can be approximated by a neural network $\mathbf{s}_\theta(\mathbf{x}_t, t)$ trained via score matching. Moreover, the reverse SDE admits an equivalent probability flow ordinary differential equation (ODE) that shares the same marginal distributions (Song et al., 2021a;b):

$$\frac{d\mathbf{x}_t}{dt} = f(t)\mathbf{x}_t - \frac{1}{2}g^2(t)\nabla_{\mathbf{x}_t} \log p_t(\mathbf{x}_t). \quad (3)$$

By numerically solving either the reverse-time SDE (2) or the probability flow ODE (3), one can generate samples from the original data distribution.

## 2.2. Diffusion-based zero-shot inverse problem solvers

The goal of an inverse problem is to recover an unknown signal from degraded observations. Let $X \in \mathbb{R}^{n_x}$ and $Y \in \mathbb{R}^{n_y}$ denote random variables representing the signal and the observation, respectively. The observation model induces a likelihood $p_{Y|X}(\mathbf{y} \mid \mathbf{x})$ through

$$\mathbf{y} = \mathcal{A}(\mathbf{x}) + \sigma_{\mathbf{y}}\mathbf{n}, \quad (4)$$

where $\mathbf{n} \sim \mathcal{N}(\mathbf{0}, \mathbf{I})$, $\sigma_{\mathbf{y}}$ is the standard deviation of the additive noise, and $\mathcal{A}$ denotes a (possibly non-invertible and non-linear) degradation operator. Given an observation $\mathbf{y}$, inverse problems amount to estimating the original $\mathbf{x}$, or equivalently constructing an estimator $p_{\hat{X}|Y}$ that approximates the posterior distribution $p_{X|Y}$ or its relevant statistics. By Bayes' rule, the posterior distribution is given by

$$p_{X|Y}(\mathbf{x} \mid \mathbf{y}) \propto p_X(\mathbf{x})p_{Y|X}(\mathbf{y} \mid \mathbf{x}), \quad (5)$$

where the likelihood $p_{Y|X}(\mathbf{y} \mid \mathbf{x})$ is explicitly determined by the observation model. As diffusion models have demonstrated remarkable capability in modeling complex data priors $p_X(\mathbf{x})$ (Song et al., 2021b; Dhariwal & Nichol, 2021; Rombach et al., 2022; Li & Yan, 2024), they provide a principled framework for solving inverse problems in a zero-shot manner without retraining on task-specific paired data. Recent studies (Kawar et al., 2022; Chung et al., 2023; Song et al., 2023; Rout et al., 2023; Song et al., 2024; Rout et al., 2024; Mardani et al., 2024; Zhang et al., 2024a; 2025a;b; Alkhouri et al., 2025) have shown that diffusion-based methods achieve strong performance across a wide

range of inverse problems, including image super-resolution, denoising, deblurring, and inpainting.

One of the dominant paradigms for diffusion-based inverse problem solving is posterior sampling, which aims to draw samples directly from the posterior distribution $p_{X|Y}$. Following the score-based formulation, the reverse-time SDE for posterior sampling is given by (Song et al., 2021b)

$$d\mathbf{x}_t = \left(f(t)\mathbf{x}_t - g^2(t)\nabla_{\mathbf{x}_t}\log p_t(\mathbf{x}_t|\mathbf{y})\right)dt + g(t)d\overline{\mathbf{w}}_t, \quad (6)$$

with $\mathbf{x}_T \sim p_T(\mathbf{x}_T \mid \mathbf{y})$. Note that the posterior score function admits the following decomposition:

$$\nabla_{\mathbf{x}_t}\log p_t(\mathbf{x}_t \mid \mathbf{y}) = \nabla_{\mathbf{x}_t}\log p_t(\mathbf{x}_t) + \nabla_{\mathbf{x}_t}\log p_t(\mathbf{y} \mid \mathbf{x}_t), \quad (7)$$

where the prior score $\nabla_{\mathbf{x}_t}\log p_t(\mathbf{x}_t)$ is provided by a pre-trained diffusion model. Existing diffusion-based posterior sampling methods differ primarily in how they approximate the likelihood term $\nabla_{\mathbf{x}_t}\log p_t(\mathbf{y} \mid \mathbf{x}_t)$ under various assumptions. For example, DPS (Chung et al., 2023) approximates the likelihood term by

$$p_t(\mathbf{y} \mid \mathbf{x}_t) \simeq p(\mathbf{y} \mid \hat{\mathbf{x}}_0), \quad (8)$$

where

$$\hat{\mathbf{x}}_0 = \frac{1}{\sqrt{\overline{\alpha}_t}}\left(\mathbf{x}_t + (1 - \overline{\alpha}_t)\nabla_{\mathbf{x}_t}\log p_t(\mathbf{x}_t)\right), \quad (9)$$

which follows Tweedie's formula (Efron, 2011). The resulting approximation error is shown to be bounded by the Jensen gap (Chung et al., 2023). In the sequel, we denote a reasonable approximation of the posterior score by

$$\mathbf{s}_\theta(\mathbf{x}_t, \mathbf{y}, t) \simeq \nabla_{\mathbf{x}_t}\log p_t(\mathbf{x}_t \mid \mathbf{y}). \quad (10)$$

### 2.3. The distortion-perception tradeoff

In image restoration, it has long been observed that improved distortion metrics (e.g., RMSE or PSNR) do not necessarily translate into better perceptual quality. This phenomenon was formally characterized by Blau and Michaeli (2018) and termed the distortion-perception tradeoff (also known as the perception-distortion tradeoff). Specifically, the distortion-perception (D-P) function is defined as

$$D(P) = \min_{p_{\hat{X}|Y}}\{\mathbb{E}_{(\mathbf{x},\hat{\mathbf{x}})\sim p_{X\hat{X}}}[\Delta(\mathbf{x},\hat{\mathbf{x}})] : d_p(p_X, p_{\hat{X}}) \leq P\},$$

where $p_{X\hat{X}}$ is the joint distribution induced by $p_{X|Y}$ and $p_{\hat{X}|Y}$, and we assume that $X$ and $\hat{X}$ are conditionally independent given $Y$. Here $\Delta(\cdot,\cdot)$ denotes a distortion measure, while $d_p(\cdot,\cdot)$ measures the discrepancy between probability distributions. A key property of the D-P function is that it is monotonically non-increasing, implying that improving perceptual quality may incur a degradation in distortion performance, and vice versa.

Dror et al. (2021) provided a detailed analysis of the D-P function when the distortion metric is the mean squared error (MSE) and the perception metric is the Wasserstein-2 ($W_2$) distance. For two probability measures $\mu, \gamma \in \mathcal{P}(\mathbb{R}^n)$, the $W_2$ distance is defined as

$$W_2^2(\mu, \gamma) = \inf_{\nu\in\Pi(\mu,\gamma)}\int \|\mathbf{x} - \hat{\mathbf{x}}\|^2 d\nu(\mathbf{x}, \hat{\mathbf{x}}), \quad (11)$$

where

$$\Pi(\mu, \gamma) = \left\{\nu \in \mathcal{P}(\mathbb{R}^{2n}) : P_{1\#}\nu = \mu, P_{2\#}\nu = \gamma\right\}, \quad (12)$$

which denotes the set of all couplings between $\mu$ and $\gamma$. Here $P_1$ and $P_2$ are the projections onto the first and last $n$ coordinates, respectively, and $f_\#$ denotes the pushforward measure induced by a measurable mapping $f$. Under this setting, Dror et al. (2021) showed that the MSE-$W_2$ tradeoff admits a closed-form characterization:

$$D(P) = D^* + [(P^* - P) \vee 0]^2, \quad (13)$$

where $\vee$ denotes the maximum operator and

$$D^* = \mathbb{E}\|X - X_{\text{MMSE}}\|^2, \ P^* = W_2\left(p_X, p_{X_{\text{MMSE}}}\right). \quad (14)$$

Here $X_{\text{MMSE}} = \mathbb{E}[X \mid Y]$ is the MMSE estimator corresponding to the lower-right endpoint of the D-P curve. Moreover, an estimator achieving $D(P)$ for any given $P$ can be constructed as

$$\hat{X}_P = \left(1 - \frac{P}{P^*}\right)\hat{X}_0 + \frac{P}{P^*}X_{\text{MMSE}}, \quad (15)$$

where $\hat{X}_0$ is an estimator achieving $D(0)$, which corresponds to the upper-left endpoint of the D-P curve. This immediately provides a principled way to traverse the D-P curve via linear interpolation between a perceptually optimal estimator and the MMSE solution. In this work, we focus on such MSE-$W_2$ tradeoff.

## 3. Stage-wise distortion-perception traversal with diffusion models

In this section, we propose a two-stage framework for D-P traversal, termed MAP-RPS. In the first stage, we employ MAP estimation to approximate the MMSE solution, corresponding to the optimal distortion point, and we provide the approximation error bound together with an efficient implementation. In the second stage, we reintroduce noise to the MAP solution and perform posterior sampling, and theoretically show that the perception quality can be controlled by adjusting the re-noising timestep. Moreover, we present a latent-space variant of our method, termed LMAP-RPS, which further improves the generality and applicability of the proposed framework.

## 3.1. Stage 1: MMSE approximation for strongly log-concave posteriors

We first aim to approximate the optimal distortion point, which corresponds to the MMSE estimator under the MSE criterion. Recall that computing the MMSE estimator for an arbitrary posterior distribution requires evaluating the posterior mean $X_{\mathrm{MMSE}} = \mathbb{E}[X \mid Y]$. Although diffusion-based posterior sampling methods can, in principle, generate samples from the posterior distribution, accurately estimating the posterior mean typically requires a large number of samples. This becomes particularly costly for zero-shot diffusion-based inverse problem solvers, as each sample often entails hundreds of neural network evaluations. To address this challenge, we analyze posteriors that are strongly log-concave and establish an approximation error bound for using the MAP estimator $X_{\mathrm{MAP}}$ as a surrogate for the MMSE solution. Building on this analysis, we further develop an efficient diffusion-based algorithm to obtain the MAP estimate.

The strong log-concavity of a probability measure is defined as follows.

**Definition 3.1.** A probability measure $p(\mathbf{x})$ is said to be $\mu$-strongly log-concave if its negative log-density $-\log p(\mathbf{x})$ is $\mu$-strongly convex, i.e.,

$$-\nabla^2 \log p(\mathbf{x}) \succeq \mu \mathbf{I}, \tag{16}$$

for all $\mathbf{x}$ in the support of $p$.

Strong log-concavity implies unimodality and induces sharp concentration of measure around the global mode. It has been validated that approximating the posterior of inverse problems using strongly log-concave distributions is feasible and theoretically well-grounded (Bohr & Nickl, 2024). This assumption is also intuitive for general image restoration tasks, where a unique or dominant "ground-truth" image typically exists, implying that the posterior distribution is highly concentrated and approximately unimodal.

Under this condition, Theorem 3.2 establishes an explicit bound on the expected error incurred when approximating the MMSE estimator by the MAP estimator.

**Theorem 3.2.** *Assume that the posterior distribution $p_{X|Y}$ is $\mu$-strongly log-concave and satisfies*

$$\mu \geq 2\pi \left( \sup p_{X|Y} \right)^{-n_x/2}, \tag{17}$$

*for almost every realization of $Y$. Then the MAP approximation for the MMSE estimator satisfies*

$$\mathbb{E}\|X_{MAP} - X_{MMSE}\| \leq \sqrt{n_x/\mu}, \tag{18}$$

*and moreover,*

$$\mathbb{E}\|X - X_{MAP}\|^2 \leq D^* + \frac{n_x}{\mu}, \tag{19}$$

*where $n_x$ denotes the dimension of $X$, and $D^*$ is the optimal distortion error achieved by the MMSE estimator.*

All proofs for this section are deferred to Appendix B. Theorem 3.2 indicates that the expected error incurred by approximating the MMSE estimator with the MAP estimator is bounded by $\mathcal{O}\left(n_x^{1/2}\right)$. We note that in many practical inverse problems, the posterior distribution is highly concentrated, corresponding to a large strong log-concavity parameter $\mu$. In this regime, the approximation error can be further tightened, which supports the adoption of the MAP estimator as a principled surrogate for the MMSE solution. We note that though the theoretical analysis relies on the strong log-concavity assumption, empirical observations in Appendix E.1 indicate that the proposed approximation remains effective even in more general settings.

Solving the MAP problem is substantially less computationally demanding than computing the MMSE. By Bayes' rule, the MAP estimate can be written as

$$\mathbf{x}_{\mathrm{MAP}} = \arg \max_{\mathbf{x}} \log p_{Y|X}(\mathbf{y}|\mathbf{x}) + \log p_X(\mathbf{x}). \tag{20}$$

The likelihood term $\log p_{Y|X}(\mathbf{y} \mid \mathbf{x})$ is explicitly tractable and reduces to an $\ell_2$ term with Gaussian observation noise. Consequently, the MAP objective admits gradient-based optimization that relies solely on the prior score $\nabla_{\mathbf{x}} \log p_X(\mathbf{x})$.

Given a pretrained diffusion model that provides an estimation of the score function $\nabla_{\mathbf{x}_t} \log p_t(\mathbf{x}_t)$, the following theorem establishes a tractable approximation of the prior gradient.

**Theorem 3.3.** *Assume that, for some $t_1$ and $r_{t_1}$, the conditional distribution satisfies*

$$p_{X_0|X_{t_1}}(\mathbf{x}_0 \mid \mathbf{x}_{t_1}) \propto \mathcal{N}(\mathbb{E}(X_0 \mid X_{t_1} = \mathbf{x}_{t_1}), r_{t_1}^2 I), \tag{21}$$

*then the gradient of the log-prior follows*

$$\nabla_{\mathbf{x}} \log p_X(\mathbf{x}) = \frac{1 - \overline{\alpha}_{t_1}}{r_{t_1}^2 \sqrt{\overline{\alpha}_{t_1}}} \mathbb{E}_{p_{X_{t_1}|X_0}}(\mathbf{x}_{t_1}|\mathbf{x}) \nabla_{\mathbf{x}_{t_1}} \log p_{t_1}(\mathbf{x}_{t_1}).$$

Theorem 3.3 provides a practical way to estimate the gradient of the log-prior using a diffusion model, thereby enabling MAP inference via stochastic gradient-based optimization. We note that the assumption (21) underlying Theorem 3.3, which has also been adopted in prior work (e.g., (Song et al., 2023)), is approximately satisfied when either the timestep $t_1$ is sufficiently small or the prior $p_X(\mathbf{x})$ is well dominated by a unimodal Gaussian with high probability. Empirically, we observe that choosing a small $t_1$ already leads to accurate MAP estimates in practice. Further discussion of this assumption is provided in Appendix F.1.

Together, the above results complete Stage 1 and yield a tractable approximation of the MMSE estimator, providing a low-distortion starting point for subsequent D-P traversal.

## 3.2. Stage 2: Traversing the D-P curve via re-noised posterior sampling

In Stage 2, we traverse the D-P curve by first injecting noise into the MAP estimate obtained in Stage 1 up to a prescribed timestep $t_0$, and then performing posterior sampling from the resulting initialization to improve perception quality. The motivation stems from the observation that vanilla diffusion-based posterior sampling draws samples following $X_{\mathrm{PS}} \sim p_{X|Y}(\mathbf{x} \mid \mathbf{y})$, whose marginal distribution satisfies $p_{X_{\mathrm{PS}}} = p_X$ almost everywhere. Consequently, the posterior sampling estimator achieves optimal perception error as $W_2(p_X, p_{X_{\mathrm{PS}}}) = 0$. By controlling the re-noising time $t_0$, our method enables a continuous interpolation between the MAP estimator and the posterior sampling estimator. In particular, when $t_0 = T$, the proposed procedure recovers standard diffusion-based posterior sampling.

To analyze the evolution of the perception metric during Stage 2, we introduce the following notation. Let $\mathcal{T}(s,t)_\#$ denote the pushforward operator associated with the forward SDE (1) from $s$ to $t$, for any $s < t$. Similarly, let $\tilde{\mathcal{T}}(t,s;\mathbf{y})_\#$ denote the pushforward operator induced by the posterior sampling SDE (6) from $t$ to $s$ conditioned on the observation $\mathbf{y}$. We impose the following assumptions on the approximate posterior score function $\mathbf{s}_\theta(\mathbf{x}_t, \mathbf{y}, t)$.

**Assumption 3.4.** We assume

(A.1) there exists a global constant $L_s$ such that $\mathbf{s}_\theta$ is one-sided $L_s$-Lipschitz with respect to its first argument;

(A.2) there exists a finite constant $\epsilon_{\mathrm{score}}$ such that the weighted posterior score estimation error satisfies

$$\int_0^T g^2(r)\, (\overline{\alpha}_r)^{\frac{1}{2} - L_s}\, \Delta_r^{1/2}\, \mathrm{d}r \le \epsilon_{\mathrm{score}},$$

where

$$\Delta_r := \mathbb{E}_{p_r(\mathbf{x}_r|\mathbf{y})} \left\| \nabla_{\mathbf{x}_r} \log p_r(\mathbf{x}_r \mid \mathbf{y}) - s_\theta(\mathbf{x}_r, \mathbf{y}, r) \right\|^2 ;$$

(A.3) all regularity conditions imposed in Kwon et al. (2022) hold for $p_{X|Y}$ and its associated SDE.

Then the following theorem establishes an upper bound on the $W_2$ distance in the re-noised posterior sampling stage.

**Theorem 3.5.** *Denote the estimator obtained by the proposed re-noised posterior sampling scheme as*

$$p_{0 \to t_0 \to 0}(\mathbf{x} \mid \mathbf{y}) := \tilde{\mathcal{T}}(t_0, 0; \mathbf{y})_\# \, \mathcal{T}(0, t_0)_\# \, \delta(\mathbf{x} - \mathbf{x}_{\mathrm{MAP}}),$$

*and let $p_{0 \to t_0 \to 0}(\mathbf{x})$ denote the marginal distribution obtained by integrating out $\mathbf{y}$. Then under the assumptions of Theorem 3.2 and Assumption 3.4, the $W_2$ distance is bounded as follows:*

$$W_2(p_X, \, p_{0 \to t_0 \to 0}(\mathbf{x})) \le (\overline{\alpha}_{t_0})^{1 - L_s} \sqrt{\frac{2 n_x}{\mu}} + \epsilon_{\mathrm{score}}.$$

**Algorithm 1** MAP-RPS

**Require:** Observation $\mathbf{y}$, measurement operator $\mathcal{A}$, re-noising time $t_0$, timestep for gradient computing $t_1$, approximated posterior score $\mathbf{s}_\theta$, iterations of optimization $N$, step size $\gamma$

**Ensure:** Reconstructed sample $\hat{\mathbf{x}}_0$

1: **Stage 1: MAP estimation**
2: Initialize $\mathbf{x}^{(0)}$
3: **for** $n = 0, \ldots, N-1$ **do**
4:     Calculate stochastic gradient $\mathbf{g}_n$ for $\mathbf{x}^{(n)}$ following Theorem 3.3 and (4).
5:     $\mathbf{x}^{(n+1)} \leftarrow \mathbf{x}^{(n)} + \gamma \mathbf{g}_n$.
6: **end for**
7: Set $\mathbf{x}_{\mathrm{MAP}} \leftarrow \mathbf{x}^{(N)}$

8: **Stage 2: Re-noised posterior sampling**
9: Initialize $\mathbf{x}_{t_0} \sim \mathcal{T}(0, t_0)_\# \delta_{\mathbf{x}_{\mathrm{MAP}}}$
10: **for** $t = t_0, \ldots, 1$ **do**
11:     Update $\mathbf{x}_{t-1}$ by one Euler–Maruyama step of the posterior sampling SDE (6)
12: **end for**
13: **return** $\hat{\mathbf{x}}_0 \leftarrow \mathbf{x}_0$

The upper bound in Theorem 3.5 consists of an exponential decay term controlled by $\overline{\alpha}_{t_0}$, and an accumulated posterior score estimation error. When the posterior score is estimated with sufficient accuracy, the $W_2$ distance is dominated by the exponential term, which leads to the following corollary.

**Corollary 3.6.** *If $L_s < 1$, the upper bound in Theorem 3.5 is monotonically decreasing with respect to $t_0$.*

Corollary 3.6 implies that the perception error can be controlled by adjusting the re-noising timestep $t_0$: increasing $t_0$ reduces the $W_2$ distance. We note that the monotonic decay of the upper bound relies on a Lipschitz constant smaller than 1. Such an assumption is commonly adopted in a wide range of generative models, such as Wasserstein GANs (Arjovsky et al., 2017) and Contractive Residual Flows (Papamakarios et al., 2021), as well as in several Bayesian inverse problem solvers such as Proximal Gradient RED (Reehorst & Schniter, 2018).

For solving the posterior sampling SDE, any existing posterior sampling algorithm can be employed, such as DPS (Chung et al., 2023), ΠGDM (Song et al., 2023), or even more general inverse problem solvers, as most of them can be unified under the posterior sampling framework (Daras et al., 2024; Zhang et al., 2025b). Further discussion on the choice and rationale of different posterior samplers is provided in Appendix E.3. Taken together, we achieve the MAP-RPS method accompanied by theoretical analysis and a practical implementation. The pseudocode of the algorithm is presented in Algorithm 1.

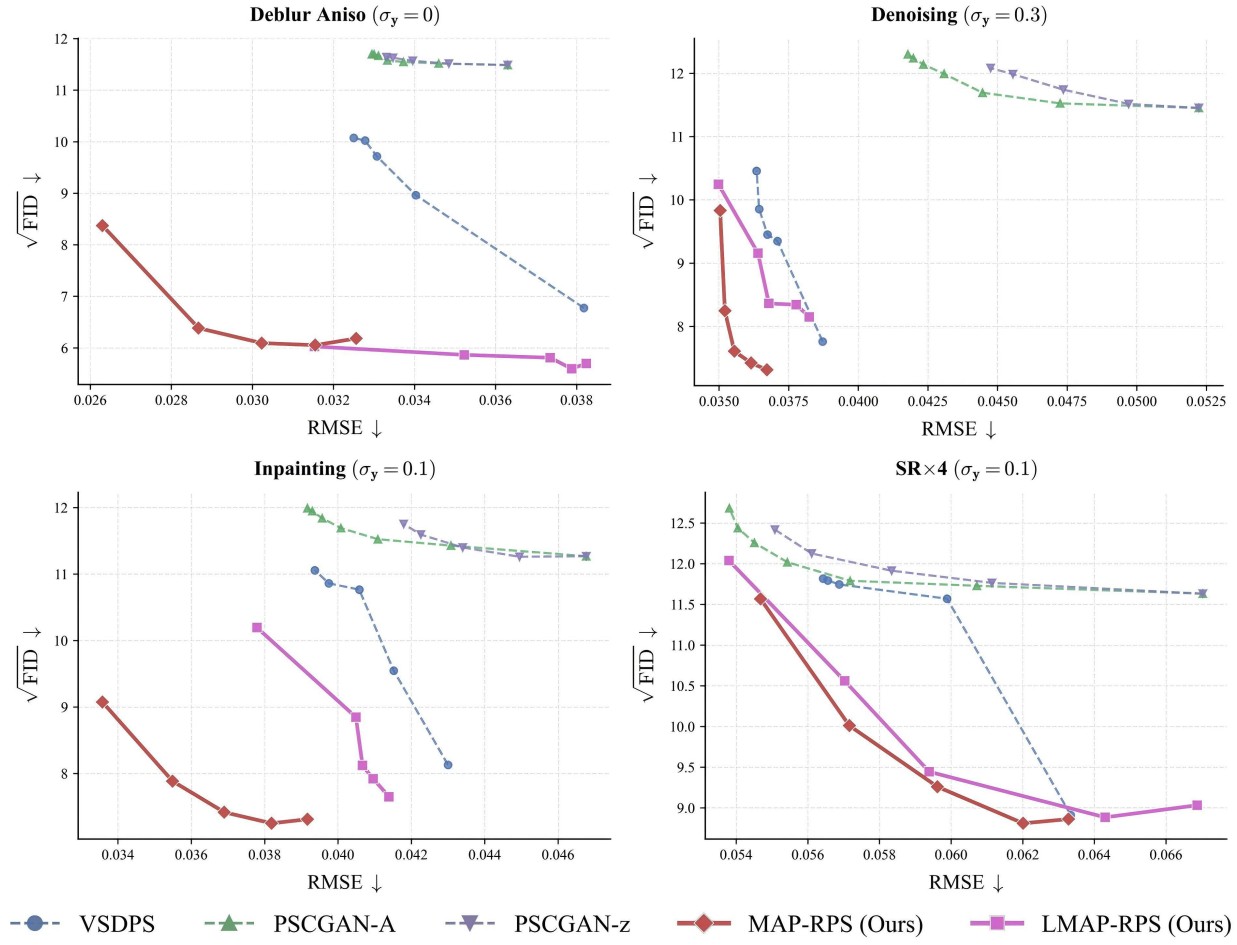

*Figure 1.* Distortion–Perception tradeoff of different algorithms on FFHQ.

### 3.3. MAP-RPS in latent space

In this section, we extend the proposed MAP-RPS framework to latent diffusion models, referred to as LMAP-RPS. Latent diffusion models introduce an encoder $\mathcal{E}$ and a decoder $\mathcal{D}$ to establish a mapping between the original data space and a lower-dimensional latent space in which a diffusion model is trained. We denote $Z \in \mathbb{R}^d$ the latent variable mapped from $X$ by the encoder $\mathcal{E}$ and $\mathbf{z}$ a realization.

The overall procedure of LMAP-RPS follows the same structure as MAP-RPS. The key difference lies in that both the MAP optimization and the subsequent re-noised posterior sampling are carried out in the latent space and then mapped back to the data space via the decoder. In particular, the latent likelihood term needs to be approximated as

$$\log p_{Y|Z}(\mathbf{y} \mid \mathbf{z}) \approx \log p_{Y|X}(\mathbf{y} \mid \mathcal{D}(\mathbf{z})), \qquad (22)$$

which applies the decoder output as a point estimate of $p_{X|Z}$, enabling the observation $\mathbf{y}$ to impose an explicit constraint on the latent variable.

We defer the complete theoretical guarantee for LMAP-RPS

to Theorem C.2 and Theorem C.3 in Appendix C. Implementation in the latent space endows LMAP-RPS with broader applicability, as most large-scale pretrained diffusion models are currently trained in latent spaces (e.g., text-to-image models). We demonstrate the practical potential of LMAP-RPS in the experiments section.

## 4. Experiments

In this section, we first evaluate the D-P traversal of MAP-RPS and LMAP-RPS, and then validate the effectiveness of LMAP-RPS on near-real-world image inverse problems.

### 4.1. Distortion–Perception traversal on FFHQ

We follow prior works (Wang et al., 2025; Ohayon et al., 2021) to evaluate D-P traversal on the FFHQ256 dataset (Karras et al., 2019). A total of 100 images are randomly sampled from the test set. The comparison methods include VSDPS (Wang et al., 2025), a pixel space diffusion model-based approach, and PSCGAN (Ohayon et al., 2021), a

*Table 1.* Quantitative comparison on near-real-world inverse problems on MS-COCO. LMAP-RPS (0) and LMAP-RPS (600) denote our LMAP-RPS algorithm with $t_0 = 0$ and $t_0 = 600$, respectively. **Bold** indicates the best result, and underline indicates the second-best.

| | Inpainting | | | | SR $4\times$ | | | | Deblur Aniso | | | |
|---|---|---|---|---|---|---|---|---|---|---|---|---|
| **Method** | PSNR↑ | SSIM↑ | LPIPS↓ | FID↓ | PSNR↑ | SSIM↑ | LPIPS↓ | FID↓ | PSNR↑ | SSIM↑ | LPIPS↓ | FID↓ |
| Latent-DPS | 27.08 | 0.7502 | 0.3252 | 101.05 | 24.27 | 0.6588 | 0.3566 | 90.82 | 24.22 | 0.6376 | 0.3838 | 117.23 |
| ReSample | 26.19 | 0.7302 | 0.3359 | 105.76 | 24.70 | 0.6761 | 0.3642 | 91.43 | 25.18 | 0.6507 | 0.3654 | 85.16 |
| PSLD | 26.64 | 0.7170 | 0.3620 | 83.08 | 24.20 | 0.6411 | 0.3892 | 100.96 | 23.26 | 0.5009 | 0.5394 | 138.19 |
| STSL | 26.39 | 0.7268 | 0.3545 | 121.04 | 24.72 | 0.6685 | 0.3920 | 105.60 | 23.86 | 0.5867 | 0.4526 | 124.94 |
| LDIR | 27.29 | 0.7709 | 0.3568 | 115.04 | 25.01 | 0.6973 | 0.4137 | 122.17 | 24.66 | 0.6749 | 0.4419 | 151.00 |
| Latent-DCDP | 26.71 | 0.7488 | 0.3189 | 72.05 | 22.79 | 0.5380 | 0.5252 | 183.22 | 25.39 | 0.6746 | 0.3535 | **81.27** |
| Latent-DMAP | 27.50 | 0.7875 | 0.3078 | 89.21 | 23.57 | 0.6286 | 0.3721 | 93.33 | 25.49 | 0.7039 | 0.3944 | 122.62 |
| Latent-DAPS | 26.81 | 0.7407 | 0.3385 | 74.24 | 22.43 | 0.4496 | 0.5132 | 181.31 | 24.06 | 0.6302 | 0.4323 | 136.19 |
| Latent-SITCOM | 28.06 | 0.7950 | 0.3097 | 81.16 | 24.09 | 0.6299 | 0.4402 | 141.22 | 26.28 | 0.7244 | 0.3550 | 106.35 |
| **LMAP-RPS (0)** | **28.14** | **0.7989** | **0.2769** | **61.35** | **25.03** | **0.6999** | 0.3888 | 107.57 | **26.42** | **0.7322** | 0.3504 | 90.80 |
| **LMAP-RPS (600)** | 27.22 | 0.7615 | 0.3084 | 92.23 | 24.49 | 0.6827 | **0.3505** | **87.20** | 25.59 | 0.6948 | **0.3503** | 85.01 |

| | CS $2\times$ | | | | HDR | | | | Nonlinear Deblur | | | |
|---|---|---|---|---|---|---|---|---|---|---|---|---|
| **Method** | PSNR↑ | SSIM↑ | LPIPS↓ | FID↓ | PSNR↑ | SSIM↑ | LPIPS↓ | FID↓ | PSNR↑ | SSIM↑ | LPIPS↓ | FID↓ |
| Latent-DPS | 21.57 | 0.6567 | 0.3912 | 139.85 | 22.67 | 0.6612 | 0.3999 | 117.96 | 22.15 | 0.5803 | 0.4281 | 133.63 |
| ReSample | 21.82 | 0.6753 | 0.3806 | 113.36 | 22.61 | 0.7259 | 0.3666 | 111.89 | 23.00 | 0.6030 | 0.4157 | 129.93 |
| PSLD | 21.57 | 0.6642 | 0.4011 | 119.91 | – | – | – | – | – | – | – | – |
| STSL | 19.92 | 0.6052 | 0.4410 | 174.21 | 20.45 | 0.5656 | 0.4638 | 211.78 | 21.39 | 0.5384 | 0.4639 | 163.77 |
| LDIR | 19.79 | 0.5712 | 0.4127 | 134.53 | 21.22 | 0.6317 | 0.4491 | 142.87 | 23.13 | 0.6307 | 0.4550 | 171.53 |
| Latent-DCDP | 20.54 | 0.6563 | 0.3834 | 129.78 | 23.31 | 0.6827 | 0.3789 | 117.87 | 22.88 | 0.5989 | 0.4225 | 140.32 |
| Latent-DMAP | 20.96 | 0.6784 | 0.3866 | 143.56 | 22.93 | 0.6902 | 0.3892 | 113.28 | 21.68 | 0.5893 | 0.4770 | 172.23 |
| Latent-DAPS | 19.50 | 0.5444 | 0.5204 | 208.00 | 22.81 | 0.6988 | 0.4000 | 109.66 | 21.62 | 0.5292 | 0.5006 | 197.71 |
| Latent-SITCOM | 19.74 | 0.5727 | 0.5074 | 210.49 | 23.26 | 0.7046 | 0.3911 | 123.49 | 22.06 | 0.6001 | 0.4669 | 188.24 |
| **LMAP-RPS (0)** | 22.87 | 0.7340 | 0.3498 | **113.06** | **25.86** | **0.7425** | **0.3595** | **108.79** | 24.27 | 0.6599 | 0.3942 | 119.43 |
| **LMAP-RPS (600)** | **22.90** | **0.7341** | **0.3497** | 113.58 | 23.06 | 0.6796 | 0.3944 | 116.08 | **24.29** | **0.6614** | **0.3933** | **118.69** |

conditional GAN-based baseline. For VSDPS and MAP-RPS, we employ the pretrained diffusion model from Chung et al. (2023); for LMAP-RPS, we use the checkpoint from Rombach et al. (2022).

Four inverse problems are considered: (1) denoising with noise standard deviation $\sigma_{\mathbf{y}} = 0.3$ (with respect to the data range $[0, 1]$, hereafter the same); (2) $4\times$ super-resolution with $\sigma_{\mathbf{y}} = 0.1$; (3) $50\%$ random inpainting with $\sigma_{\mathbf{y}} = 0.1$; and (4) anisotropic deblurring with $\sigma_{\mathbf{y}} = 0$. We note that PSCGAN was originally trained only for the denoising task with $\sigma_{\mathbf{y}} = 0.3$ on FFHQ at $128 \times 128$ resolution. To ensure a fair comparison, we fine-tune PSCGAN for 100 epochs on the target resolution, noise level, and task. We denote PSCGAN-A and PSCGAN-z as the variants obtained by sample averaging and controlling the input noise variance, respectively (Ohayon et al., 2021). For VSDPS, the step size is carefully adjusted for each variance scale to ensure a monotonic tradeoff. For MAP-RPS and LMAP-RPS, we fix $t_1 = 10$ and $t_1 = 50$, respectively. Further implementation details are provided in Appendix D.

Figure 1 presents the $\sqrt{\text{FID}}$-RMSE curves obtained by the

five algorithms. Both MAP-RPS and LMAP-RPS consistently lie closer to the lower-left corner across all tasks, indicating that they achieve a more favorable D-P tradeoff across diverse tasks and noise levels. Figure 2 further visualizes representative reconstruction results, and additional visualizations can be found in Appendix G. We note that, in most cases, MAP-RPS slightly outperforms LMAP-RPS, which is consistent with previous findings that pixel-space algorithms often outperform their latent-space counterparts (Zhang et al., 2025a; Alkhouri et al., 2025). This is likely due to the fact that pixel-space diffusion avoids potential information loss and approximation errors introduced by the decoder of the VAE. More ablation studies and discussions are postponed to Appendix E and F due to space constraints.

### 4.2. Inverse problem solving on MS-COCO

We further evaluate the proposed framework as a general-purpose inverse algorithm on near-real-world image inverse problems on MS-COCO (Lin et al., 2014). Stable Diffusion v1.5 (Rombach et al., 2022) is adopted as the backbone latent diffusion model. To adapt the images to the input

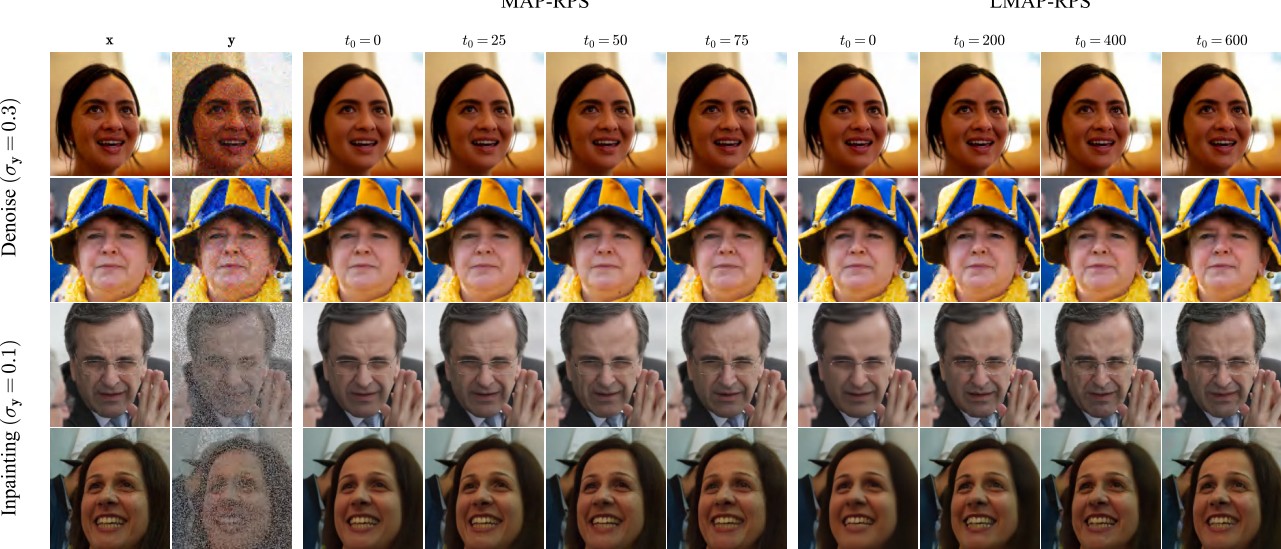

*Figure 2.* Visualizations of MAP-RPS and LMAP-RPS with varying $t_0$.

*Table 2.* Inference time on MS-COCO (seconds per image).

| Method | Inp | SR $4\times$ | CS $2\times$ | HDR |
|---|---|---|---|---|
| Latent-DPS | 242 | 250 | 240 | 241 |
| ReSample | 1102 | 1205 | 1327 | 1104 |
| PSLD | 262 | 273 | 265 | – |
| STSL | 444 | 457 | 463 | 457 |
| LDIR | 294 | 303 | 295 | 293 |
| Latent-DCDP | 110 | 224 | 334 | 110 |
| Latent-DMAP | 283 | 294 | 284 | 284 |
| Latent-DAPS | 464 | 475 | 485 | 464 |
| Latent-SITCOM | 269 | 211 | 342 | 280 |
| **LMAP-RPS (0)** | 45 | 23 | 68 | 89 |
| **LMAP-RPS (600)** | 189 | 172 | 213 | 230 |

resolution, we first filter the test set to retain only images with both height and width greater than 512 pixels, then resize the shorter side to 512 pixels and apply center cropping. 100 images are randomly sampled for evaluation from the filtered set. We consider six widely adopted inverse problems, including four linear tasks (50% random inpainting, $4\times$ super-resolution, anisotropic deblurring, and $2\times$ compressed sensing) and two nonlinear tasks (high dynamic range reconstruction and nonlinear deblurring). For all tasks, Gaussian observation noise with $\sigma_{\mathbf{y}} = 0.05$ is added.

We evaluate the performance of LMAP-RPS with two fixed re-noising timesteps: $t_0 = 0$, corresponding to directly decoding the MAP estimate, and $t_0 = 600$. The timestep for computing the prior gradient is fixed at $t_1 = 50$. By default, Latent-DPS (Song et al., 2024) is used for the posterior sampling stage, except for the anisotropic deblurring task, where we employ Latent-DCDP (Li et al., 2024b). Comparisons are performed against nine recent SOTA latent diffusion-

based inverse algorithms: Latent-DPS (Song et al., 2024), ReSample (Song et al., 2024), PSLD (Rout et al., 2023), STSL (Rout et al., 2024), LDIR (He et al., 2023), Latent-DCDP (Li et al., 2024b), Latent-DMAP (Xu et al., 2025), Latent-DAPS (Zhang et al., 2025a), and Latent-SITCOM (Alkhouri et al., 2025). For all methods, the classifier-free guidance weight is fixed at 1.5, and other key hyperparameters are optimized via grid search on two validation images to ensure competitive performance.

We report two distortion metrics (PSNR and SSIM (Wang et al., 2004)) and two perception metrics (LPIPS (Zhang et al., 2018) and FID (Heusel et al., 2017)) as evaluation criteria. Table 1 presents the quantitative performance of LMAP-RPS and comparison baselines. LMAP-RPS achieves SOTA results across nearly all tasks and metrics. Table 2 also summarizes part of the computational overhead for all algorithms. LMAP-RPS at $t_0 = 0$ is significantly faster than all baselines, and remains comparatively efficient at $t_0 = 600$, highlighting its computational efficiency. Qualitative results are postponed to Appendix G. Notably, even at $t_0 = 0$, the pure MAP outputs already achieve highly competitive performance, indicating that diffusion-based inverse problem solving may not always require iterative annealing schedules. Employing a single fixed timestep suffices to achieve strong performance.

We note that on some tasks, such as random inpainting and HDR, LMAP-RPS with large $t_0$ may degrade both distortion and perception metrics. This behavior is task-dependent: in these experiments on MS-COCO, the observation noise is $\sigma_{\mathbf{y}} = 0.05$, which is smaller than the noise level used in the D-P traversal experiments. Such a low-noise setting induces a more concentrated posterior, making distortion

and perception metrics more aligned.

## 5. Conclusion

In this paper, we propose MAP-RPS, a stage-wise method for traversing the distortion-perception tradeoff in zero-shot inverse problems using a single diffusion model. The approach first employs MAP estimation to approximate the MMSE solution, providing a low-distortion initialization, and then progressively improves perceptual performance through a re-noised posterior sampling stage. Extensive experiments demonstrate that the proposed MAP-RPS algorithm, along with its latent-space counterpart LMAP-RPS, achieves closer approximations to optimal tradeoff and attains SOTA performance on practical inverse problems.

**Limitations and future work.** Our theoretical analysis indicates that MAP-RPS and LMAP-RPS inevitably incur approximation errors when approaching the optimal tradeoff curve. Specifically, in Stage 1, approximating the MMSE solution with the MAP estimate is more accurate when the posterior is approximately unimodal, while its effectiveness for complex multimodal posteriors requires further investigation. The upper bound in Theorem 3.2 is established under strongly log-concave posteriors, and extending the analysis to more general posterior distributions remains an important direction for future work. Meanwhile, Stage 2 in this paper mainly relies on DPS and its variants, which introduce additional estimation errors in the posterior score. It would be promising to incorporate asymptotically consistent posterior sampling methods (Xu & Chi, 2024) into Stage 2 to further improve the D-P traversal of MAP-RPS.

## Acknowledgements

This work was supported by the National Key Research and Development Program of China (Grant No. 2025YFF0515600) and the National Natural Science Foundation of China (NSAF U2230201).

## Impact Statement

This paper presents work whose goal is to advance the field of Machine Learning. To empirically evaluate the proposed methods, we conduct experiments on widely used facial benchmarks such as FFHQ. While our method is general-purpose, we acknowledge that the FFHQ dataset used in this paper has demographic imbalances, which may cause our results to fail to generalize to underrepresented groups. A more diverse and balanced dataset is needed in the field to ensure fairness across factors such as ethnicity, gender, and age. A more representative diffusion model trained on such data is also preferred to better support real-world applications.

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

# Appendix

In the appendix, we provide additional discussions and technical details that are omitted from the main paper due to space limitations. Appendix A provides a more detailed discussion of strategies for navigating the D–P tradeoff in diffusion-based inverse problem solving, and clarifies how our work differs from and complements existing approaches. Appendix B presents detailed proofs of Theorems 3.2, 3.3, and 3.5 related to MAP-RPS. Appendix C derives the formal error bound for LMAP-RPS, establishing its theoretical soundness. Appendix D provides comprehensive experimental details, including inverse problem settings and all hyperparameter configurations. Appendix E provides additional experimental results and comparisons, as well as further discussions on closely related methods. Appendix F includes ablation studies and a more thorough analysis of the observed empirical behaviors. Finally, Appendix G presents further qualitative visualization results.

## A. More related works

Since the formal introduction of the D-P tradeoff, a growing body of work has investigated how to explicitly traverse the D–P curve while solving inverse problems. Whang et al. (2022) identify two generic mechanisms for controlling the D-P tradeoff: sample averaging and adjusting the number of sampling steps. The idea of sample averaging is conceptually straightforward. If an algorithm is capable of drawing samples from the posterior distribution $p_{X|Y}$, it achieves zero perception error $P = 0$, and the MMSE estimator can be asymptotically approximated by averaging an increasing number of samples. Consequently, using fewer samples typically results in lower perception error but higher distortion error, whereas increasing the number of samples improves distortion quality at the cost of higher perception error. The second mechanism relies on controlling the number of discretization steps of the reverse SDE. It is commonly argued that each sampling step injects additional stochasticity, which enhances sample diversity but increases distortion error. As a result, reducing the number of sampling steps tends to lower distortion while sacrificing perception due to reduced randomness, whereas more steps improve perception at the expense of distortion performance. Both strategies have been applied and empirically validated in subsequent studies. For example, the steps controlling method appears in Delbracio & Milanfar (2023), Ren et al. (2023), Yue et al. (2024), Luo et al. (2024), Zhussip et al. (2024), and Cohen et al. (2025), while sample averaging is adopted in Li et al. (2024a). We note that sample averaging is computationally expensive for diffusion-based zero-shot inverse algorithms, as each sample typically requires hundreds of iterative steps. Meanwhile, adjusting the number of sampling steps is often heuristic. In Appendix F.4, we further explore the compatibility of the proposed MAP-RPS with both strategies.

In the context of zero-shot diffusion-based inverse algorithms, some works have empirically observed that tuning certain algorithmic hyperparameters enables partial control over the D-P tradeoff. For example, Mardani et al. (2024) and Zhang et al. (2024a) report that adjusting the learning-rate hyperparameter in super-resolution tasks affects the PSNR-LPIPS tradeoff, while Zhang et al. (2025b) achieve D-P traversal by reweighting the loss term of learnable high-order solvers. More recently, several works have begun to address the D-P tradeoff from a more principled perspective. Dornbusch et al. (2025) leverage the theoretical MSE-$W_2$ tradeoff in the denoising task and proposes to linearly combine a better-perception solution and a low-distortion solution. The key idea is to treat a single-step solution as a low-distortion estimator, while using the full trajectory solution as an estimator with low perception error. Another representative approach is VSDPS (Wang et al., 2025), which provides a more theoretically grounded mechanism for controlling the D-P tradeoff. From the perspective of mean propagation (Xue et al., 2024), VSDPS rescales the noise variance in posterior sampling and considers the following SDE:

$$\mathrm{d}\mathbf{x}_t = \left(f(t)\mathbf{x}_t - g^2(t)\nabla_{\mathbf{x}_t} \log p_t(\mathbf{x}_t|\mathbf{y})\right)\mathrm{d}t + \eta g(t)\mathrm{d}\overline{\mathbf{w}}_t, \tag{23}$$

where $\eta$ is a variance scaling factor. When $\eta = 1$, the process reduces to standard posterior sampling and yields low perception error, while when $\eta = 0$, the dynamics degenerate toward the posterior mean. It is shown that (Wang et al., 2025) when the posterior distribution is Gaussian and the discretization step size tends to zero, varying $\eta$ enables traversal of the optimal D-P curve. In practice, the posterior score is typically approximated using DPS, and VSDPS often requires jointly tuning the step size $\xi$ in DPS together with $\eta$ to achieve a desirable tradeoff.

The proposed MAP-RPS and LMAP-RPS adopt a different perspective from VSDPS. Our methods first approximate one endpoint of the D-P curve, namely the optimal distortion solution, and then gradually transition toward improved perceptual quality. This philosophy is conceptually more related to sample averaging, which starts from the optimal perception solution and moves towards lower distortion. We note that our approaches are significantly more computationally efficient. In the worst case, they require only a single MAP estimation procedure and one full posterior sampling process, without repeatedly drawing samples. This is particularly advantageous since repeated sampling is expensive for diffusion-based

inverse algorithms.

# B. Theoretical analysis for MAP-RPS

## B.1. Assumptions and lemmas

We first collect and restate all assumptions required for the theoretical analysis of MAP-RPS.

**Assumption B.1.** We assume the following conditions:

(B.1.1) The posterior distribution $p_{X|Y}$ admits a $\mu$-strongly log-concave density. That is, for all $\mathbf{x}$,

$$-\nabla_{\mathbf{x}}^2 \log p_{X|Y}(\mathbf{x} \mid \mathbf{y}) \succeq \mu \mathbf{I}. \tag{24}$$

Moreover, we assume

$$\mu \geq 2\pi (\sup p_{X|Y})^{-n_x/2}, \tag{25}$$

uniformly for all realizations of $Y$.

(B.1.2) There exists a global constant $L_s$ such that the approximate posterior score $\mathbf{s}_\theta$ is one-sided $L_s$-Lipschitz with respect to its first argument, i.e.,

$$\|\mathbf{s}_\theta(\mathbf{x}_1, \mathbf{y}, t) - \mathbf{s}_\theta(\mathbf{x}_2, \mathbf{y}, t)\|_2 \leq L_s \|\mathbf{x}_1 - \mathbf{x}_2\|_2, \tag{26}$$

for all $\mathbf{x}_1, \mathbf{x}_2 \in \text{supp}(X_t)$.

(B.1.3) There exists a universal constant $\epsilon_{\text{score}}$ such that the weighted score estimation error is bounded as

$$\int_0^T g^2(r) \left(\overline{\alpha}_r\right)^{\frac{1}{2} - L_s} \left[\mathbb{E}_{p_r(\mathbf{x}_r|\mathbf{y})} \|\nabla_{\mathbf{x}_r} \log p_r(\mathbf{x}_r|\mathbf{y}) - \mathbf{s}_\theta(\mathbf{x}_r, \mathbf{y}, r)\|^2\right]^{1/2} \mathrm{d}r \leq \epsilon_{\text{score}}. \tag{27}$$

(B.1.4) There exist $t_1$ and $r_{t_1}$ such that

$$p_{X_0|X_{t_1}}\left(\mathbf{x}_0 \mid \mathbf{x}_{t_1}\right) \propto \mathcal{N}\left(\mathbb{E}_{p_{X_0|X_{t_1}}}\left(\mathbf{x}_0 \mid \mathbf{x}_{t_1}\right), r_{t_1}^2 \mathbf{I}\right). \tag{28}$$

(B.1.5) We further assume that all regularity conditions imposed in Kwon et al. (2022) hold for the posterior distribution $p_{X|Y}$ and its associated SDE. These conditions ensure the basic well-posedness of the SDE solution.

Now we present several technical lemmas useful for our analysis. We begin with the following lemma (Brascamp & Lieb, 1976) that characterizes a fundamental property of strongly log-concave densities.

**Lemma B.2.** *(Brascamp & Lieb, 1976) Suppose that $-\log p(\mathbf{x})$ is twice continuously differentiable and strongly convex. Then for any test function $h$, the following inequality holds:*

$$\mathbb{E}|h(\mathbf{x}) - \mathbb{E}(h(\mathbf{x}))|^2 \leq \mathbb{E}\left[\nabla h(\mathbf{x})^T \left(\nabla^2(-\log p(\mathbf{x}))\right)^{-1} \nabla h(\mathbf{x})\right]. \tag{29}$$

The next lemma establishes a differential equation governing the evolution of the $W_2$ distance along the posterior sampling SDE, which is a posterior-sampling counterpart of Proposition 2 in Kwon et al. (2022).

**Lemma B.3.** *(Kwon et al., 2022). Let $p_t(\cdot, \mathbf{y})$ denote the solution of the forward diffusion SDE (1) with initial condition $\mathbf{x}_0 \sim p_{X|Y}(\mathbf{x} \mid \mathbf{y})$, and let $q_t(\cdot, \mathbf{y})$ denote the solution of the posterior sampling SDE (6) where the posterior score is approximated by $\mathbf{s}_\theta$. Under Assumption (B.1.2) and (B.1.5), the following inequality holds:*

$$-\frac{\mathrm{d}}{\mathrm{d}t} W_2(p_t, q_t) \leq \left(f(t) + L_s g^2(t)\right) W_2(p_t, q_t) + g^2(t) \Delta_t, \tag{30}$$

*where*

$$\Delta_t = [\mathbb{E}_{p_t} \|\nabla_{\mathbf{x}_t} \log p_t(\mathbf{x}_t \mid \mathbf{y}) - \mathbf{s}_\theta(\mathbf{x}_t, \mathbf{y}, t)\|^2]^{1/2}. \tag{31}$$

The following lemma is adapted from Theorem 1 in Kwon et al. (2022). We extend the result to a general time $t$ and provide explicit coefficients for the VP setting.

**Lemma B.4.** *Let $q_{t_0}(\mathbf{x}_{t_0} \mid \mathbf{y})$ be an initial distribution at timestep $t_0$. Consider solving the reverse-time SDE from $q_{t_0}$ using $\mathbf{s}_\theta$ as an approximation of the posterior score, then under Assumption (B.1.2) and (B.1.5),*

$$W_2\left(p_{X|Y}(\mathbf{x} \mid \mathbf{y}), \tilde{\mathcal{T}}(t_0, 0, \mathbf{y})_{\#} q_{t_0}(\mathbf{x}_{t_0} \mid \mathbf{y})\right) \leq (\overline{\alpha}_{t_0})^{\frac{1}{2} - L_s} W_2(p_{t_0}(\mathbf{x}_{t_0} \mid \mathbf{y}), q_{t_0}(\mathbf{x}_{t_0} \mid \mathbf{y})) + \int_0^{t_0} g^2(r) (\overline{\alpha}_r)^{\frac{1}{2} - L_s} \Delta_r \mathrm{d}r.$$
(32)

*Proof.* Under the VP setting, we have

$$\int_0^t \left(f(r) + L_s g^2(r)\right) \mathrm{d}r = \int_0^t \mathrm{d}\log\sqrt{\overline{\alpha}_r} + L_s \int_0^t \left(\mathrm{d}(1 - \overline{\alpha}_r) - (1 - \overline{\alpha}_r)\mathrm{d}\log\overline{\alpha}_r\right) = \log\sqrt{\overline{\alpha}_t} - L_s \log\overline{\alpha}_t.$$
(33)

Then by Lemma B.3, the Wasserstein distance $W_2(p_t, q_t)$ satisfies

$$-\frac{\mathrm{d}}{\mathrm{d}t}\left(W_2(p_t, q_t)\overline{\alpha}_t^{\frac{1}{2} - L_s}\right) \leq g^2(t)\overline{\alpha}_t^{\frac{1}{2} - L_s}\Delta_t.$$
(34)

Integrating both sides from $0$ to $t_0$ yields

$$-W_2\left(p_{t_0}(\mathbf{x}_{t_0} \mid \mathbf{y}), q_{t_0}(\mathbf{x}_{t_0} \mid \mathbf{y})\right)\overline{\alpha}_{t_0}^{\frac{1}{2} - L_s} + W_2\left(p_{X|Y}(\mathbf{x} \mid \mathbf{y}), \tilde{\mathcal{T}}(t_0, 0, \mathbf{y})_{\#} q_{t_0}(\mathbf{x}_{t_0} \mid \mathbf{y})\right)\overline{\alpha}_{t_0}^{\frac{1}{2} - L_s} \leq \int_0^{t_0} g^2(r)\overline{\alpha}_r^{\frac{1}{2} - L_s}\Delta_r \mathrm{d}r.$$
(35)

Noting that $\overline{\alpha}_0 = 1$ and rearranging the terms completes the proof. □

The following lemma shows that the $W_2$ distance is contractive under convolution with an independent random variable.

**Lemma B.5.** *Let $X$ and $\hat{X}$ be two random variables, and let $Z$ be a random variable independent of both $X$ and $\hat{X}$. Then*

$$W_2\left(p_{X+Z}, p_{\hat{X}+Z}\right) \leq W_2\left(p_X, p_{\hat{X}}\right).$$
(36)

*Proof.* For any coupling $\left(X, \hat{X}\right) \sim \pi \in \Pi\left(p_X, p_{\hat{X}}\right)$, we can construct a new coupling

$$(X + Z, \hat{X} + Z) \sim \overline{\pi} \in \Pi\left(p_{X+Z}, p_{\hat{X}+Z}\right)$$
(37)

by exploiting the independence of $Z$ from $(X, \hat{X})$. With this coupling, we have

$$\int \|X_1 - X_2\|^2 \mathrm{d}\overline{\pi}(X_1, X_2) = \mathbb{E}_Z \int \|X + Z - (\hat{X} + Z)\|^2 \mathrm{d}\pi(X, \hat{X}) = \int \|X - \hat{X}\|^2 \mathrm{d}\pi(X, \hat{X}).$$
(38)

Since the construction holds for arbitrary coupling $\pi \in \Pi\left(p_X, p_{\hat{X}}\right)$, taking the infimum yields

$$W_2\left(p_{X+Z}, p_{\hat{X}+Z}\right) \leq \inf_{\pi \in \Pi\left(p_X, p_{\hat{X}}\right)} \int \|X_1 - X_2\|^2 \mathrm{d}\overline{\pi}(X_1, X_2) = \inf_{\pi \in \Pi\left(p_X, p_{\hat{X}}\right)} \int \|X - \hat{X}\|^2 \mathrm{d}\pi\left(X, \hat{X}\right) = W_2\left(p_X, p_{\hat{X}}\right).$$
(39)

This completes the proof. □

## B.2. Proof of Theorem 3.2

*Proof.* Under Assumption (B.1.1), we have

$$\nabla_{\mathbf{x}}^2(-\log p_{X|Y}(\mathbf{x} \mid \mathbf{y})) \succeq \mu\mathbf{I}.$$
(40)

Let $f_i(\mathbf{x}) = \langle \mathbf{e}_i, \mathbf{x} \rangle$, where $\mathbf{e}_i$ denotes the $i$-th canonical basis vector. Applying Lemma B.2 yields

$$\mathbb{E}_{p_{X|Y}} |\langle \mathbf{e}_i, X - \mathbb{E}(X \mid Y) \rangle|^2 \le \mathbb{E}\left[ \mathbf{e}_i^\top \frac{1}{\mu} \mathbf{I} \mathbf{e}_i \right] = \frac{1}{\mu}. \tag{41}$$

Summing over all coordinates gives

$$\mathbb{E}_{p_{X|Y}} \|X - X_{\text{MMSE}}\|^2 = \sum_{i=1}^{n_x} \mathbb{E}_{p_{X|Y}} |\langle \mathbf{e}_i, X - \mathbb{E}(X \mid Y) \rangle|^2 \le \frac{n_x}{\mu}. \tag{42}$$

where $X_{\text{MMSE}} = \mathbb{E}[X \mid Y]$. Now we denote the MAP solution by $\mathbf{x}_{\text{MAP}} = \arg\max_{\mathbf{x}} p_{X|Y}(\mathbf{x} \mid \mathbf{y})$. Then under the strong log-concavity, we have

$$p_{X|Y}(\mathbf{x} \mid \mathbf{y}) \le p_{X|Y}(\mathbf{x}_{\text{MAP}} \mid \mathbf{y}) \exp\left( -\frac{\mu}{2} \|\mathbf{x} - \mathbf{x}_{\text{MAP}}\|^2 \right). \tag{43}$$

Conditioning on a realization of $Y$, the squared distance between the MAP and MMSE estimators can be bounded as

$$\begin{aligned}
\|X_{\text{MAP}} - X_{\text{MMSE}}\|^2 = \left\| X_{\text{MAP}} - \int X p_{X|Y} \mathrm{d}X \right\|^2 &\le \int \|X_{\text{MAP}} - X\|^2 p_{X|Y} \mathrm{d}X \\
&\le \sup p_{X|Y} \int \|X_{\text{MAP}} - X\|^2 \exp\left( -\frac{\mu}{2} \|X - X_{\text{MAP}}\|^2 \right) \mathrm{d}X \\
&= \frac{n_x}{\mu} \sup p_{X|Y} \, (2\pi/\mu)^{n_x/2} \\
&\le \frac{n_x}{\mu},
\end{aligned} \tag{44}$$

where the last inequality follows from Assumption (B.1.1). Taking expectations yields

$$\mathbb{E}\|X_{\text{MAP}} - X_{\text{MMSE}}\| \le \sqrt{n_x/\mu}. \tag{45}$$

Finally,

$$\mathbb{E}\|X - X_{\text{MAP}}\|^2 = \mathbb{E}\|X - X_{\text{MMSE}}\|^2 + \mathbb{E}\|X_{\text{MMSE}} - X_{\text{MAP}}\|^2 \le D^* + \frac{n_x}{\mu}, \tag{46}$$

which completes the proof. $\square$

## B.3. Proof of Theorem 3.3

*Proof.* The gradient of the log prior can be written as

$$\begin{aligned}
\nabla_{\mathbf{x}} \log p_X(\mathbf{x}) &= \frac{1}{p_X(\mathbf{x})} \int p_{X_{t_1}}(\mathbf{x}_{t_1}) \nabla_{\mathbf{x}} p_{X|X_{t_1}}(\mathbf{x} \mid \mathbf{x}_{t_1}) \mathrm{d}\mathbf{x}_{t_1} \\
&= \int p_{X_{t_1}|X}(\mathbf{x}_{t_1} \mid \mathbf{x}) \nabla_{\mathbf{x}} \log p_{X|X_{t_1}}(\mathbf{x} \mid \mathbf{x}_{t_1}) \mathrm{d}\mathbf{x}_{t_1}.
\end{aligned} \tag{47}$$

Under Assumption (B.1.4), we have

$$\nabla_{\mathbf{x}} \log p_{X|X_{t_1}}(\mathbf{x} \mid \mathbf{x}_{t_1}) = -\frac{1}{r_{t_1}^2} \left( \mathbf{x} - \mathbb{E}_{p_{X|X_{t_1}}}[\mathbf{x} \mid \mathbf{x}_{t_1}] \right) = -\frac{1}{r_{t_1}^2} \left( \mathbf{x} - \frac{\mathbf{x}_{t_1} + (1 - \overline{\alpha}_{t_1}) \nabla \log p_{t_1}(\mathbf{x}_{t_1})}{\sqrt{\overline{\alpha}_{t_1}}} \right). \tag{48}$$

Combining (47) and (48) and noting that

$$p_{X_{t_1}|X}(\mathbf{x}_{t_1} \mid \mathbf{x}) = \mathcal{N}\left( \mathbf{x}_{t_1}; \sqrt{\overline{\alpha}_{t_1}} \mathbf{x}, (1 - \overline{\alpha}_{t_1}) \mathbf{I} \right), \tag{49}$$

we obtain

$$\begin{aligned}
\nabla_{\mathbf{x}} \log p_X(\mathbf{x}) &= -\frac{1}{r_{t_1}^2} \mathbb{E}_{p_{X_{t_1}|X}(\mathbf{x}_{t_1}|\mathbf{x})} \left( \mathbf{x} - \frac{\mathbf{x}_{t_1} + (1 - \overline{\alpha}_{t_1}) \nabla \log p_{t_1}(\mathbf{x}_{t_1})}{\sqrt{\overline{\alpha}_{t_1}}} \right) \\
&= \frac{1 - \overline{\alpha}_{t_1}}{r_{t_1}^2 \sqrt{\overline{\alpha}_{t_1}}} \mathbb{E}_{p_{X_{t_1}|X}(\mathbf{x}_{t_1}|\mathbf{x})} [\nabla \log p_{X_{t_1}}(\mathbf{x}_{t_1})].
\end{aligned} \tag{50}$$

This completes the proof. $\square$

## B.4. Proof of Theorem 3.5

*Proof.* We first consider the MAP point-mass distribution

$$p_{X_{\mathrm{MAP}}|Y}(\mathbf{x} \mid \mathbf{y}) = \delta(\mathbf{x} - \mathbf{x}_{\mathrm{MAP}}). \tag{51}$$

By the definition of $W_2$ distance and Theorem 3.2, we have

$$W_2^2(p_{X_{\mathrm{MAP}}|Y}, p_{X|Y}) = \mathbb{E}_{\mathbf{x} \sim p_{X|Y}} \|\mathbf{x} - \mathbf{x}_{\mathrm{MAP}}\|^2 = \|\mathbf{x}_{\mathrm{MAP}} - \mathbf{x}_{\mathrm{MMSE}}\|^2 + \mathbb{E}_{\mathbf{x} \sim p_{X|Y}} \|\mathbf{x} - \mathbf{x}_{\mathrm{MMSE}}\|^2 \le \frac{2n_x}{\mu}, \tag{52}$$

where the last inequality follows from (42) and (44).

Note that the forward diffusion operator $\mathcal{T}(0, t_0)_{\#}$ acts as a linear scaling by $\sqrt{\overline{\alpha}_{t_0}}$, followed by a convolution with an independent Gaussian random variable. By Lemma B.5, we obtain

$$W_2\left(\mathcal{T}(0, t_0)_{\#} p_{X_{\mathrm{MAP}}|Y}, \mathcal{T}(0, t_0)_{\#} p_{X|Y}\right) \le \sqrt{\frac{2\overline{\alpha}_{t_0} n_x}{\mu}}. \tag{53}$$

Applying Lemma B.4 yields

$$W_2\left(p_{X|Y}(\mathbf{x} \mid \mathbf{y}), p_{0 \to t_0 \to 0}(\mathbf{x} \mid \mathbf{y})\right) \le (\overline{\alpha}_{t_0})^{1-L_s} \sqrt{2n_x/\mu} + \int_0^{t_0} g^2(r) (\overline{\alpha}_r)^{\frac{1}{2}-L_s} \Delta_r \mathrm{d}r \le (\overline{\alpha}_{t_0})^{1-L_s} \sqrt{2n_x/\mu} + \epsilon_{\mathrm{score}}, \tag{54}$$

for any $\mathbf{y}$. Finally, this bound implies the existence of a transport plan between $p_X$ and $p_{0 \to t_0 \to 0}(\mathbf{x})$ induced by the optimal conditional transport between $p_{X|Y}$ and $p_{0 \to t_0 \to 0}(\mathbf{x} \mid \mathbf{y})$, whose cost is bounded by

$$(\overline{\alpha}_{t_0})^{1-L_s} \sqrt{2n_x/\mu} + \epsilon_{\mathrm{score}}. \tag{55}$$

This concludes Theorem 3.5. $\qquad\square$

# C. Theoretical analysis for LMAP-RPS

## C.1. Formal assumptions and error bounds

Consider a latent-space formulation of the inverse problem. Let $Z \in \mathbb{R}^d$ denote the latent random variable associated with the data $X$ where $d \ll n_x$ is the latent dimensionality, and let $\mathbf{z}$ denote a realization of $Z$. We assume that $Z$ and $X$ are related through an encoder–decoder pair, where the encoder $\mathcal{E} : \mathbb{R}^{n_x} \to \mathbb{R}^d$ maps data to latent representations and the decoder $\mathcal{D} : \mathbb{R}^d \to \mathbb{R}^{n_x}$ maps latent variables back to the data space. The latent-space MMSE estimator is defined as $Z_{\mathrm{MMSE}} = \mathbb{E}[Z \mid Y]$, and the latent-space MAP estimator is given by $Z_{\mathrm{MAP}} = \arg\max_Z p_{Z|Y}$. We impose the following assumptions throughout the theoretical analysis of LMAP-RPS.

**Assumption C.1.** We assume the following conditions:

(C.1.1) For any realization $\mathbf{y}$ of $Y$, the posterior distribution $p_{Z|Y}(\cdot \mid \mathbf{y})$ admits a $\mu$-strongly log-concave density. Moreover, we assume the strong log-concavity parameter satisfies

$$\mu \ge 2\pi (\sup p_{Z|Y})^{-d/2}. \tag{56}$$

(C.1.2) The decoder $\mathcal{D} : \mathbb{R}^d \to \mathbb{R}^{n_x}$ is $L_{\mathcal{D}}$-Lipschitz continuous, i.e., for any $\mathbf{z}_1, \mathbf{z}_2 \in \mathrm{supp}(Z)$,

$$\|\mathcal{D}(\mathbf{z}_1) - \mathcal{D}(\mathbf{z}_2)\|_2 \le L_{\mathcal{D}} \|\mathbf{z}_1 - \mathbf{z}_2\|_2. \tag{57}$$

(C.1.3) The decoder $\mathcal{D}$ exactly pushes forward the latent posterior to the data posterior, i.e.,

$$\mathcal{D}_{\#} p_{Z|Y} = p_{X|Y}, \tag{58}$$

almost everywhere for any realization of $Y$.

(C.1.4) There exists a global constant $L_s$ such that the approximated posterior score in latent space, $\mathbf{s}_\theta(\mathbf{z}_t, \mathbf{y}, t)$, is one-sided $L_s$-Lipschitz with respect to its first argument, namely,

$$\|\mathbf{s}_\theta(\mathbf{z}_1, \mathbf{y}, t) - \mathbf{s}_\theta(\mathbf{z}_2, \mathbf{y}, t)\|_2 \leq L_s \|\mathbf{z}_1 - \mathbf{z}_2\|_2, \tag{59}$$

for any $\mathbf{z}_1, \mathbf{z}_2 \in \mathrm{supp}(Z_t)$.

(C.1.5) There exists a universal constant $\epsilon_{\mathrm{score}}$ such that the weighted latent-space score estimation error is bounded by

$$\int_0^T g^2(r) \, (\overline{\alpha}_r)^{\frac{1}{2} - L_s} \, [\mathbb{E}_{p_{Z_r|Y}(\mathbf{z}_r|\mathbf{y})} \|\nabla_{\mathbf{z}_r} \log p_{Z_r|Y}(\mathbf{z}_r \mid \mathbf{y}) - \mathbf{s}_\theta(\mathbf{z}_r, \mathbf{y}, r)\|^2]^{1/2} \mathrm{d}r \leq \epsilon_{\mathrm{score}}. \tag{60}$$

(C.1.6) All regularity conditions in Kwon et al. (2022) are assumed to hold on the posterior distribution $p_{Z|Y}$ and its associated SDE.

Then the following theorem provides an error bound for the MAP stage in LMAP-RPS.

**Theorem C.2.** *Under Assumption C.1, we have*

$$\mathbb{E}\|\mathcal{D}(Z_{MAP}) - X_{MMSE}\| \leq 2L_\mathcal{D}\sqrt{d/\mu}, \tag{61}$$

*and*

$$\mathbb{E}\|X - \mathcal{D}(Z_{MAP})\|^2 \leq D^* + 4L_\mathcal{D}^2 \frac{d}{\mu}. \tag{62}$$

Analogous to Theorem 3.5, let $p_{0 \to t_0 \to 0}(\mathbf{z} \mid \mathbf{y})$ denote the estimator obtained from re-noised posterior sampling in the latent space, and define

$$\tilde{p}_{0 \to t_0 \to 0}(\mathbf{x}) = \mathbb{E}_{\mathbf{y} \sim p_Y}[\mathcal{D}_\# p_{0 \to t_0 \to 0}(\mathbf{z} \mid \mathbf{y})]. \tag{63}$$

Theorem C.3 provides the upper bound on $W_2$ distance for LMAP-RPS.

**Theorem C.3.** *Under Assumption C.1, we have*

$$W_2(p_X, \tilde{p}_{0 \to t_0 \to 0}(\mathbf{x})) \leq L_\mathcal{D}(\overline{\alpha}_{t_0})^{1 - L_s} \sqrt{2d/\mu} + L_\mathcal{D} \epsilon_{score}.$$

## C.2. Proof of Theorem C.2

*Proof.* Following an argument analogous to the proof of Theorem 3.2 in Section B.2, we obtain

$$\|Z_{MAP} - Z_{MMSE}\|^2 \leq \frac{d}{\mu} \sup\left(p_{Z|Y}\right) \left(\frac{2\pi}{\mu}\right)^{d/2} \leq \frac{d}{\mu}. \tag{64}$$

as well as

$$\mathbb{E}\|Z - Z_{MMSE}\|^2 \leq \frac{d}{\mu}. \tag{65}$$

By Assumption (C.1.2), we have

$$\mathbb{E}\|\mathcal{D}(Z_{MAP}) - \mathcal{D}(Z_{MMSE})\|^2 \leq L_\mathcal{D}^2 \frac{d}{\mu}. \tag{66}$$

Moreover, by Assumption (C.1.3), it follows that

$$\|X_{MMSE} - \mathcal{D}(Z_{MMSE})\|^2 = \|\mathbb{E}[\mathcal{D}(Z) \mid Y] - \mathcal{D}(Z_{MMSE})\|^2 \leq \mathbb{E}[\|\mathcal{D}(Z) - \mathcal{D}(Z_{MMSE})\|^2 \mid Y]$$
$$\leq L_\mathcal{D}^2 \mathbb{E}\left[\|Z - Z_{MMSE}\|^2 \mid Y\right] \leq L_\mathcal{D}^2 \frac{d}{\mu}. \tag{67}$$

Thus

$$\mathbb{E}\|\mathcal{D}(Z_{MAP}) - X_{MMSE}\| \leq \mathbb{E}\|\mathcal{D}(Z_{MAP}) - \mathcal{D}(Z_{MMSE})\| + \mathbb{E}\|\mathcal{D}(Z_{MMSE}) - X_{MMSE}\| \leq 2L_\mathcal{D}\sqrt{\frac{d}{\mu}}, \tag{68}$$

which completes the proof. $\square$

*Table 3.* Hyperparameters for MAP-RPS on FFHQ.

| Parameter | Description | Denoising $(\sigma_y = 0.3)$ | SR $4\times$ $(\sigma_y = 0.1)$ | Inpainting $50\%$ $(\sigma_y = 0.1)$ | Deblur Aniso $(\sigma_y = 0)$ |
|---|---|---|---|---|---|
| $\eta_0$ | Initial step size for MAP optimization | 0.5 | 0.5 | 0.5 | 0.5 |
| $\eta_{\min}$ | Minimum learning rate for cosine annealing | $10^{-5}$ | $10^{-5}$ | $10^{-5}$ | $10^{-5}$ |
| $N$ | Number of MAP optimization iterations | 60 | 300 | 400 | 200 |
| $w$ | Weight of the prior gradient term | 2.0 | 0.25 | 0.7 | 0.02 |
| $g_{\text{likelihood}}$ | Likelihood formulation | $\|\mathbf{y} - \mathcal{A}(\mathbf{x})\|_2^2$ | $\|\mathbf{y} - \mathcal{A}(\mathbf{x})\|_2^2$ | $\|\mathbf{y} - \mathcal{A}(\mathbf{x})\|_2^2$ | $\|\mathbf{y} - \mathcal{A}(\mathbf{x})\|_2^2$ |
| $t_1$ | Time step for prior gradient estimation | 10 | 10 | 10 | 10 |
| $t_0$ | Re-noising timesteps used to reproduce Figure 1 | $\{0, 25, 50, 75, 100\}$ | $\{0, 100, 200, 300, 400\}$ | $\{0, 25, 50, 75, 100\}$ | $\{0, 25, 50, 75, 100\}$ |
| PS method | Posterior sampling method in RPS stage | DPS ($\xi = 1.0$) | DPS ($\xi = 1.0$) | DPS ($\xi = 1.0$) | DPS ($\xi = 2.0$) |

*Table 4.* Hyperparameters for LMAP-RPS on FFHQ.

| Parameter | Description | Denoising $(\sigma_y = 0.3)$ | SR $4\times$ $(\sigma_y = 0.1)$ | Inpainting $50\%$ $(\sigma_y = 0.1)$ | Deblur Aniso $(\sigma_y = 0)$ |
|---|---|---|---|---|---|
| $\eta_0$ | Initial step size for MAP optimization | 0.5 | 0.5 | 0.5 | 0.5 |
| $\eta_{\min}$ | Minimum learning rate for cosine annealing | $10^{-5}$ | $10^{-5}$ | $10^{-5}$ | $10^{-5}$ |
| $N$ | Number of MAP optimization iterations | 70 | 60 | 120 | 100 |
| $w$ | Weight of the prior gradient term | 0.6 | 0.25 | 0.3 | 0.001 |
| $g_{\text{likelihood}}$ | Likelihood formulation | $\|\mathbf{y} - \mathcal{A}(\mathcal{D}(\mathbf{z}))\|_2^2$ | $\|\mathbf{y} - \mathcal{A}(\mathcal{D}(\mathbf{z}))\|_2^2$ | $\|\mathbf{y} - \mathcal{A}(\mathcal{D}(\mathbf{z}))\|_2^2$ | $\|\mathbf{y} - \mathcal{A}(\mathcal{D}(\mathbf{z}))\|_2^2$ |
| $t_1$ | Time step for prior gradient estimation | 50 | 50 | 50 | 50 |
| $t_0$ | Re-noising timesteps used to reproduce Figure 1 | $\{0, 200, 400, 600, 800\}$ | $\{0, 200, 400, 600, 800\}$ | $\{0, 200, 400, 600, 800\}$ | $\{0, 25, 50, 75, 100\}$ |
| PS method | Posterior sampling method in RPS stage | Latent-DPS ($\xi = 0.015$) | Latent-DPS ($\xi = 0.08$) | Latent-DPS ($\xi = 0.05$) | PSLD ($\xi = 0.04$) |

## C.3. Proof of Theorem C.3

*Proof.* The proof follows the same strategy as Section B.4. In particular, we obtain

$$W_2(p_{Z|Y}, p_{0 \to t_0 \to 0}(\mathbf{z} \mid \mathbf{y})) \leq (\overline{\alpha}_{t_0})^{1 - L_s} \sqrt{2d/\mu} + \int_0^{t_0} g^2(r) (\overline{\alpha}_r)^{\frac{1}{2} - L_s} \Delta_r \mathrm{d}r, \tag{69}$$

where

$$\Delta_r = [\mathbb{E}_{p_{Z_r|Y}} \|\nabla_{\mathbf{z}_r} \log p_{Z_r|Y}(\mathbf{z}_r \mid \mathbf{y}) - \mathbf{s}_\theta(\mathbf{z}_r, \mathbf{y}, r)\|^2]^{1/2}. \tag{70}$$

By Assumption (C.1.2), we have

$$W_2(\mathcal{D}_{\#} p_{Z|Y}, \mathcal{D}_{\#} p_{0 \to t_0 \to 0}(\mathbf{z} \mid \mathbf{y})) \leq L_{\mathcal{D}} W_2(p_{Z|Y}, p_{0 \to t_0 \to 0}(\mathbf{z} \mid \mathbf{y})), \tag{71}$$

which holds as for any coupling $\pi \in \Pi(p_{Z|Y}, p_{0 \to t_0 \to 0}(\mathbf{z} \mid \mathbf{y}))$, we may construct the pushforward coupling

$$\tilde{\pi} = \pi \circ \begin{bmatrix} \mathcal{D} & 0 \\ 0 & \mathcal{D} \end{bmatrix}^{-1} \in \Pi(\mathcal{D}_{\#} p_{Z|Y}, \mathcal{D}_{\#} p_{0 \to t_0 \to 0}(\mathbf{z} \mid \mathbf{y})), \tag{72}$$

which satisfies

$$\int \|X_1 - X_2\|^2 \mathrm{d}\tilde{\pi}(X_1, X_2) = \int \|\mathcal{D}(Z_1) - \mathcal{D}(Z_2)\|^2 \mathrm{d}\pi(Z_1, Z_2) \leq L_{\mathcal{D}}^2 \int \|Z_1 - Z_2\|^2 \mathrm{d}\pi(Z_1, Z_2). \tag{73}$$

Considering Assumptions (C.1.3) and (C.1.5), we obtain

$$W_2(p_{X|Y}, \mathcal{D}_{\#} p_{0 \to t_0 \to 0}(\mathbf{z} \mid \mathbf{y})) = W_2(\mathcal{D}_{\#} p_{Z|Y}, \mathcal{D}_{\#} p_{0 \to t_0 \to 0}(\mathbf{z} \mid \mathbf{y})) \leq L_{\mathcal{D}} (\overline{\alpha}_{t_0})^{1 - L_s} \sqrt{2d/\mu} + L_{\mathcal{D}} \epsilon_{\text{score}}. \tag{74}$$

Since the bound holds for any realization $\mathbf{y}$, we may further construct a transport plan between the marginals $p_X$ and $\mathbb{E}_{\mathbf{y} \sim p_Y}[\mathcal{D}_{\#} p_{0 \to t_0 \to 0}(\mathbf{z} \mid \mathbf{y})]$ by averaging the optimal conditional couplings between $p_{X|Y}$ and $\mathcal{D}_{\#} p_{0 \to t_0 \to 0}(\mathbf{z} \mid \mathbf{y})$. Thus, we conclude. $\square$

*Table 5.* Hyperparameters of LMAP-RPS on MS-COCO for super-resolution, inpainting, and anisotropic deblurring.

| Parameter | Description | SR $4\times$ | Inpainting $50\%$ | Deblur Aniso |
|---|---|---|---|---|
| $\eta_0$ | Initial step size in MAP stage | 2.0 | 2.0 | 2.0 |
| $\eta_{\min}$ | Minimum learning rate in cosine annealing | $10^{-2}$ | $10^{-2}$ | $10^{-2}$ |
| $N$ | Number of MAP optimization iterations | 100 | 200 | 200 |
| $w$ | Weight of the prior term | 0.15 | 0.3 | 0.1 |
| $g_{\text{likelihood}}$ | Likelihood formulation | $\|\mathbf{y} - \mathcal{A}(\mathcal{D}(\mathbf{z}))\|_2^2$ | $\|\mathbf{y} - \mathcal{A}(\mathcal{D}(\mathbf{z}))\|_2^2$ | $\|\mathbf{y} - \mathcal{A}(\mathcal{D}(\mathbf{z}))\|_2^2$ |
| $t_1$ | Time step for prior gradient estimation | 50 | 50 | 50 |
| PS method | Posterior sampling method in RPS stage | Latent-DPS ($\xi = 0.1$) | Latent-DPS ($\xi = 0.01$) | Latent-DCDP (lr $= 0.1$, iters$= 100$) |

*Table 6.* Hyperparameters of LMAP-RPS on MS-COCO for compressed sensing, HDR reconstruction, and nonlinear deblurring.

| Parameter | Description | CS $2\times$ | HDR | Nonlinear Deblur |
|---|---|---|---|---|
| $\eta_0$ | Initial step size in MAP stage | 0.5 | 2.0 | 2.0 |
| $\eta_{\min}$ | Minimum learning rate in cosine annealing | $10^{-2}$ | $10^{-2}$ | $10^{-5}$ |
| $N$ | Number of MAP optimization iterations | 300 | 400 | 500 |
| $w$ | Weight of the prior term | 0.005 | 0.005 | 0.002 |
| $g_{\text{likelihood}}$ | Likelihood formulation | $\|\mathbf{y} - \mathcal{A}(\mathcal{D}(\mathbf{z}))\|_2$ | $\|\mathbf{y} - \mathcal{A}(\mathcal{D}(\mathbf{z}))\|_2$ | $\|\mathbf{y} - \mathcal{A}(\mathcal{D}(\mathbf{z}))\|_2$ |
| $t_1$ | Time step for prior gradient estimation | 50 | 50 | 50 |
| PS method | Posterior sampling method in RPS stage | Latent-DPS ($\xi = 1.5$) | Latent-DPS ($\xi = 0.5$) | Latent-DPS ($\xi = 1.5$) |

## D. Experimental and implementation details

### D.1. Inverse problem implementations

All inverse problem implementations follow DDRM (Kawar et al., 2022) and DPS (Chung et al., 2023). Specifically, for inpainting, we apply a random mask with a missing ratio of $50\%$ independently sampled for each pixel. For $4\times$ super-resolution, we use $4 \times 4$ averaging pooling as the downsampling operator. For anisotropic deblurring, we employ Gaussian blur kernels whose standard deviations along the two spatial directions are set to 20 and 1, respectively. For $2\times$ compressed sensing, we adopt the Walsh-Hadamard transform (Wang & Zhao, 2016) with a sampling rate of $50\%$. For high dynamic range reconstruction, we use the following pixel-wise observation function:

$$f(\mathbf{x}_i) = \begin{cases} -1, & 2\mathbf{x}_i \leq -1, \\ 2\mathbf{x}_i, & -1 < 2\mathbf{x}_i < 1, \\ 1, & 2\mathbf{x}_i \geq 1. \end{cases} \tag{75}$$

For nonlinear deblurring, we employ the neural network-based blurring kernel from Tran et al. (2021).

For all noisy inverse problems, the noise standard deviation used in diffusion-based methods is multiplied by a factor of 2 to match the input scaling of diffusion models. For example, in the denoising experiments of the D-P tradeoff with a noise standard deviation of 0.3, the actual noise added to data in $[-1, 1]$ range has a standard deviation of 0.6. Similarly, for all inverse problems on the MS-COCO dataset, the effective noise standard deviation added to the observations is 0.1.

### D.2. Implementation details for MAP-RPS and LMAP-RPS

For MAP estimation in Stage 1, we optimize the objective using the AdamW optimizer (Loshchilov & Hutter, 2019) for $N$ iterations with an initial learning rate $\eta_0$. A cosine annealing learning rate schedule is employed with a minimum learning rate $\eta_{\min}$. For linear measurements $\mathbf{A}$, we initialize MAP-RPS and LMAP-RPS using the pseudoinverse solution $\mathbf{A}^\dagger \mathbf{y}$ and its encoded version $\mathcal{E}(\mathbf{A}^\dagger \mathbf{y})$, respectively. For HDR reconstruction, the initialization is set to $\mathbf{y}/2$ and $\mathcal{E}(\mathbf{y}/2)$, and for nonlinear deblurring, we directly use $\mathbf{y}$ and $\mathcal{E}(\mathbf{y})$ as initializations.

To compute the gradient of the log-prior term, we follow Theorem 3.3 and estimate its stochastic gradient as

$$g_{\mathbf{x},\text{prior}} = \frac{1 - \overline{\alpha}_{t_1}}{r_{t_1}^2 \sqrt{\overline{\alpha}_{t_1}}} \nabla \log p_{X_{t_1}}(\mathbf{x}_{t_1}), \quad \mathbf{x}_{t_1} \sim \mathcal{N}\left(\sqrt{\overline{\alpha}_{t_1}}\mathbf{x}, (1 - \overline{\alpha}_{t_1})\mathbf{I}\right). \tag{76}$$

For diffusion models parameterized via the standard $\epsilon$-prediction formulation, the score term is further approximated as

$$\nabla \log p_{X_{t_1}}(\mathbf{x}_{t_1}) \approx -\frac{1}{\sqrt{1 - \bar{\alpha}_{t_1}}} \epsilon_\theta(\mathbf{x}_{t_1}, t_1). \tag{77}$$

We note that the prior gradient in (76) involves a parameter $r_{t_1}$, which can be interpreted as a weighting factor balancing the prior term and the likelihood term. For clarity and simplicity, we therefore introduce a single scalar weight $w$ to absorb all constant coefficients in the prior gradient and the likelihood term. Since the observation noise is assumed to be Gaussian, the gradient of the log-likelihood is

$$\nabla_{\mathbf{x}} \log p_{Y|X}(\mathbf{y} \mid \mathbf{x}) \propto -\nabla_{\mathbf{x}} \|\mathbf{y} - \mathcal{A}(\mathbf{x})\|_2^2. \tag{78}$$

Combining (76) and (78), the stochastic gradient used for MAP optimization is

$$\mathbf{g}_{\mathbf{x}} \approx -\nabla_{\mathbf{x}} \|\mathbf{y} - \mathcal{A}(\mathbf{x})\|_2^2 - w\epsilon_\theta\left(\sqrt{\bar{\alpha}_{t_1}}\mathbf{x} + \sqrt{1 - \bar{\alpha}_{t_1}}\epsilon, t_1\right), \quad \epsilon \sim \mathcal{N}(\mathbf{0}, \mathbf{I}). \tag{79}$$

Similarly, for LMAP-RPS, the stochastic gradient in the latent space is given by

$$\mathbf{g}_{\mathbf{z}} \approx -\nabla_{\mathbf{z}} \|\mathbf{y} - \mathcal{A}(\mathcal{D}(\mathbf{z}))\|_2^2 - w\epsilon_\theta\left(\sqrt{\bar{\alpha}_{t_1}}\mathbf{z} + \sqrt{1 - \bar{\alpha}_{t_1}}\epsilon, t_1\right), \quad \epsilon \sim \mathcal{N}(\mathbf{0}, \mathbf{I}). \tag{80}$$

Note that the likelihood term used in (80) corresponds to a single-point approximation of the marginal likelihood in the latent space, i.e.,

$$\nabla_{\mathbf{z}} \log p(\mathbf{y} \mid \mathbf{z}) \approx \left. \nabla_{\mathbf{z}} \log p(\mathbf{y} \mid \mathbf{x}) \right|_{\mathbf{x}=\mathcal{D}(\mathbf{z})}, \tag{81}$$

which implicitly assumes a single point estimation of $p_{X|Z}$. This approximation introduces a slight bias compared to the exact marginalization over $\mathbf{x}$, but enables a highly efficient and scalable implementation in practice.

We observe that, for certain inverse problems, directly using the MSE (78) as the likelihood term can lead to severe numerical instability during optimization. We note that in the implementations of several prior works (Chung et al., 2023; Mardani et al., 2024; Zhang et al., 2025a), the $\ell_2$ norm $\|\mathbf{y} - \mathcal{A}(\mathbf{x})\|_2$ is used in place of the squared error to improve stability. Following this practice, we adopt the $\ell_2$ loss as the likelihood term for compressed sensing, HDR reconstruction, and nonlinear deblurring tasks on MS-COCO.

For Stage 2, we use DPS (Chung et al., 2023) and Latent-DPS (Song et al., 2024) by default, both of which involve a step-size hyperparameter denoted by $\xi$. For the anisotropic deblurring task in the DP-tradeoff experiments, we employ PSLD (Rout et al., 2023), whose step size is also denoted by $\xi$ for consistency. For the anisotropic deblurring task on MS-COCO, we use Latent-DCDP (Li et al., 2024b), where the optimizer is AdamW with a learning rate of 0.1 and the number of optimization iterations is set to 100.

### D.3. Detailed hyperparameters settings

All hyperparameter tuning is performed on the first two test samples, following common practice in prior work. For all baseline methods, we fix the sampling timesteps according to the recommended values in the original papers, and then conduct a task-by-task grid search for the remaining hyperparameters, including the step size, the number of optimization steps, and other method-specific parameters. For LMAP-RPS, we only tune the hyperparameters in the LMAP stage, while the RPS stage directly adopts the optimal hyperparameters of the corresponding posterior sampling method. This tuning protocol largely ensures that all baseline methods are evaluated under their recommended sampling settings and achieve strong performance on each task.

The complete hyperparameter configurations for MAP-RPS and LMAP-RPS in the DP-tradeoff experiments on FFHQ are reported in Table 3 and Table 4, respectively. The detailed hyperparameter settings for the inverse problem experiments on MS-COCO are provided in Table 5 and Table 6. All FFHQ experiments are conducted on a single NVIDIA RTX 3090 GPU, while all MS-COCO experiments are conducted on an NVIDIA RTX A800 GPU.

## E. Additional results and comparisons

### E.1. Results on more challenging tasks

Note that the MAP approximation to the MMSE estimator used in Stage 1 is theoretically accurate primarily when the posterior distribution is approximately unimodal. In the presence of multimodal posteriors, the validity of the MAP stage is

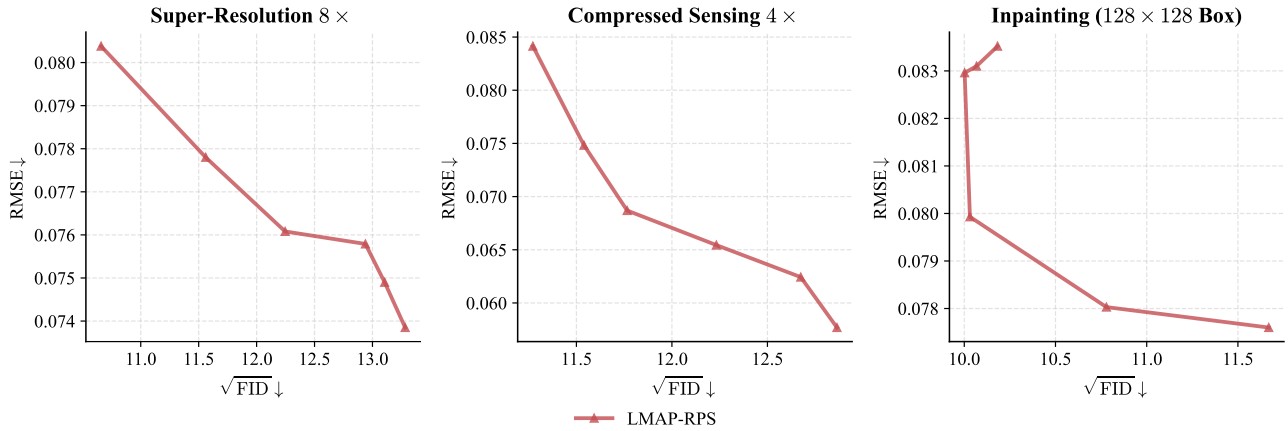

*Figure 3.* The performance of LMAP-RPS on $8\times$ super-resolution, $4\times$ compressed sensing and $128 \times 128$ box inpainting on FFHQ ($\sigma_{\mathbf{y}} = 0.1$).

no longer covered by Theorem 3.2 or Theorem C.2. To further evaluate the proposed method under challenging settings, we consider inverse problems with substantially more severe degradations, which are expected to induce more complex posterior structures.

The considered tasks include $8\times$ super-resolution, $4\times$ compressed sensing, and $128 \times 128$ box inpainting on FFHQ with $\sigma_{\mathbf{y}} = 0.1$. The corresponding D-P curves obtained by the proposed LMAP-RPS are shown in Figure 3. Representative visualizations are provided in Figure 14. Across all these tasks, LMAP-RPS achieves satisfactory distortion-perception tradeoffs.

Moreover, we further evaluate on $128 \times 128$ box inpainting on ImageNet (Deng et al., 2009), where the posterior distribution is expected to deviate more significantly from the unimodal assumption due to the increased complexity and diversity of the dataset. The resulting D-P curve of LMAP-RPS is shown in Figure 4. Notably, LMAP-RPS still achieves a favorable distortion-perception tradeoff even when the posterior may exhibit highly multimodal structures. Visualizations are provided in Figure 5. We can observe that at $t_0 = 0$, i.e., the optimal distortion point, the reconstructions appear more averaged and may even collapse to background content. In contrast, at $t_0 = 800$, the reconstructions are closer aligned with posterior sampling and become substantially more realistic and detailed, albeit with larger deviations from the ground truth.

These experiments demonstrate that the proposed method can generalize to more challenging tasks beyond the strong log-concavity regime, and can remain effective even under potentially multimodal posteriors. Qualitatively, although the approximation error may become larger beyond the log-concave setting and is no longer theoretically bounded as in Theorem 3.2, using MAP estimation as an approximation to MMSE may still be practically effective for a broader class of posteriors due to its computational simplicity and compatibility with diffusion models. We acknowledge that the theoretical mechanism underlying its effectiveness in more complex tasks, especially under highly multimodal posteriors, remains underexplored. We leave further theoretical analysis and the design of improved approximations of the MMSE solution for complex multimodal posteriors to future work.

### E.2. Comparison with more pixel-space inverse algorithms

Here, we further compare MAP-RPS with existing pixel-space inverse algorithms, including DDNM (Wang et al., 2023), DPS (Chung et al., 2023), RED-diff (Mardani et al., 2024), DMPS (Meng & Kabashima, 2025), DiffPIR (Zhu et al., 2023), DAPS (Zhang et al., 2025a), and SITCOM (Alkhouri et al., 2025). We evaluate all methods on $50\%$ random inpainting, anisotropic deblurring, and $4\times$ super-resolution using the FFHQ dataset. For all tasks, Gaussian observation noise with $\sigma_{\mathbf{y}} = 0.05$ (with respect to the data range $[0, 1]$) is applied. For MAP-RPS, we adopt DDNM for the renoised posterior sampling stage and report performance for both $t_0 = 0$ and $t_0 = 300$. The results are presented in Table 7. MAP-RPS consistently outperforms competing methods on both distortion and perceptual metrics.

Similar to LMAP-RPS, we also observe cases in inpainting where the distortion and perceptual metrics of MAP-RPS improve or deteriorate simultaneously, as discussed in Section 4.2. We note that the underlying mechanism is similar: under

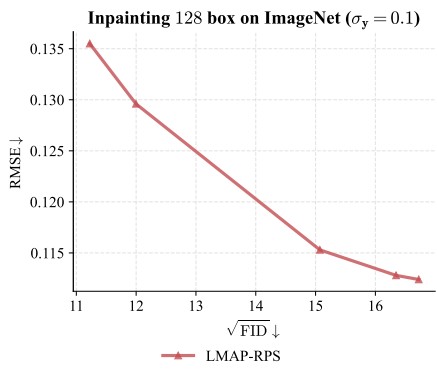

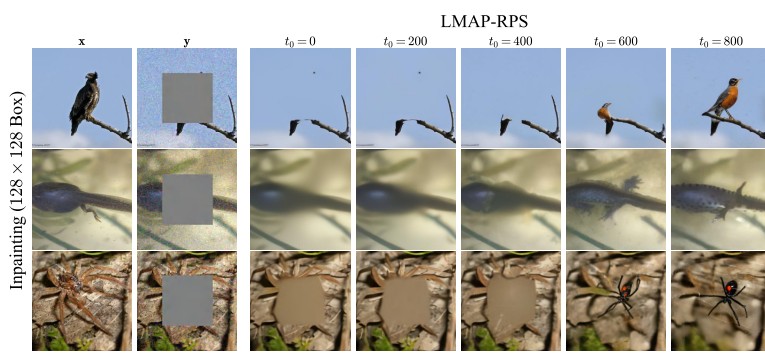

*Figure 4.* D-P curve of LMAP-RPS on $128 \times 128$ box inpainting on ImageNet with $\sigma_{\mathbf{y}} = 0.1$.

*Figure 5.* Visualizations of LMAP-RPS on $128 \times 128$ box inpainting on ImageNet with varying $t_0$s.

*Table 7.* Quantitative comparison for pixel-space inverse algorithms on FFHQ. MAP-RPS (0) and MAP-RPS (300) denote our MAP-RPS algorithm with $t_0 = 0$ and $t_0 = 300$, respectively. **Bold** indicates the best result, and underline indicates the second-best.

| Method | Inpainting | | | | SR $4\times$ | | | | Deblur Aniso | | | |
|---|---|---|---|---|---|---|---|---|---|---|---|---|
| | PSNR↑ | SSIM↑ | LPIPS↓ | FID↓ | PSNR↑ | SSIM↑ | LPIPS↓ | FID↓ | PSNR↑ | SSIM↑ | LPIPS↓ | FID↓ |
| DDNM | 33.56 | 0.9173 | 0.1511 | 58.07 | 29.76 | 0.8430 | 0.2191 | 80.47 | 28.80 | 0.8235 | 0.2431 | 88.53 |
| DPS | 33.10 | 0.9065 | 0.1380 | 39.77 | 27.11 | 0.7626 | 0.2337 | 69.80 | 24.49 | 0.6871 | 0.3065 | 90.64 |
| RED-diff | 28.50 | 0.7648 | 0.3161 | 103.10 | 27.72 | 0.7762 | 0.3261 | 98.31 | 26.90 | 0.6675 | 0.4427 | 123.97 |
| DMPS | 27.93 | 0.8338 | 0.2293 | 89.91 | 28.09 | 0.7989 | 0.2653 | 97.05 | 26.01 | 0.7452 | 0.3082 | 111.64 |
| DiffPIR | 30.73 | 0.8586 | 0.2363 | 97.45 | 27.56 | 0.7772 | 0.3041 | 114.19 | 25.47 | 0.7188 | 0.3473 | 120.30 |
| DAPS | 30.46 | 0.7788 | 0.2394 | 61.54 | 28.76 | 0.7833 | 0.2767 | 77.38 | 27.70 | 0.7649 | 0.2942 | 86.68 |
| SITCOM | 30.04 | 0.7677 | 0.2541 | 78.84 | 27.14 | 0.6590 | 0.3578 | 108.59 | 24.70 | 0.5433 | 0.5053 | 155.80 |
| **MAP-RPS (0)** | 34.84 | 0.9025 | 0.1626 | 47.68 | **31.56** | 0.8568 | 0.2387 | 68.57 | **29.06** | 0.8217 | 0.2888 | 93.86 |
| **MAP-RPS (300)** | **35.84** | **0.9429** | **0.1103** | **36.48** | 30.70 | **0.8652** | **0.1778** | **56.87** | 28.84 | **0.8259** | **0.2319** | **75.37** |

random inpainting with low additive noise, the posterior distribution becomes more concentrated, making distortion and perceptual metrics more aligned and primarily determined by the accuracy of the inverse algorithm. Unlike the phenomenon in Table 1, both the distortion and perceptual metrics of MAP-RPS at $t_0 = 300$ are better than those at $t_0 = 0$. This is because we apply DDNM in the RPS stage, which introduces smaller estimation errors at small $t$s, benefiting from the absence of additional decoder errors in pixel-space diffusion models.

### E.3. Compatibility with different posterior samplers.

In our experiments, we primarily use DPS and Latent-DPS as the posterior sampling method in Stage 2. It is worth noting that the theoretical analysis of Stage 2 does not rely on a specific posterior sampling algorithm, and thus our framework can be combined with any reasonable posterior sampler. To further examine this compatibility, we evaluate MAP-RPS on denoising with $\sigma_{\mathbf{y}} = 0.3$, using DPS, ΠGDM, and DMPS as the posterior sampler, respectively. The results are shown in Figure 6a. The D-P curves remain consistently reasonable across different samplers, while the final accuracy depends on the quality of the underlying sampler.

One may note that recent studies suggest that the update rule of DPS and its variants may be closer to a so-called "local MAP" update rather than exact posterior sampling (Xu et al., 2025), which can be formulated as

$$X_{t-1} \sim \delta \left( X - \arg\max p(X_{t-1} \mid X_t, \mathbf{y}) \right). \tag{82}$$

We emphasize that such a local MAP update is generally not equivalent to the global MAP solution (Zhang et al., 2025c). Moreover, despite this interpretation, DPS-type methods have been empirically shown to achieve competitive or even superior FID compared with many mainstream inverse algorithms. This makes them suitable choices for Stage 2 of MAP-RPS, where the goal is to gradually steer the solution toward a region with lower perceptual error.

We also acknowledge that more accurate posterior samplers may further improve the D-P traversal of MAP-RPS, especially

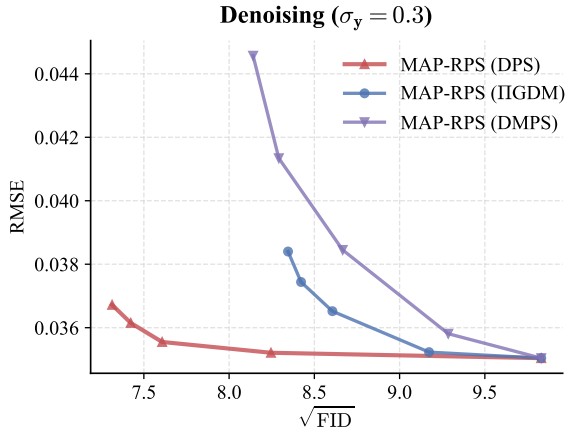

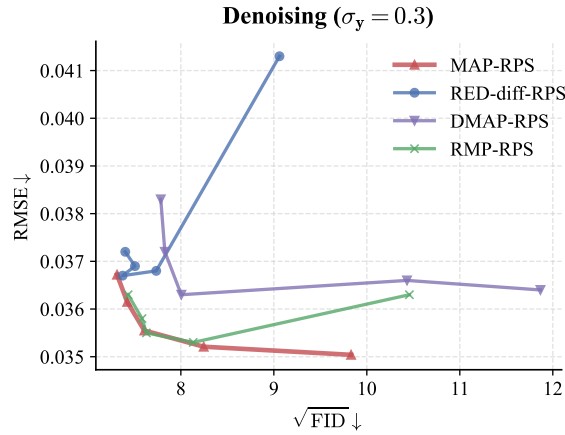

*(a)* Capability with different posterior samplers.

*(b)* Comparison with other initialization methods.

*Figure 6.* Comparison with different posterior samplers and initialization methods.

asymptotically consistent posterior sampling methods (Xu & Chi, 2024). Since our Stage 2 is not restricted to a particular sampler, integrating these methods into MAP-RPS and developing a more refined theoretical understanding are promising directions for future work.

### E.4. Alternative Stage 1 estimators.

The MAP solver proposed in Stage 1 provides an efficient way to obtain an accurate MAP estimate, as justified by Theorem 3.3. In this section, we further examine whether other inverse algorithms, especially those that can be interpreted as MAP or MMSE approximations, can also serve as the Stage 1 estimator.

Specifically, we compare our method with RED-diff (Mardani et al., 2024), RMP-RPS (Xue et al., 2025), and DMAP (Xu et al., 2025). RED-diff can be viewed as MAP estimation since it uses a point-mass distribution $\delta(\mathbf{x} - \mathbf{x}_0)$ as the variational distribution. Thus minimizing the KL divergence reduces to searching for a high-probability point. RMP-RPS attempts to propagate the MMSE solution at each timestep to approximate the posterior mean, which is accurate when the posterior is a unimodal Gaussian. We note that both RMP-RPS and our proposed MAP approximation can be regarded as practical approximations to the MMSE solution under appropriate assumptions. In addition, we include DMAP because it enforces the so-called "local MAP" update, which is closely related in name to MAP estimation. However, we emphasize again that local MAP does not generally converge to the global MAP solution. We refer to (Zhang et al., 2025c) for further discussion.

We evaluate these alternatives on denoising tasks with $\sigma_{\mathbf{y}} = 0.3$, using each method as the candidate Stage 1 estimator. The resulting D-P curves are shown in Figure 6b. Our proposed estimator achieves the best D-P tradeoff. Moreover, RED-diff, RMP-RPS, and DMAP typically require 400-1000 NFEs, whereas our Stage 1 MAP solver only uses 60 NFEs for denoising, demonstrating a clear computational advantage.

### E.5. Differences from CCDF.

One may note that CCDF (Chung et al., 2022) also adopts a two-stage framework, where an initial reconstruction is followed by a renoised posterior sampling stage. Despite this superficial similarity, MAP-RPS differs fundamentally from CCDF in its objective, theoretical focus, and implementation.

First, the two methods are designed for different purposes. CCDF aims to accelerate inverse problem solving by skipping a large number of diffusion timesteps. Accordingly, it is typically evaluated with a small number of sampling steps, e.g., starting from $t = 100$. In contrast, MAP-RPS is designed to traverse the distortion-perception tradeoff. Its Stage 2 intentionally performs posterior sampling from progressively larger noise levels, so that the solution can be steered toward lower perceptual error. This difference in objective is essential: CCDF seeks faster reconstruction, whereas MAP-RPS seeks controllable D-P traversal.

Second, the theoretical analyses focus on different quantities. CCDF mainly analyzes the $\ell_2$ reconstruction error in the

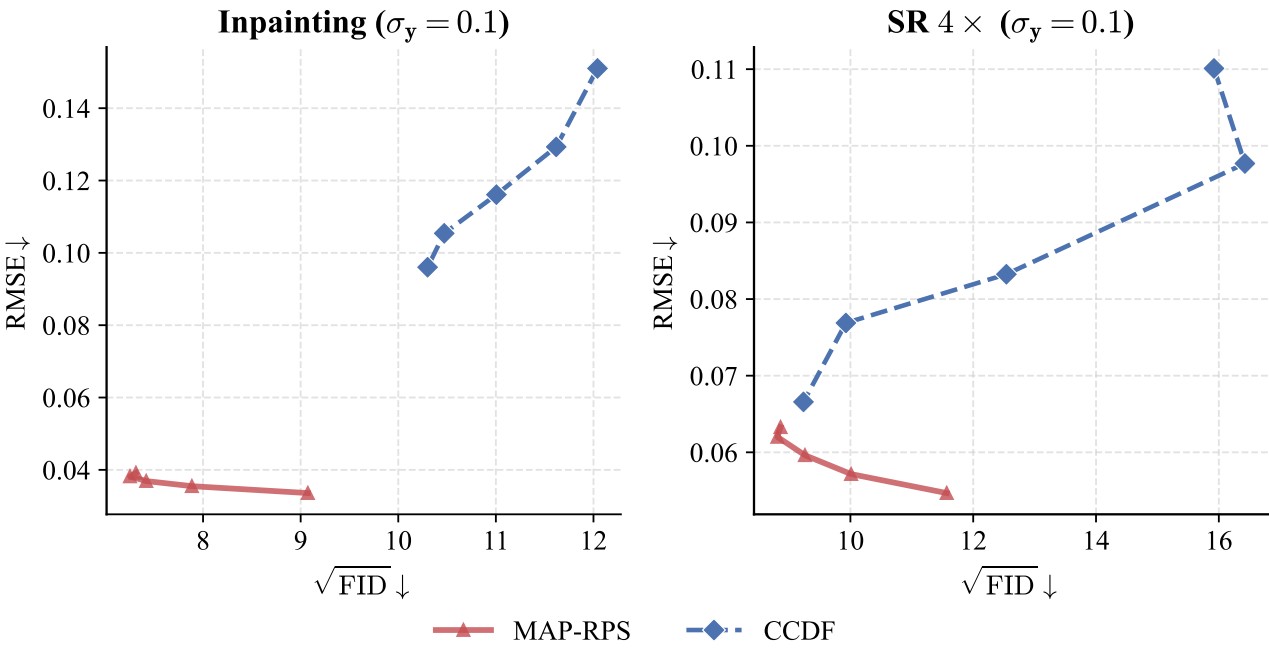

*Figure 7.* Comparison between CCDF and MAP-RPS under the same zero-shot setting on FFHQ.

posterior sampling stage, while our analysis focuses on the perceptual error, measured by $W_2$, in the RPS stage. One may note that CCDF also discusses different diffusion processes. These processes mainly correspond to different parameterizations of the drift and diffusion terms. Since such parameterizations are equivalent up to appropriate weighting factors, our analysis can be adapted to these diffusion processes as well.

Third, the implementations are different. CCDF relies on an additionally trained network to provide the initialization, thereby reducing the NFEs required by diffusion sampling. In contrast, MAP-RPS uses the same diffusion prior to compute the Stage 1 MAP estimate, without introducing any extra network. This is important for the setting considered in this work, where only a pretrained diffusion model is assumed to be available. Moreover, training such an initialization network typically requires task-specific paired data, which is not consistent with the zero-shot setting.

Finally, we provide an empirical comparison with CCDF under a unified setting, where only a single diffusion model is available and pseudo-inverse initialization is applied for CCDF. The resulting D-P curves are shown in Figure 7. We observe that CCDF generally benefits from more sampling steps in both distortion and perception metrics, whereas MAP-RPS yields a clear tradeoff between RMSE and FID. This further confirms the fundamental difference between the two methods.

### E.6. Comparison under the same NFEs.

One source of the efficiency advantage of LMAP-RPS is that it can operate with relatively few NFEs. Though we have used the recommended NFEs for each baseline in the main experiments to ensure their best performance, we further compare all methods under the same computational budget. Specifically, we evaluate on FFHQ inpainting and anisotropic deblurring tasks, where LMAP-RPS with $t_0 = 0$ uses 200 MAP optimization iterations for both tasks. We therefore reduce the NFEs of all baselines to 200 for a fair comparison. The results are reported in Table 8. Under the same computational budget, LMAP-RPS with $t_0 = 0$ still achieves competitive performance, with a clear advantage especially in distortion metrics.

*Table 8.* Comparison under the same computational budget on MS-COCO inpainting and anisotropic deblurring. All methods use 200 NFEs.

| Method | NFEs | Inpainting | | | | Deblurring | | | |
|---|---|---|---|---|---|---|---|---|---|
| | | PSNR↑ | SSIM↑ | LPIPS↓ | FID↓ | PSNR↑ | SSIM↑ | LPIPS↓ | FID↓ |
| Latent-DPS | 200 | 21.89 | 0.5527 | 0.4512 | 202.64 | 17.40 | 0.3860 | 0.5310 | 316.23 |
| Resample | 200 | 24.20 | 0.6464 | 0.3722 | 162.58 | 24.63 | 0.6267 | 0.3978 | 98.46 |
| PSLD | 200 | 25.23 | 0.6595 | 0.4118 | 131.45 | 23.19 | 0.6123 | 0.4566 | 130.54 |
| STSL | 200 | 27.66 | 0.7772 | 0.3058 | 91.53 | 22.00 | 0.5325 | 0.4785 | 167.41 |
| LDIR | 200 | 27.94 | 0.7880 | 0.3116 | 78.44 | 26.00 | 0.7228 | 0.3769 | 107.70 |
| Latent-DCDP | 200 | 26.91 | 0.7650 | 0.3205 | 80.48 | 25.46 | 0.6927 | 0.3842 | 106.60 |
| Latent-DMAP | 200 | 27.18 | 0.7636 | 0.3276 | 101.18 | 24.71 | 0.6718 | 0.4246 | 136.67 |
| Latent-DAPS | 200 | 25.90 | 0.6941 | 0.3955 | 104.29 | 23.69 | 0.5851 | 0.4555 | 139.05 |
| Latent-STICOM | 200 | 26.99 | 0.7527 | 0.3743 | 126.07 | 23.59 | 0.6271 | 0.4661 | 184.46 |
| **LMAP-RPS (0)** | 200 | **28.14** | **0.7989** | **0.2769** | **61.35** | **26.42** | **0.7322** | **0.3504** | **90.80** |

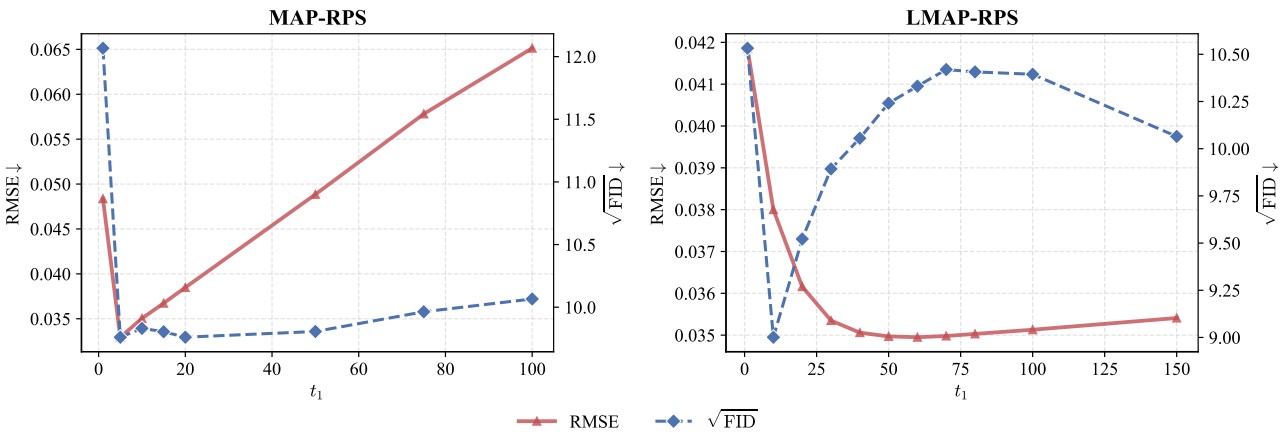

*Figure 8.* The performance of MAP-RPS and LMAP-RPS on the denoising task ($\sigma_\mathbf{y} = 0.3$) on FFHQ with varying $t_1$.

## F. Ablation studies

### F.1. Influence of $t_1$ on the MAP estimation

In Theorem 3.3, we approximate the stochastic gradient of the prior term using a fixed diffusion time step $t_1$. Note that Theorem 3.3 relies on the following assumption:

$$p_{X_0|X_{t_1}}(\mathbf{x}_0|\mathbf{x}_{t_1}) \propto \mathcal{N}\left(\mathbb{E}\left[X_0|X_{t_1} = \mathbf{x}_{t_1}\right], r_{t_1}^2 \mathbf{I}\right). \tag{83}$$

This assumption is generally only an approximation for arbitrary data distributions, and becomes more accurate when either $t_1$ is sufficiently small, or when the local structure of $X_0$ is dominated by a unimodal Gaussian component. Consequently, one would prefer to choose a small $t_1$ for gradient estimation. However, as pointed out in Zhang et al. (2024b), the learned score function exhibits a singularity at $t = 0$, which makes network approximation increasingly inaccurate as $t \to 0$. Therefore, to balance the validity of (83) and the accuracy of the learned score, $t_1$ should be chosen to be small but not excessively close to zero.

This choice is reflected in all our experiments, where we set $t_1 = 50$ for LMAP-RPS and $t_1 = 10$ for MAP-RPS. We further investigate the effect of varying $t_1$ by reporting the RMSE and FID performance on the denoising task with $\sigma_\mathbf{y} = 0.3$ on FFHQ. The results are shown in Figure 8. As $t_1$ approaches zero, the performance of both MAP-RPS and LMAP-RPS degrades sharply, which can be attributed to the large approximation error caused by the singularity at $t = 0$. On the other hand, when $t_1$ becomes larger, the performance of MAP-RPS deteriorates significantly, particularly in terms of RMSE,

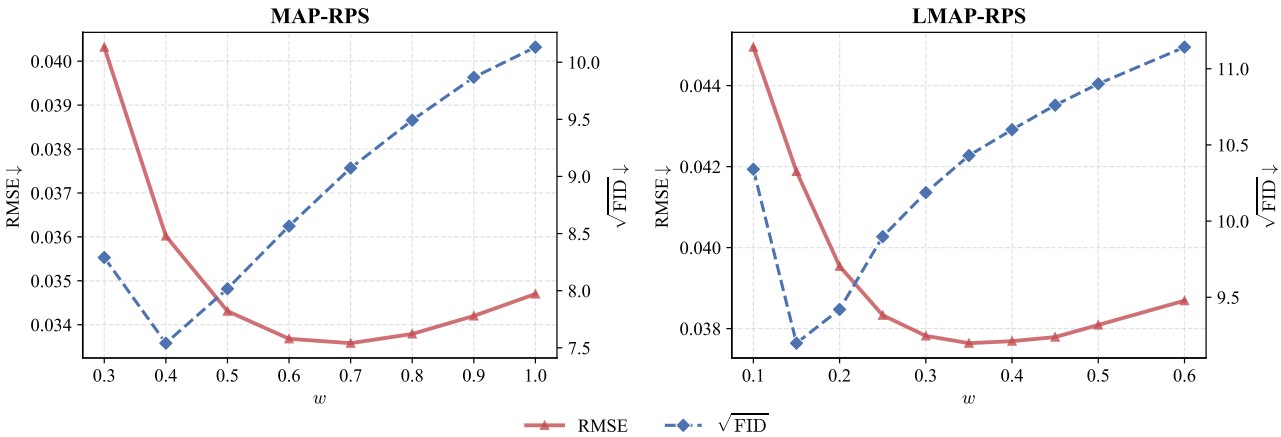

*Figure 9.* The performance of MAP-RPS and LMAP-RPS on the inpainting task ($\sigma_{\mathbf{y}} = 0.1$) on FFHQ with varying $w$.

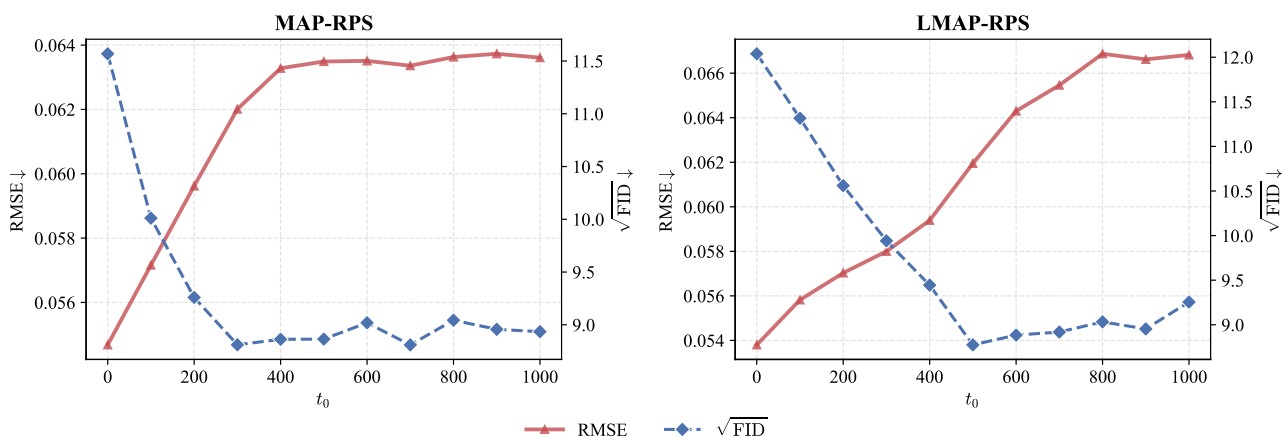

*Figure 10.* The performance of MAP-RPS and LMAP-RPS on the super-resolution task ($\sigma_{\mathbf{y}} = 0.1$) on FFHQ with varying $t_0$.

while LMAP-RPS exhibits a much milder degradation. This behavior aligns with our expectations: the VAE latent space is typically well modeled as a mixture of Gaussians, and in many regions it is dominated by a single Gaussian mode. This property closely matches the approximate validity of the assumption in Theorem 3.3, explaining why larger diffusion times do not introduce substantial estimation error for LMAP-RPS.

### F.2. Influence of the weight $w$

In the implementation of the MAP stage, we introduce a weighting parameter $w$ in Eqs. (79) and (80) to balance the prior gradient and the likelihood gradient. Here we study how perturbations of $w$ affect the performance of the MAP stage. We conduct experiments on the inpainting task with $\sigma_{\mathbf{y}} = 0.1$ on FFHQ. For MAP-RPS, we vary $w \in \{0.3, 0.4, 0.5, 0.6, 0.7, 0.8, 0.9, 1.0\}$, while for LMAP-RPS we consider $w \in \{0.1, 0.15, 0.20, 0.25, 0.30, 0.35, 0.40, 0.45, 0.5, 0.6\}$. The results are shown in Figure 9.

For both MAP-RPS and LMAP-RPS, RMSE and FID exhibit a U-shaped trend as $w$ varies, initially decreasing and then increasing. A moderately chosen $w$ provides a better balance between observation consistency and the diffusion prior, leading to improved reconstruction quality. We note that the optimal FID and optimal RMSE are not achieved at the same value of $w$: the value of $w$ minimizing FID is consistently smaller than that minimizing RMSE. This can be attributed to the fact that a smaller $w$ places more emphasis on the data fidelity term $\|\mathbf{y} - \mathcal{A}(\mathbf{x})\|^2$, which may cause the algorithm to slightly overfit the noise. Such overfitting can increase sample diversity, thereby reducing FID within a certain range.

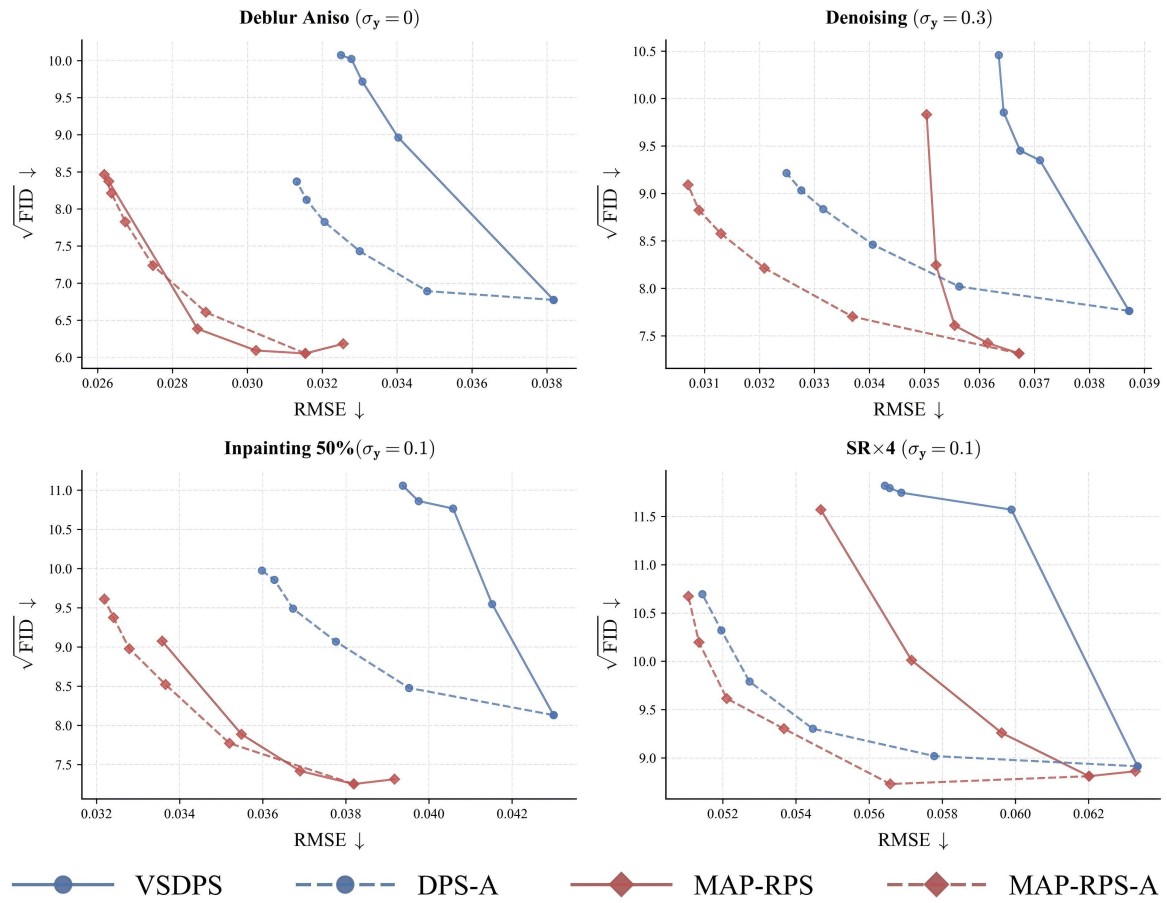

*Figure 11.* D-P curves of MAP-RPS and DPS with sample averaging.

## F.3. Performance with a full range of $t_0$

Here we present a more comprehensive analysis of how RMSE and FID vary with respect to $t_0$ for both MAP-RPS and LMAP-RPS. We consider $t_0 \in \{0, 100, 200, 300, 400, 500, 600, 700, 800, 900, 1000\}$ on the $4\times$ super-resolution task on FFHQ, and the results are shown in Figure 10. We observe that FID exhibits a desirable and consistent decreasing trend for $t_0 \leq 500$, while this trend does not persist with larger $t_0$. In particular, as $t_0 \to T$, FID no longer improves and even slightly increases.

This behavior is also substantiated by our theoretical results in (54) and (69). Specifically, the derived upper bound on the $W_2$ distance consists of two terms. The first term exhibits a decay in $(\overline{\alpha}_{t_0})^{1-L_s}$, which dominates the improvement in FID when $t_0 \leq 500$. The second term represents the cumulative estimation error of the posterior score as

$$\int_0^{t_0} g^2(r)\,(\overline{\alpha}_r)^{\frac{1}{2}-L_s}\,[\mathbb{E}_{p_r(\cdot|\mathbf{y})}\|\mathbf{s}_\theta - \nabla \log p_r(\cdot \mid \mathbf{y})\|^2]^{1/2}\mathrm{d}r. \tag{84}$$

Though this term is bounded by $\epsilon_{\text{score}}$ under our assumptions, it is monotonically increasing with respect to $t_0$. In particular, we note that commonly used posterior score estimation methods, such as DPS and PSLD, tend to be more accurate at smaller diffusion timesteps $t$. As a result, when $t_0 \to T$, the accumulated posterior score estimation error becomes non-negligible and may offset the improvement brought by the decaying $\overline{\alpha}_{t_0}$ term. Consequently, the overall performance, as measured by FID, may saturate or slightly degrade for large $t_0$.

Therefore, in practice, it is sufficient to set $t_0$ to a moderate value, typically around $500$, as also adopted in our inverse problem experiments on MS-COCO. This choice not only achieves a favorable D-P tradeoff, but also improves efficiency since it avoids computing the full posterior sampling chain.

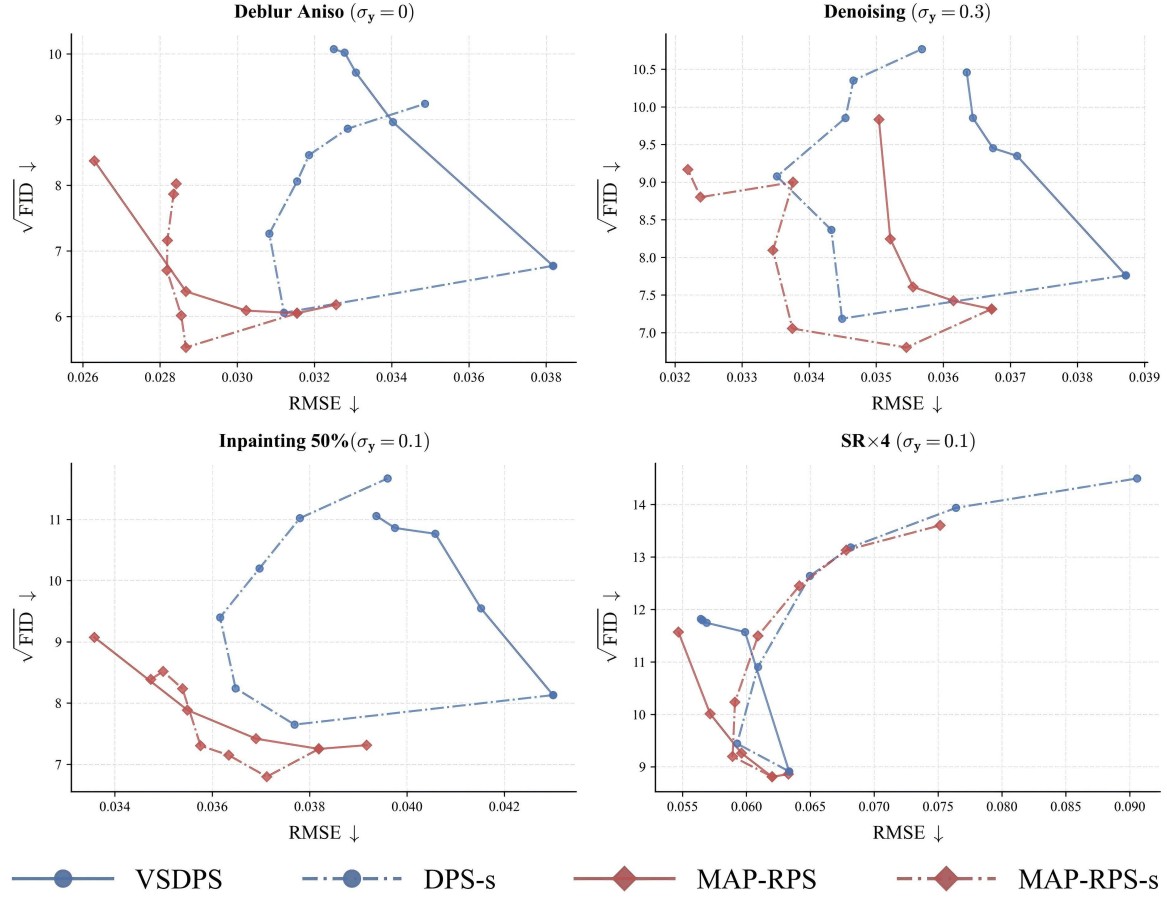

*Figure 12.* D-P curves of MAP-RPS and DPS with reduced discretization steps.

## F.4. Combining with empirical strategies

Here we investigate the possibility of combining MAP-RPS with empirical strategies discussed in Appendix A, namely sample averaging and reducing the number of discretization steps. We select the MAP-RPS with an appropriate $t_0$ that achieves the best FID as the base algorithm and apply the two strategies on top of it. For comparison, we apply the same techniques to DPS, and use the D-P curves of MAP-RPS and VSDPS as baselines. The results of sample averaging are shown in Figure 11, where we denote the resulting methods as MAP-RPS-A and DPS-A, respectively. The results obtained by reducing the number of discretization steps are shown in Figure 12, and the corresponding methods are denoted as MAP-RPS-s and DPS-s.

We observe that sample averaging leads to noticeable improvements for both methods, enabling a relatively favorable D-P tradeoff. However, this improvement comes at the cost of a significantly increased computational burden. When the number of samples ranges from $M = 1$ to $32$, the runtime required to reconstruct a single image using MAP-RPS-A and DPS-A is reported in Table 9. In particular, DPS-A requires nearly 30 minutes to process a single image when $M = 32$, which is prohibitively expensive. Even MAP-RPS-A typically takes around 10 minutes under the same settings. In contrast, the original MAP-RPS can achieve a competitive D-P tradeoff within a few seconds, highlighting its superior efficiency.

Reducing the number of discretization steps yields only limited benefits for both MAP-RPS and DPS. In most cases, this strategy enables tradeoffs only within a narrow range, and when the number of steps becomes too small, the RMSE of both methods actually increases. We note that approaches that achieve a D-P tradeoff by reducing discretization steps are mostly based on conditional diffusion models (Delbracio & Milanfar, 2023; Ren et al., 2023; Yue et al., 2024; Luo et al., 2024; Zhussip et al., 2024; Cohen et al., 2025), where the observation is injected as a condition. This observation injection allows effective restoration with relatively few sampling steps. In contrast, for zero-shot inverse problems, where no observation is injected into the neural network, an insufficient number of posterior sampling steps typically fails to ensure high-quality

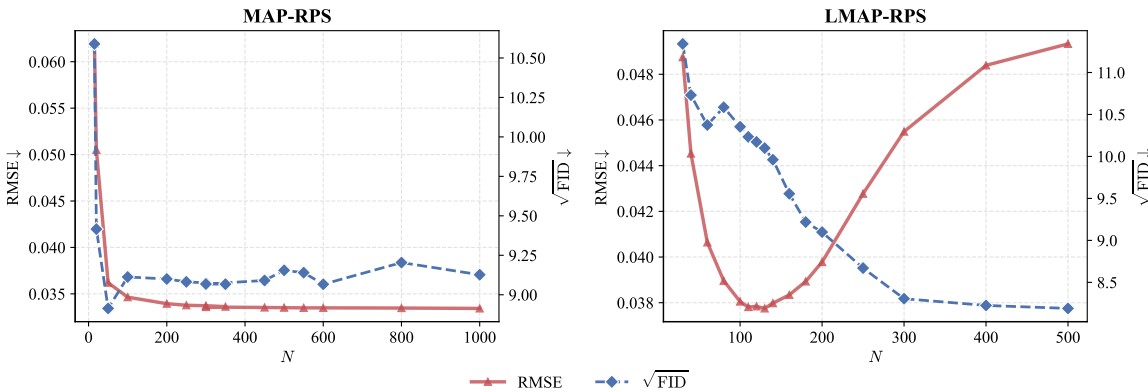

*Figure 13.* The performance of MAP-RPS and LMAP-RPS on the inpainting task ($\sigma_{\mathbf{y}} = 0.1$) on FFHQ with varying $N$.

*Table 9.* Runtime scaling of MAP-RPS-A and DPS-A with respect to $M$ (seconds per image).

| Task | Method | $M = 1$ | $M = 2$ | $M = 4$ | $M = 8$ | $M = 16$ | $M = 32$ |
|------|--------|---------|---------|---------|---------|----------|----------|
| Inpainting | MAP-RPS-A | 20 | 35 | 70 | 140 | 280 | 560 |
| | DPS-A | 58 | 118 | 236 | 472 | 944 | 1888 |
| SR 4× | MAP-RPS-A | 32 | 59 | 118 | 236 | 472 | 944 |
| | DPS-A | 60 | 122 | 244 | 488 | 976 | 1952 |
| Deblur Aniso | MAP-RPS-A | 14 | 21 | 42 | 84 | 168 | 336 |
| | DPS-A | 59 | 122 | 244 | 488 | 976 | 1952 |
| Denoising | MAP-RPS-A | 8 | 15 | 30 | 60 | 120 | 240 |
| | DPS-A | 59 | 120 | 240 | 480 | 960 | 1920 |

reconstructions.

### F.5. Influence of the optimization iterations $N$

Figure 13 illustrates the effect of the number of optimization steps $N$ in the MAP stage on both LMAP-RPS and MAP-RPS, evaluated on the inpainting task on FFHQ with $\sigma_{\mathbf{y}} = 0.1$. When $N$ is small, the performance of both methods degrades due to under-optimization. As $N$ increases, the performance of MAP-RPS quickly stabilizes, indicating that it converges reliably to a neighborhood of the MAP solution. In contrast, LMAP-RPS exhibits a different behavior when $N$ becomes large, where the RMSE gradually increases while the FID continues to decrease. We attribute this phenomenon to the approximation of the likelihood term in (80), which introduces a bias such that the optimal solution does not exactly coincide with the true MAP estimate. As a result, LMAP-RPS tends to slightly overfit the observation noise, leading to enhanced sample diversity and consequently a further reduction in FID, as discussed in Appendix F.2.

### F.6. Complete runtime comparison on MS-COCO

Table 10 reports the complete runtime comparison of eleven algorithms across six inverse problems on MS-COCO. LMAP-RPS exhibits a significant efficiency advantage at $t_0 = 0$, which mainly stems from the fact that the latent-space MAP stage only requires 100–500 optimization steps. When $t_0 = 600$, LMAP-RPS additionally performs 600 steps of posterior sampling; however, the overall computational cost remains relatively low. As a result, its runtime is still faster than most competing methods, and is only slightly slower than Latent-DCDP and Latent-DPS in certain cases.

*Table 10.* Runtime comparison on MS-COCO dataset (seconds per image).

| Method | Inpainting | SR $4\times$ | Deblur Aniso | CS $2\times$ | HDR | Nonlinear Deblur |
|---|---|---|---|---|---|---|
| Latent-DPS | 242 | 250 | 239 | 240 | 241 | 245 |
| ReSample | 1102 | 1205 | 1155 | 1327 | 1104 | 1962 |
| PSLD | 262 | 273 | 265 | 265 | – | – |
| STSL | 444 | 457 | 454 | 463 | 457 | 460 |
| LDIR | 294 | 303 | 297 | 295 | 293 | 305 |
| Latent-DCDP | 110 | 224 | 111 | 334 | 110 | 122 |
| Latent-DMAP | 283 | 294 | 287 | 284 | 284 | 287 |
| Latent-DAPS | 464 | 475 | 466 | 485 | 464 | 510 |
| Latent-SITCOM | 269 | 211 | 348 | 342 | 280 | 358 |
| LMAP-RPS (0) | 45 | 23 | 44 | 68 | 89 | 125 |
| LMAP-RPS (600) | 189 | 172 | 187 | 213 | 230 | 263 |

# G. More visualization results

Here we present additional qualitative results. The D–P traversal results of MAP-RPS and LMAP-RPS on four inverse tasks on FFHQ are shown in Figures 15, 16, 17, and 18, together with those of VSDPS, PSCGAN-A, and PSCGAN-z. In addition, qualitative comparisons between LMAP-RPS and nine other inverse methods on MS-COCO are presented in Figures 19 and 20.

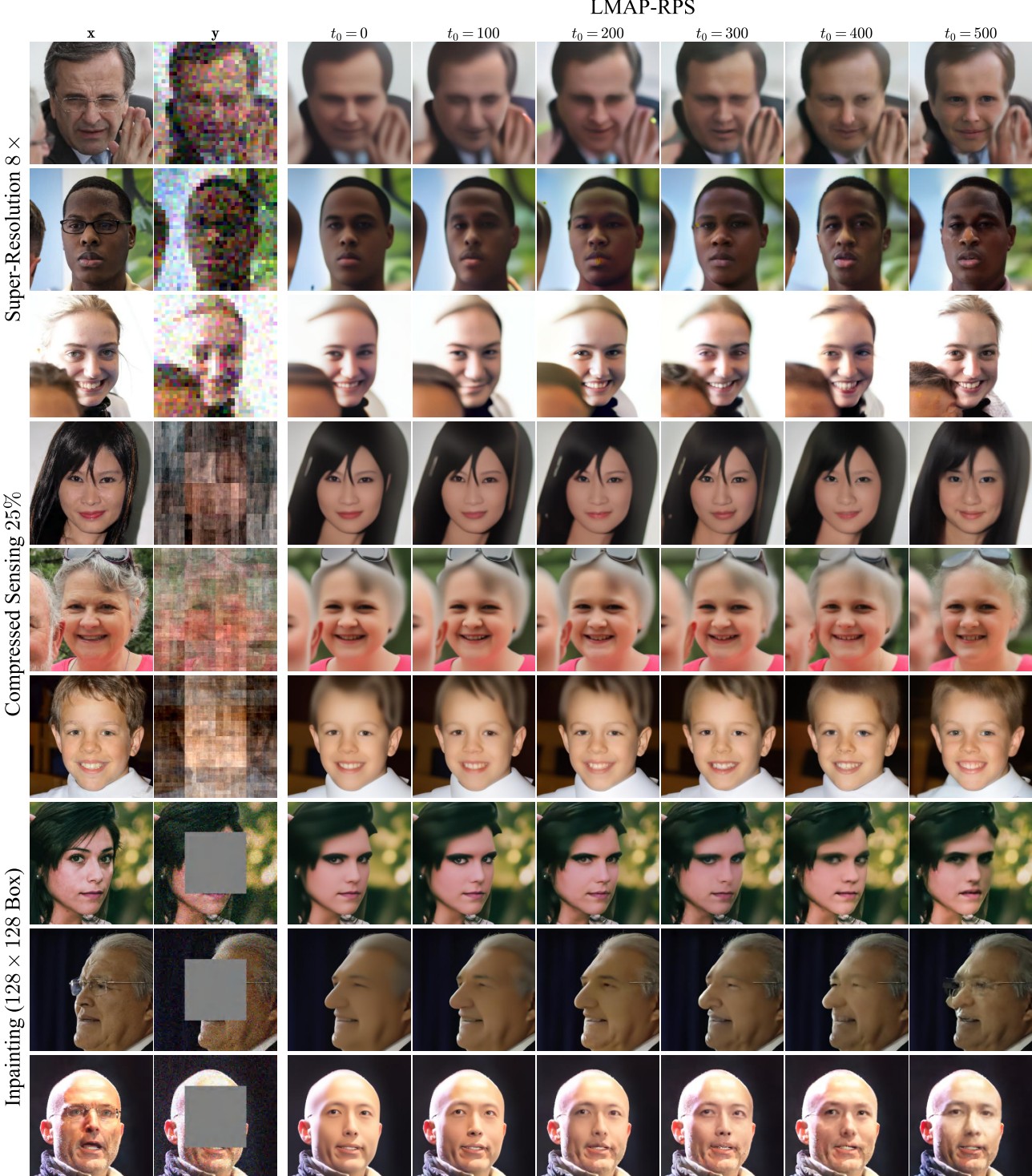

*Figure 14.* Samples obtained by LMAP-RPS with varying $t_0$ on $8\times$ super-resolution, $4\times$ compressed sensing and $128 \times 128$ box inpainting on FFHQ ($\sigma_{\mathbf{y}} = 0.1$).

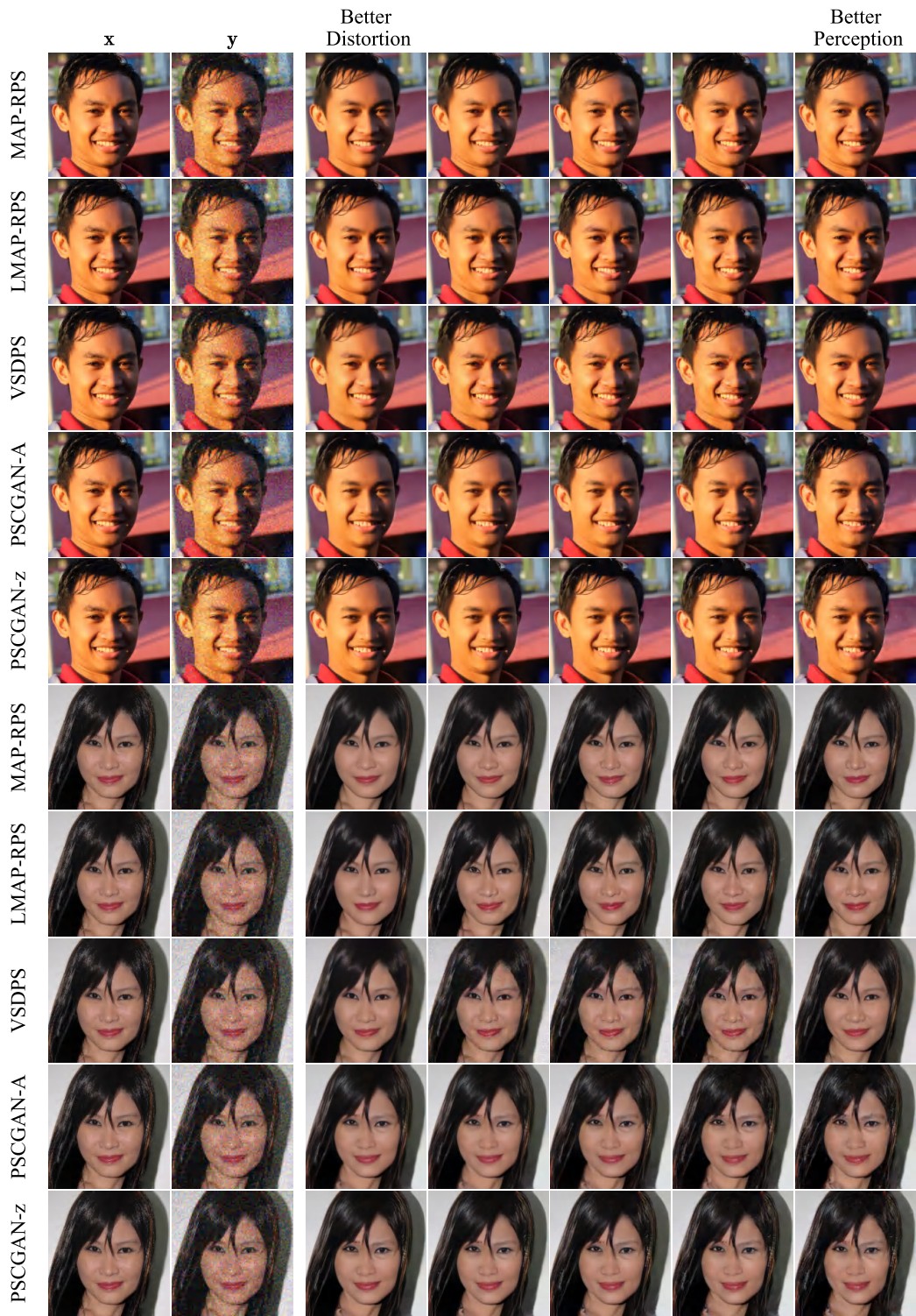

*Figure 15.* Visualizations of D-P traversal of MAP-RPS, LMAP-RPS, VSDPS, PSCGAN-A, and PSCGAN-z on the denoising task ($\sigma_{\mathbf{y}} = 0.3$) on FFHQ.

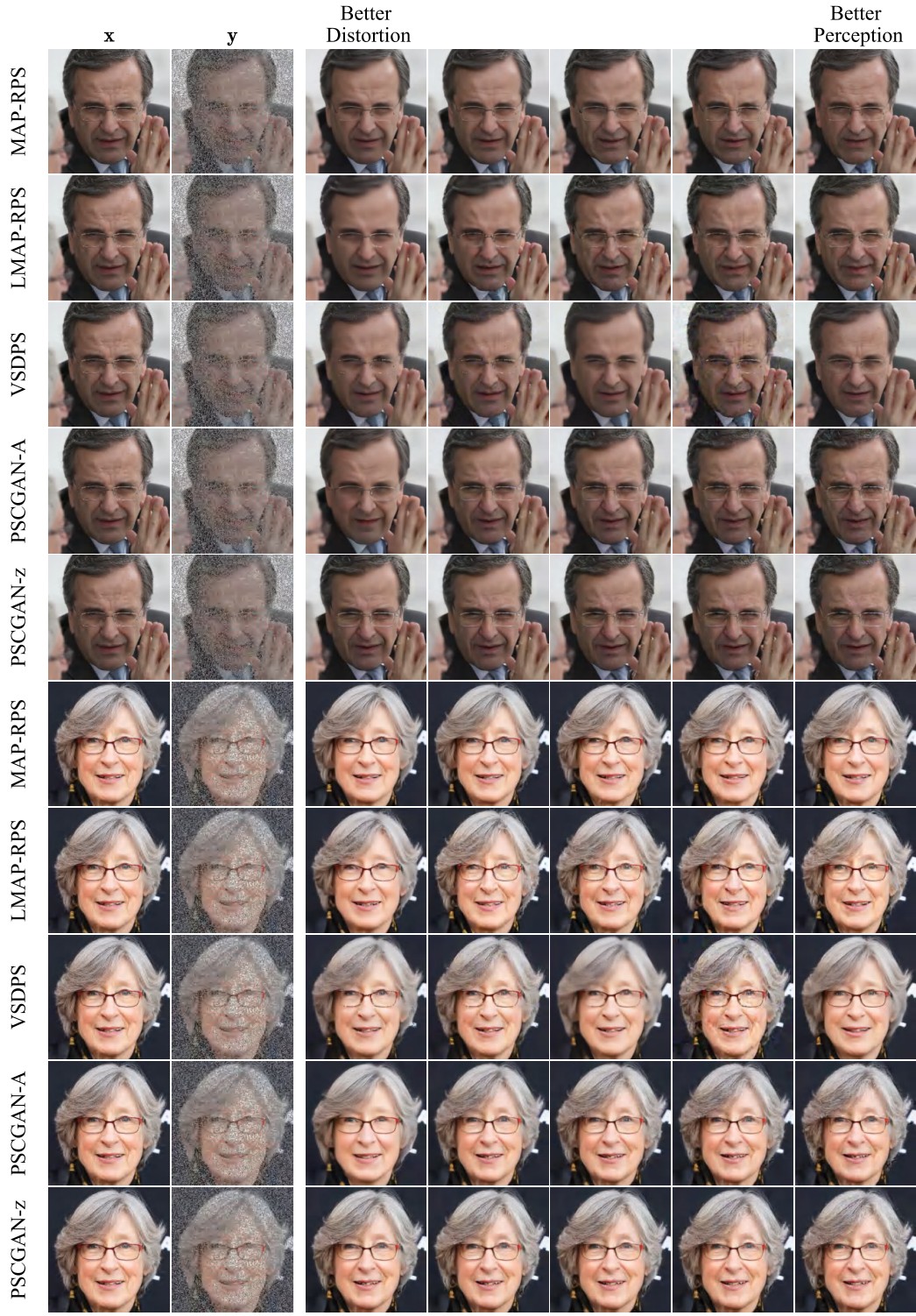

*Figure 16.* Visualizations of D-P traversal of MAP-RPS, LMAP-RPS, VSDPS, PSCGAN-A, and PSCGAN-z on the inpainting task ($\sigma_{\mathbf{y}} = 0.1$) on FFHQ.

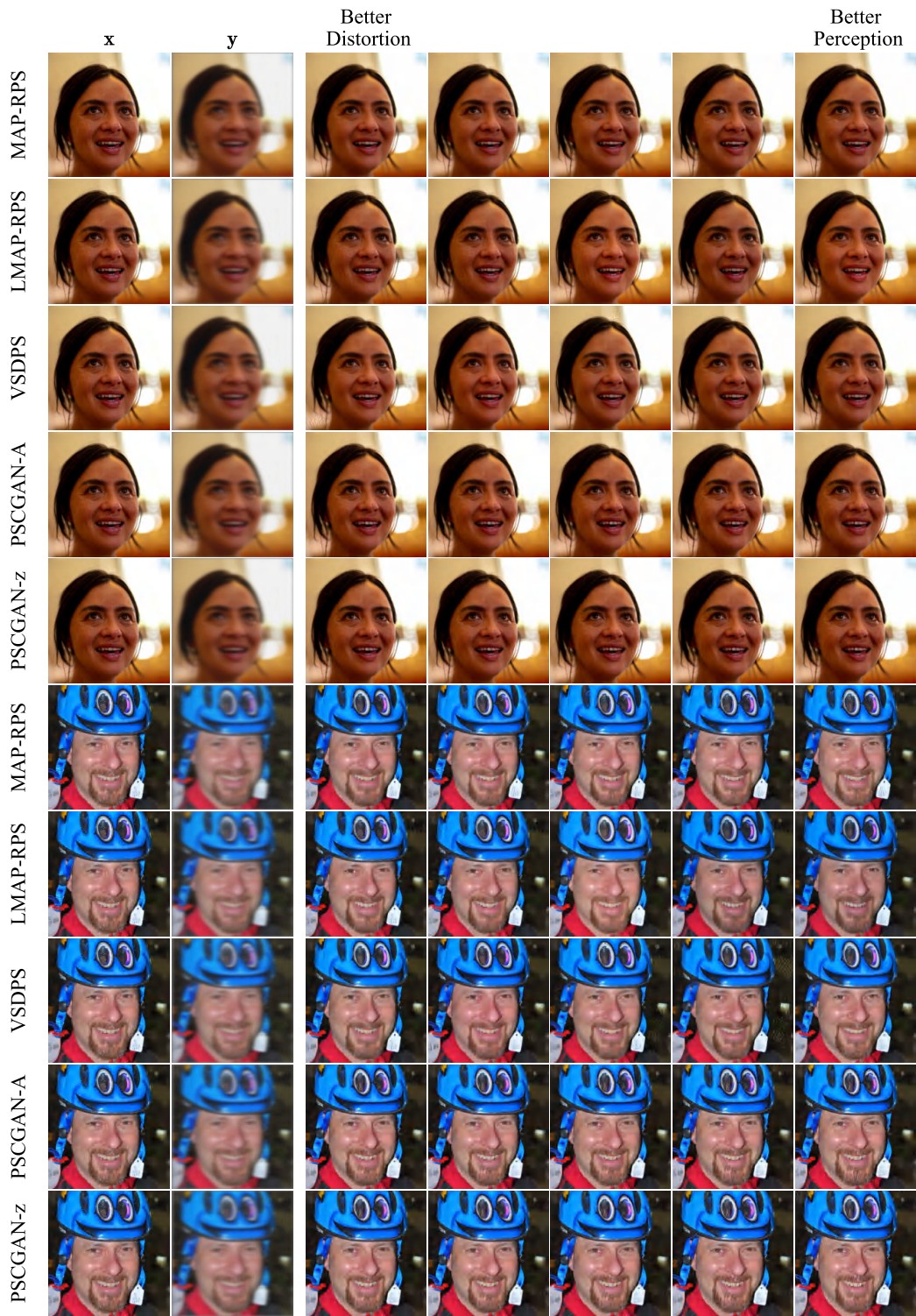

*Figure 17.* Visualizations of D-P traversal of MAP-RPS, LMAP-RPS, VSDPS, PSCGAN-A, and PSCGAN-z on the anisotropic deblurring task ($\sigma_{\mathbf{y}} = 0.0$) on FFHQ.

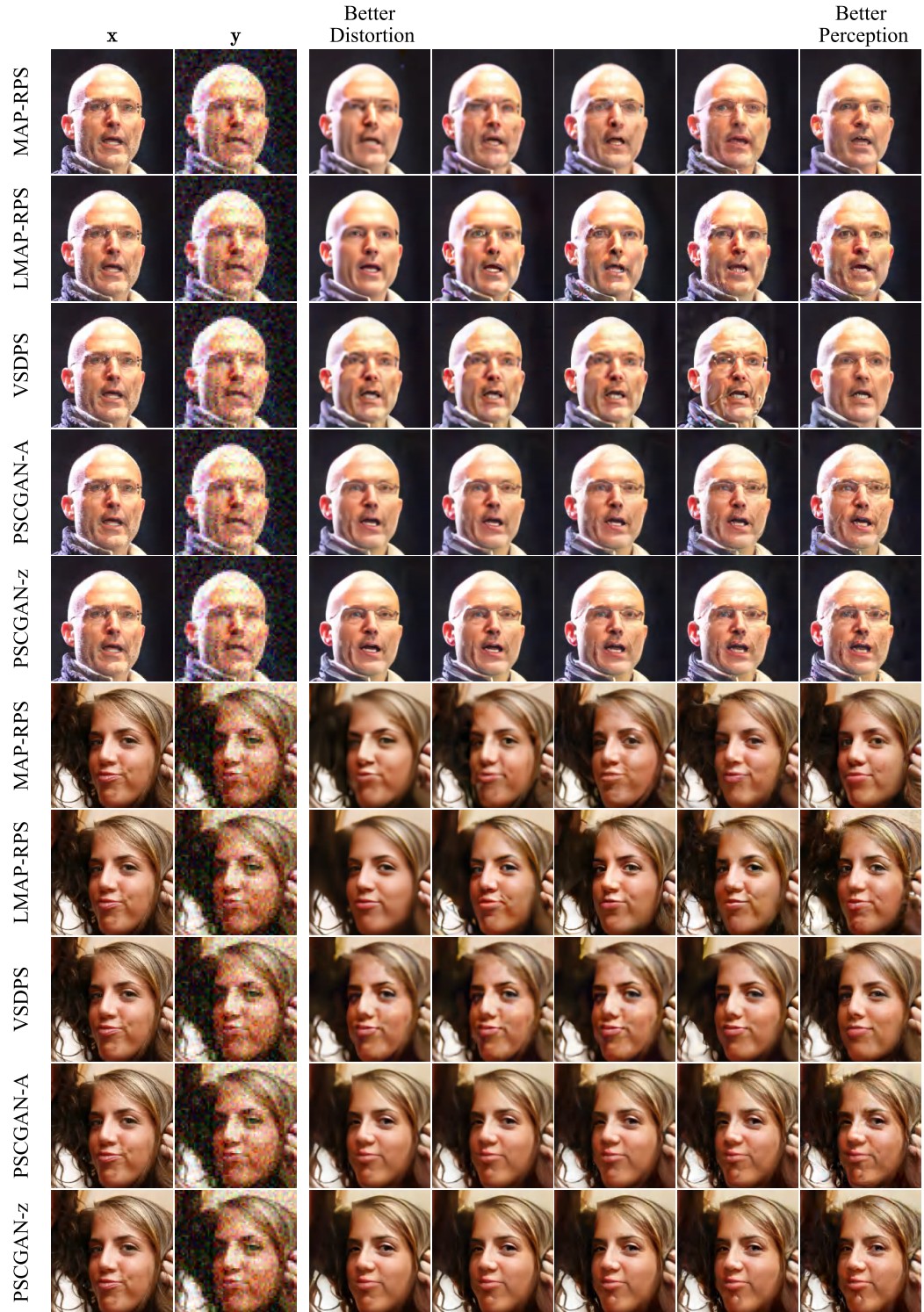

*Figure 18.* Visualizations of D-P traversal of MAP-RPS, LMAP-RPS, VSDPS, PSCGAN-A, and PSCGAN-z on the super-resolution task ($\sigma_{\mathbf{y}} = 0.1$) on FFHQ.

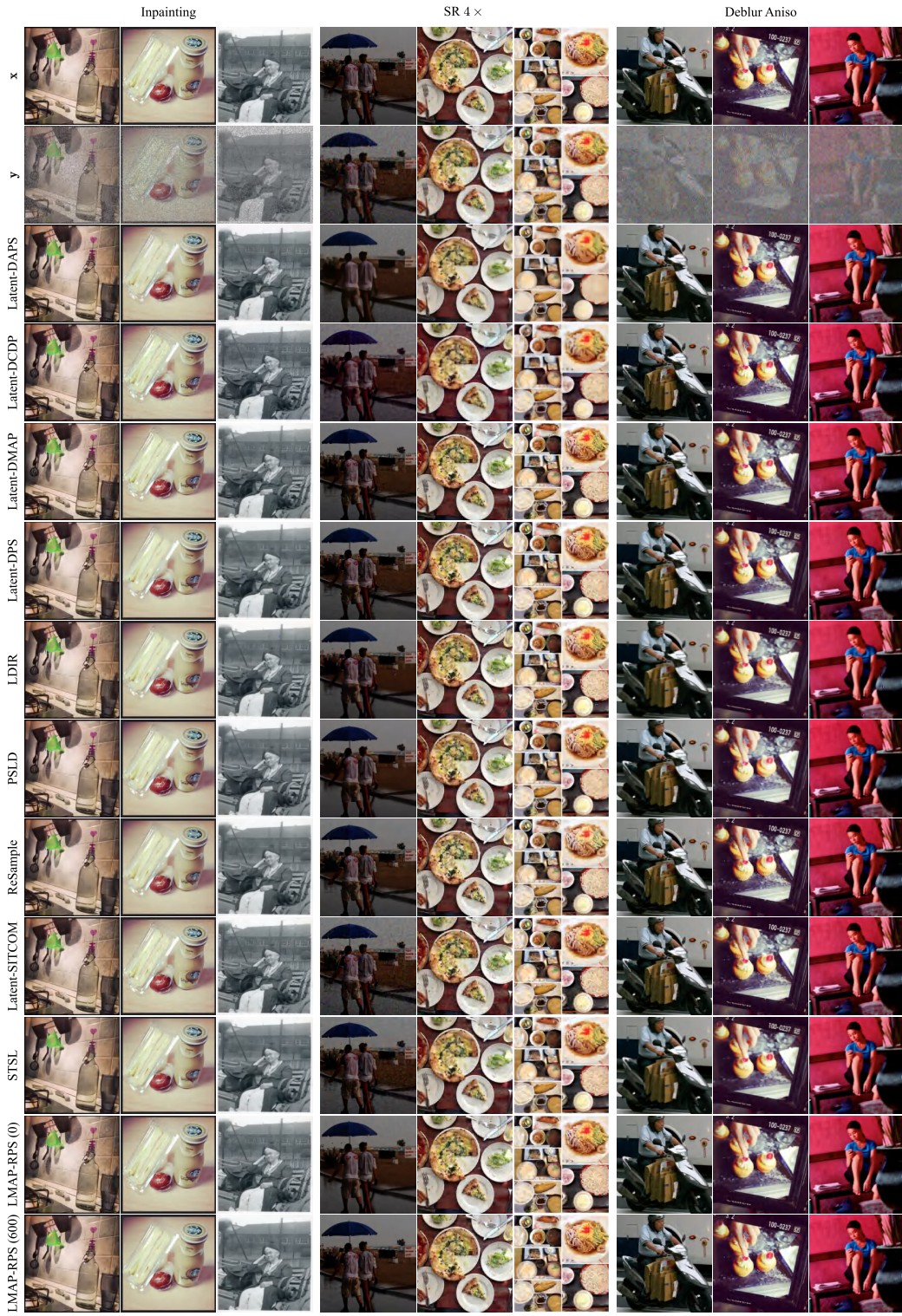

*Figure 19.* Qualitative comparison of eleven inverse algorithms on inpainting, super-resolution, and anisotropic deblurring on MS-COCO.

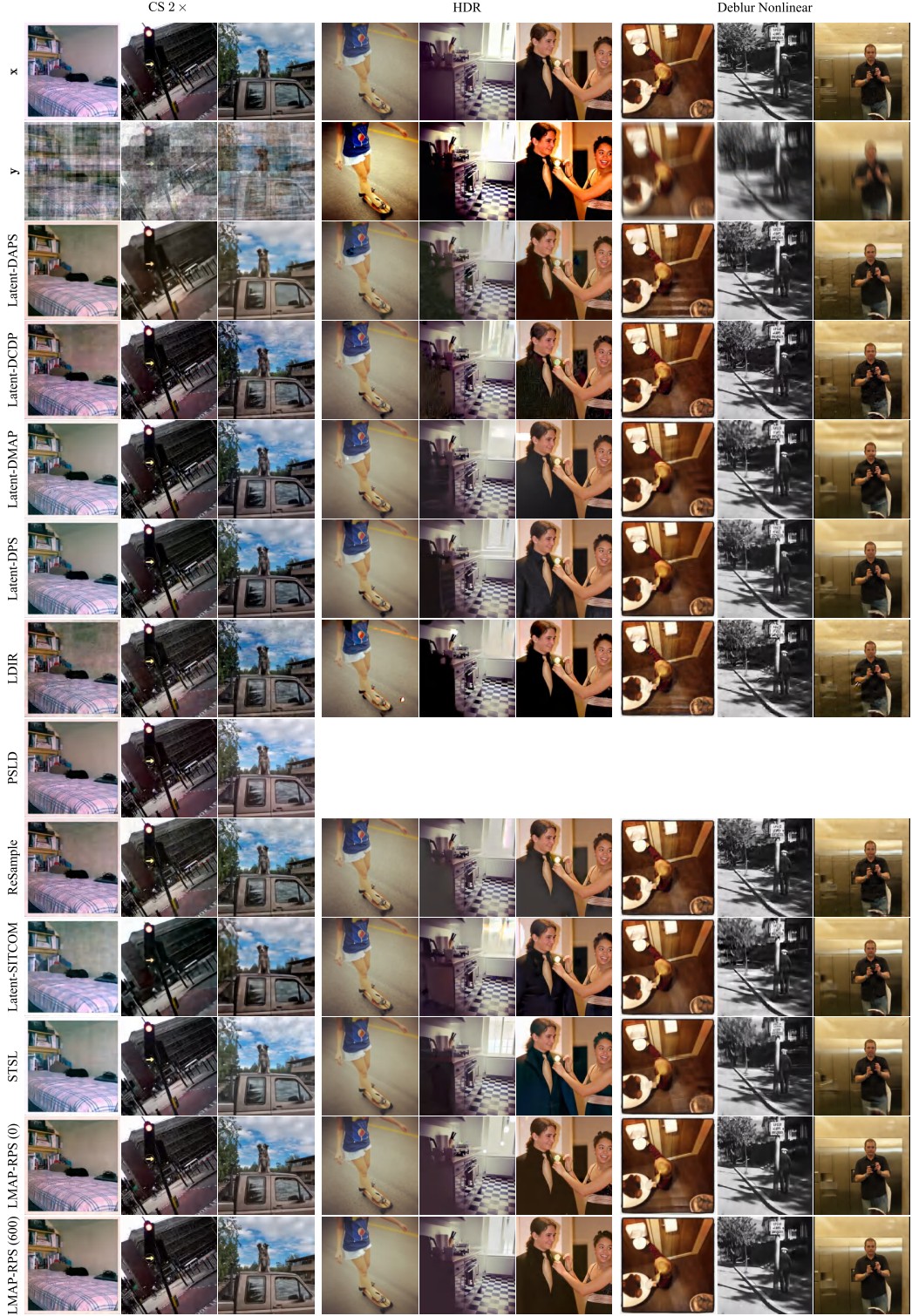

*Figure 20.* Qualitative comparison of eleven inverse algorithms on compressed sensing, high dynamic range reconstruction, and nonlinear deblurring on MS-COCO.

