# OpenReview forum: "Stage-wise Distortion–Perception Traversal in Zero-shot Inverse Problems with Diffusion Models"
_ICML.cc/2026/Conference — ICML 2026 regular_

### Official Review · Reviewer_N4k8 · 2026-03-06

**Soundness:** 3
**Presentation:** 3
**Significance:** 2
**Originality:** 2
**Overall Recommendation:** 3
**Confidence:** 4

**Summary:**

This paper studies the distortion–perception (D-P) trade-off in diffusion-based inverse problems and proposes a stage-wise framework, termed MAP-RPS, to enable controllable traversal of this trade-off at inference time. The method first obtains a low-distortion estimate through a MAP-based initialization and then improves perceptual quality via re-noised posterior sampling (RPS). The framework is theoretically analyzed for both stages and further extended to latent diffusion models (LMAP-RPS). Experiments on several inverse problems demonstrate competitive performance and flexible D-P control.

**Compliance With Llm Reviewing Policy:**

Affirmed.

**Final Justification:**

Considering the authors’ responses and the comments from other reviewers, I find that the paper still has issues with some of its assumptions. Overall, I maintain my original score of 3.

**Key Questions For Authors:**

Overall, while the MAP-RPS provides an effective approach for improving the D-P trade-off in diffusion-based inverse problems, several aspects of the method would benefit from further clarification and analysis. My main concerns are summarized in the Weaknesses section.

Please address the following points:

1. The theoretical analysis of Stage 1 assumes strongly log-concave posteriors, which may be restrictive for highly underdetermined inverse problems (e.g., large-factor SR or high compression CS). It would be helpful to analyze whether the theory holds under more challenging regimes and provide empirical evaluation when this assumption is violated.

2. The relationship between MAP-RPS and CCDF is not clearly discussed. Both methods adopt a similar two-stage pipeline with an initial reconstruction followed by RPS. Empirical comparisons and clearer discussion would help highlight the advantages of the proposed framework.

3. The design choice of using diffusion-based MAP estimation for Stage 1 initialization is not fully justified. Many modern end-to-end IR models (e.g., SwinIR, MambaIR) approximate the posterior mean estimator and provide high-quality reconstructions with a single forward pass. It would be helpful to discuss why MAP initialization is preferred over such alternatives.

4. The proposed pipeline introduces additional inference-time complexity (e.g., iterative MAP optimization and RPS). A clearer analysis of the computational overhead—such as runtime, NFEs, and comparisons with simpler baselines—would help clarify the practical efficiency of the method.

5. There appears to be a possible notation inconsistency in Eq. (47) regarding the term involving $\nabla_x \log p_{X|X_{t_1}}(\cdot)$. Clarifying whether this is a typo would help improve the consistency of the derivation.

**Limitations:**

Yes

**Strengths And Weaknesses:**

Strengths

1. A principled theoretical analysis is provided for the MAP-based initialization. Under strongly log-concave posteriors, the MAP estimator is shown to approximate the MMSE estimator with explicit error bounds, providing theoretical justification for using MAP as an efficient surrogate in diffusion-based inverse problems.

2. A RPS is proposed to traverse the D-P curve by injecting noise into the MAP estimate and performing posterior sampling. Theoretical analysis shows that the $W_2$ distance admits a decreasing upper bound with respect to the re-noising time $t_0$, providing a principled mechanism to control the D-P trade-off.

Weaknesses

1. The theoretical analysis of Stage 1 (MMSE approximation) is derived for strongly log-concave posteriors. While this assumption enables tractable analysis and error bounds, it may be restrictive for many practical inverse problems. For example, in highly underdetermined settings such as super-resolution (SR) with large factors or compressive sensing (CS) with higher compression ratios, the degradation operator $A$ is often rank-deficient, leading to non-strongly-convex likelihoods and potentially multi-modal posteriors. The current experiments appear to focus on relatively moderate settings (e.g., $4\times$ SR or $2\times$ CS). It would be helpful to analyze whether the theoretical assumptions remain reasonable under more challenging regimes (e.g., $8\times$ SR or $4\times$ CS), and to provide empirical evaluation of the method when the strong log-concavity assumption is violated.

2. MAP-RPS shares conceptual similarities with CCDF (Chung et al., 2022), which also adopts a two-stage strategy consisting of an initial reconstruction followed by re-noised posterior sampling (RPS). In CCDF, the initialization is obtained using a feed-forward neural network, whereas the current work uses diffusion-based MAP estimation before applying RPS. Moreover, CCDF provides both theoretical and empirical analyses across several diffusion formulations (e.g., SMLD, DDPM, and DDIM). MAP-RPS does not discuss the relationship with CCDF or provide empirical comparisons, making it difficult to clearly assess the advantages of the proposed framework.

3. The choice of diffusion-based MAP estimation for Stage 1 initialization is not fully justified. Many modern end-to-end image restoration (IR) models (e.g., SwinIR (Liang et al., 2021) and MambaIR (Guo et al., 2024)) are trained to approximate the posterior mean estimator $X_{\text{MMSE}} = \mathbb{E}[X|Y]$ and can produce high-quality reconstructions with an one-step inference. In contrast, diffusion-based MAP estimation requires iterative optimization, introducing additional inference cost. It would be helpful to discuss why MAP initialization is preferred over MMSE-style estimators and whether such end-to-end IR models could serve as more efficient alternatives for Stage 1.

4. MAP-RPS introduces several non-trivial inference-time operations, including iterative MAP optimization in Stage 1 and RPS in Stage 2. While the method is motivated by improved D-P control, the resulting computational overhead is not clearly analyzed. A more detailed discussion of the inference cost, such as runtime, neural function evaluations (NFEs), and comparison with simpler baselines, would help clarify the practical trade-off between reconstruction quality and efficiency.

5. There may be a minor notation inconsistency in Eq. (47). The integrand appears to involve $\nabla_x \log p_{X|X_{t_1}}(x_{t_1}\mid x)$, while the subsequent derivation in Eq. (48) computes $\nabla_x \log p_{X|X_{t_1}}(x\mid x_{t_1})$. It seems that the latter form is intended in Eq. (47) as well. Clarifying this notation would help improve the consistency of the proof.

References

Chung, H., Sim, B., & Ye, J. C. Come-closer-diffuse-faster: Accelerating conditional diffusion models for inverse problems through stochastic contraction. CVPR 2022.

Liang et al., SwinIR: Image Restoration Using Swin Transformer. ICCV 2021.

Guo et al., MambaIR: A Simple Baseline for Image Restoration with State Space Model. ECCV 2024.

---

> ### Author Rebuttal · Authors · 2026-03-31
>
> We thank the reviewer for recognizing our two-stage method and theoretical contributions. Below we address the raised concerns. All numerical results are included in this rebuttal, with corresponding visualizations at: https://anonymous.4open.science/r/anonymous_map_rps_rebuttal_vis.
>
> ---
>
> **1. Log-concavity assumption and more challenging tasks.**
>
> We appreciate the suggestion. We supplement the D-P traversal of LMAP-RPS on $8\times$ SR, $25$% CS, and $128$ box inpainting. The numerical results are shown below, while curves and samples are shown in Fig. R.2 & R.3 in the anonymous link. LMAP-RPS maintains desired D-P traversal on these tasks, demonstrating effectiveness for more general posteriors.
>
> We note that the log-concave assumption aligns with practical applications. Many real-world image inverse problems focus on restoration, often corresponding to posteriors concentrated around the ground truth, making a log-concave approximation reasonable. Additionally, [Bohr & Nickl, 2024] (our reference in line 470) theoretically shows that using log-concave functions as surrogate posteriors in high-dimensional inverse problems yields high-confidence estimates.
>
> |||$t_0=0$|$t_0=100$|$t_0=200$|$t_0=300$|$t_0=400$|$t_0=500$|
> |-|-|-|-|-|-|-|-|
> |SR $8\times$|RMSE|0.0739|0.0749|0.0758|0.0761|0.0778|0.0804|
> ||FID|176.4|171.7|167.4|150.0|133.7|113.6|
> |CS $25$%|RMSE|0.0577|0.0624|0.0654|0.0687|0.0748|0.0841|
> ||FID|165.5|160.6|149.6|138.4|133.2|127.1|
> |Inp ($128$ box)|RMSE|0.0776|0.0780|0.0799|0.0830|0.0831|0.0835|
> ||FID|136.2|116.2|100.6|100.0|101.4|103.7|
>
> **2. Comparison with CCDF.**
>
> We thank the reviewer for the question. Our method differs fundamentally from CCDF in objective, theory, and implementation.
>
> **First**, CCDF aims to accelerate inverse problem solving by reducing the number of diffusion steps, while MAP-RPS targets the D-P tradeoff.
>
> **Second**, CCDF focuses on analyzing $\ell_2$ reconstruction error in the posterior sampling stage, while Stage 2 of MAP-RPS analyzes perception error ($W_2$). The different diffusion processes considered in CCDF are equivalent under specific reweighting, and our analysis extends to such variants without essential modification.
>
> **Third**, CCDF relies on a trained feed-forward network for initialization, while we do not assume access to an additional network and task-specific paired data for training in the zero-shot setting.
>
> Finally, we provide a comparison under a unified setting: assuming only a single diffusion model and using pseudo-inverse initialization for CCDF. RMSE / FID are shown below, with D-P curves shown in Fig R.4. in the anonymous link. CCDF generally benefits from more steps, whereas MAP-RPS improves perception at the cost of distortion as timesteps increase, further highlighting their distinct objectives.
>
> |Task|Algo|$t_0=0$|$t_0=25$|$t_0=50$|$t_0=75$|$t_0=100$|
> |-|-|-|-|-|-|-|
> |Inp|MAP-RPS|0.0336 / 82.33|0.0355 / 62.18|0.0369 / 55.02|0.0382 / 52.58|0.0392 / 53.47|
> ||CCDF|0.1510 / 145.0|0.1293 / 135.0|0.1161 / 121.1|0.1054 / 109.6|0.0960 / 106.1|
> |||$t_0=0$|$t_0=100$|$t_0=200$|$t_0=300$|$t_0=400$|
> |SR4|MAP-RPS|0.0547 / 133.8|0.0572 / 100.2|0.0596 / 85.74|0.0620 / 77.63|0.0633 / 78.54|
> ||CCDF|0.1101 / 253.5|0.0977 / 269.8|0.0832 / 157.3|0.0769 / 98.54|0.0666 / 85.30|
>
> **3. "Why is MAP initialization preferred over MMSE-style estimators?"**
>
> We thank the reviewer for the question. In the zero-shot inverse problem setting with a single diffusion model, introducing an additional network or training with task-specific paired data is undesirable. An extra network violates the single-model inference setting, while paired data contradicts the zero-shot assumption.
>
> **4. Computational overhead, NFEs, and simpler baselines.**
>
> We thank the reviewer for the question.
>
> For computational overhead, we refer to Table 2 (p.8) and Table 8 (p.25). LMAP-RPS is faster than most baselines.
>
> For NFEs, MAP-RPS requires the sum of Stage 1 optimization iterations and Stage 2 sampling timesteps, as reported in Tables 3–6. We will annotate the NFEs for each method in the final version.
>
> Regarding simpler baselines, we refer to our response to Reviewer J7Js (point 3), where we further compare MAP-RPS with seven pixel-space inverse algorithms. We apologize for any inconvenience caused by the rebuttal length constraint.
>
> **5. Typo in Eq (47).**
>
> We thank the reviewer for pointing this out. The correct expression is
> $$
> \nabla\_{\mathbf{x}}\log p\_X(\mathbf{x})=\frac{1}{p\_X(\mathbf{x})}\int p\_{X\_{t\_1}}(\mathbf{x}\_{t\_1})\nabla\_{\mathbf{x}}p\_{X\mid X\_{t\_1}}(\mathbf{x}\mid\mathbf{x}\_{t\_1})\mathrm{d}\mathbf{x}\_{t\_1}=\int p\_{X\_{t\_1}\mid X}(\mathbf{x}\_{t\_1}\mid \mathbf{x})\nabla\_{\mathbf{x}}\log p\_{X\mid X\_{t\_1}}(\mathbf{x}\mid \mathbf{x}\_{t\_1}).
> $$
> We will correct this in the final version.
>
> ---
> We thank the reviewer for the valuable comments and suggestions. All experiments and discussions will be included in the final version, and we welcome further discussion.

---

> > ### Author Rebuttal · Reviewer_N4k8 · 2026-04-01
> >
> > The authors have provided a comprehensive response that has adequately addressed my concerns. I will further take into account the opinions of the other reviewers and consider raising my score accordingly.

---

> > > ### Author Response · Authors · 2026-04-01
> > >
> > > We sincerely thank the reviewer for the positive feedback. We are highly encouraged to hear that our rebuttal has adequately addressed your concerns. We would greatly appreciate it if you would consider raising your score after taking all factors into account.

---

### Official Review · Reviewer_6254 · 2026-03-11

**Soundness:** 3
**Presentation:** 3
**Significance:** 3
**Originality:** 3
**Overall Recommendation:** 4
**Confidence:** 3

**Summary:**

This paper introduces a stage-wise framework, MAP-RPS, designed to navigate the fundamental distortion-perception (D-P) tradeoff in zero-shot inverse problems using a single pretrained diffusion model. The method operates in two distinct stages: first, it employs a maximum a posteriori (MAP) estimation to approximate the minimum mean squared error (MMSE) solution, establishing a low-distortion starting point. Second, it utilizes a re-noised posterior sampling strategy to progressively improve perceptual quality by adjusting the re-noising timestep. The authors also provide a latent-space extension, LMAP-RPS, which leverages powerful pre-trained latent diffusion backbones for broader applicability and improved computational efficiency. Theoretical analyses support both stages, demonstrating that MAP-RPS effectively approximates the ideal D-P curve across various image restoration tasks like super-resolution, inpainting, and deblurring. Extensive experiments on datasets such as FFHQ and MS-COCO show that MAP-RPS and LMAP-RPS achieve state-of-the-art performance, outperforming existing latent diffusion-based algorithms in both reconstruction fidelity and perceptual quality while maintaining lower complexity. Overall, the framework offers a principled and efficient mechanism for flexible D-P traversal at inference time.

**Compliance With Llm Reviewing Policy:**

Affirmed.

**Final Justification:**

I thank the authors for their response. I keep my original score.

**Key Questions For Authors:**

1. Was a validation set utilized to optimize the hyperparameters $N$, $t_0$, and $t_1$? If not, could you clarify the criteria used for parameter selection? To ensure a rigorous comparison, it is essential that the baseline methods are afforded an equivalent level of hyperparameter optimization.

2. Error accumulation in the re-noised posterior sampling: Theorem 3.5 includes an $\epsilon_{score}$ term representing accumulated posterior score estimation error. In the LMAP-RPS extension, how does the error introduced by the latent space mapping (VAE/Autoencoder) interact with this score error?

**Limitations:**

yes

**Strengths And Weaknesses:**

Strengths:

1. Superior Efficiency and Practicality: MAP-RPS is significantly more computationally efficient than alternative strategies like sample averaging. And it achieves good performance on both distortion and perception metrics.

2. Effective D-P Traversal: The framework allows for flexible and principled traversal of the D-P curve using a single pretrained diffusion model.

Weaknesses:

1. Dependence on Log-Concavity Assumptions: The theoretical validity of using the MAP estimator as a surrogate for the MMSE solution relies on the posterior distribution being strongly log-concave.

2. Sensitivity to Optimization Hyperparameters: The performance of the MAP stage is sensitive to the number of optimization iterations ($N$).

---

> ### Author Rebuttal · Authors · 2026-03-31
>
> We thank the reviewer for recognizing the efficiency, practicality, and effectiveness of our proposed algorithm. Below we provide responses to the questions raised. All numerical results are included in this rebuttal, with corresponding visualizations available at: https://anonymous.4open.science/r/anonymous_map_rps_rebuttal_vis.
>
> ---
>
> **1. "Dependence on Log-Concavity Assumptions."**
>
> We appreciate the reviewer's attention to this assumption. We would like to emphasize that the log-concave assumption is reasonable in practical applications and is supported by rigorous theoretical guarantees. First, many real-world image inverse problems, particularly so-called image restoration tasks, have posteriors that are typically concentrated around the ground truth and are approximately unimodal. Examples include Image Super-resolution, Deblurring, and Denoising. These tasks have been extensively studied in the literature and hold practical significance. Second, [Bohr & Nickl, 2024] (see our reference in line 470) theoretically analyzed the rationale and feasibility of approximating high-dimensional complex inverse problems with a log-concave posterior, providing support for our assumption.
>
> Moreover, while the error bound in Theorem 3.2 depends on the log-concavity assumption, our algorithm generalizes to a broader set of tasks. The table below shows the D-P tradeoff of LMAP-RPS on $8\times$ Super-resolution, $25$% Compressed Sensing, and $128\times128$ box inpainting (also see the D-P curve and qualitative results in Fig. R.2 and R.3 in the anonymous link). These tasks involve more severe degradation, where the posterior may be less log-concave. LMAP-RPS still demonstrates desired D-P traversal performance in these scenarios, validating the generality of our approach.
>
> |||$t_0=0$|$t_0=100$|$t_0=200$|$t_0=300$|$t_0=400$|$t_0=500$|
> |-|-|-|-|-|-|-|-|
> |SR $8\times$|RMSE|0.0739|0.0749|0.0758|0.0761|0.0778|0.0804|
> ||FID|176.4|171.7|167.4|150.0|133.7|113.6|
> |CS $25$%|RMSE|0.0577|0.0624|0.0654|0.0687|0.0748|0.0841|
> ||FID|165.5|160.6|149.6|138.4|133.2|127.1|
> |Inp ($128$ box)|RMSE|0.0776|0.0780|0.0799|0.0830|0.0831|0.0835|
> ||FID|136.2|116.2|100.6|100.0|101.4|103.7|
>
> **2. "Sensitivity to Optimization Hyperparameters $N$."**
>
> We thank the reviewer for pointing this out. Similar to many diffusion-based inverse algorithms, our method also requires hyperparameter tuning. While the number of optimization iterations $N$ has effect on the RMSE of LMAP-RPS, our ablation study in Section E.5 shows that the FID performance of LMAP-RPS, as well as the overall performance of MAP-RPS, is fairly robust with respect to $N$. Moreover, we note that since the MAP stage is lightweight, adjusting $N$ is very fast and convenient. The detailed procedure for tuning the hyperparameters is addressed in our response to point 3 below.
>
> **3. Validation set and hyperparameter tuning for baseline methods.**
>
> We thank the reviewer for this question. All hyperparameter tuning is performed on the first two test samples, which is a common practice in existing literature. For all baseline methods, our hyperparameter search procedure is as follows: we fix the sampling timesteps according to the recommended values in the original papers, and perform task-by-task grid search for other hyperparameters, including step size (a.k.a. learning rate), number of optimization steps, and so on. For LMAP-RPS, we only search over the hyperparameters in the LMAP stage (e.g., $N, t_0$), while the RPS stage directly uses the optimal hyperparameters corresponding to the posterior sampling method. This hyperparameter tuning procedure largely ensures that all baseline methods achieve optimal performance for each task under their recommended settings.
>
> **4. "How does $\epsilon_\text{score}$ interact with VAE?"**
>
> We thank the reviewer for pointing out this theoretical detail. For LMAP-RPS, $\epsilon_\text{score}$ represents the posterior score estimation error in the latent space. It is mapped to the data space through the Lipschitz decoder via a scaling inequality on the Wasserstein distance (i.e., Eq. (71), line 970, page 18). More specifically, for any latent-space coupling between $p_Z$ and $p_{\hat Z}$, we can always construct a corresponding coupling between $\mathcal{D}(Z)$ and $\mathcal{D}(\hat Z)$ such that the scaling of the Wasserstein distance is controlled by the $L_{\mathcal{D}}$-Lipschitz decoder $\mathcal{D}$. Therefore, the $\epsilon_\text{score}$ term in Theorem C.3 differs from that in Theorem 3.5 by an additional scaling factor of $L_\mathcal{D}$.
>
> ---
>
> We would like to once again thank the reviewer for their valuable suggestions. All additional experiments and discussions will be incorporated into the final version. If there are any remaining questions, we welcome further discussion.

---

> > ### Author Rebuttal · Reviewer_6254 · 2026-04-02
> >
> > thanks for your response.

---

> > > ### Author Response · Authors · 2026-04-03
> > >
> > > Thank you for the positive acknowledgment. We are highly encouraged to hear that our rebuttal has adequately addressed all your concerns.

---

### Official Review · Reviewer_J7Js · 2026-03-13

**Soundness:** 3
**Presentation:** 4
**Significance:** 2
**Originality:** 2
**Overall Recommendation:** 4
**Confidence:** 3

**Summary:**

This paper mainly studies the distortion-perception tradeoff in inverse problem solving based on diffusion models. Through the re-noising idea, the MAP-RPS framework and its variant LMAP-RPS are designed to adjust the distortion and perception tradeoff in the inverse problem process.

**Compliance With Llm Reviewing Policy:**

Affirmed.

**Final Justification:**

The rebuttal addressed all my concerns and show their contribution is not simple as I thought initilally. Therefore I raise my score to "4, weak accpet".

**Key Questions For Authors:**

1. The algorithms currently adopted by the authors in MAP and RPS are mainly DPS or Latent-DPS. Can the authors provide the application of their framework on other inverse problem solving algorithms, and give performance comparisons before and after the application to demonstrate the applicability of the proposed framework?
2. See Weaknesses 2.
3. In Table 1, on the Inpainting and HDR tasks, LMAP-RPS(600) shows degradation in both LPIPS and FID compared to LMAP-RPS(0). Therefore, the method that only performs MAP operation ultimately performs better in perception than the re-noised result. Does this mean that the proposed framework cannot well balance distortion and perception?
4. See Weaknesses 3.

**Limitations:**

The main limitation of this paper lies in the misalignment between its methodological starting point and the final experimental results (refer to Question 3), as well as the algorithm design being merely a combination of existing algorithms, which is too similar to existing methods.

**Strengths And Weaknesses:**

**Strengths**

1. The theory provided in this paper is well justified.
2. The writing logic of this paper is clear, and the algorithm design is explicit.

**Weaknesses**

1. The algorithmic improvement proposed in this paper is relatively simple. The MAP algorithm and RPS algorithm adopted are both from prior work, with only the re-noising step added.
2. The experimental results presented in this paper may be unfair, because different algorithms use different numbers of optimization iterations N, which leads to differences in both time and performance. For example, the method proposed in this paper uses fewer time steps, and therefore shows improvement in time performance. The authors need to provide performance comparisons of the algorithm under the same number of optimization iterations.
3. Although this paper compares many Latent Diffusion Model methods, the comparison with Pixel Diffusion Model-based methods is relatively limited. The authors should compare with more Pixel Diffusion Model methods on top of the existing basis.

---

> ### Author Rebuttal · Authors · 2026-03-31
>
> We thank the reviewers for recognizing our theoretical contributions. Our responses to all comments are below. **All metrics are reported as PSNR / SSIM / LPIPS / FID.** Visualizations are available at: https://anonymous.4open.science/r/anonymous_map_rps_rebuttal_vis.
>
> ---
> **1. "The algorithmic improvement is relatively simple."**
>
> We respectfully disagree. Our novelty is threefold: (1) we propose a principled two-stage framework for D-P traversal in zero-shot inverse problems; (2) we derive a novel, lightweight MAP estimator using a pretrained diffusion prior; (3) we provide theoretical guarantees and extensive experimental validation.
>
> We'd like to highlight that our primary objective is to enable D-P traversal, rather than proposing a new inverse algorithm. The principled two-stage design, together with our theoretical analysis, directly supports this goal, and our MAP estimator is also novel and effective.
>
> **2. Comparison under the same NFEs.**
>
> We thank the reviewer for the suggestion. Here we present results on MS-COCO Inpainting and Deblurring tasks where the NFEs of all baseline methods are set to 200. LMAP-RPS achieves highly competitive performance under the same NFEs.
>
> We'd also like to clarify that for all our experiments, baseline hyperparameters are set as follows: we use the timesteps (NFEs) recommended in the original papers, and tune other hyperparameters (e.g., step size, optimization iterations) via grid search on the first two test samples. This largely ensures that each baseline reflects its optimal performance.
> ||NFEs|Inp|Deblur|
> |-|-|-|-|
> |Latent-DPS|200|21.9 / 0.553 / 0.451 / 203|17.4 / 0.386 / 0.531 / 316|
> |Resample|200|24.2 / 0.646 / 0.372 / 163|24.6 / 0.627 / 0.398 / **98.5**|
> |PSLD|200|25.2 / 0.660 / 0.412 / 131|23.2 / 0.612 / 0.457 / 131|
> |STSL|200|27.7 / 0.777 / 0.306 / 91.5|22.0 / 0.533 / 0.479 / 167|
> |LDIR|200|27.9 / 0.788 / 0.312 / 78.4|26.0 / 0.723 / **0.377** / 108|
> |Latent-DCDP|200|26.9 / 0.765 / 0.321 / 80.5|25.5 / 0.693 / 0.384 / 107|
> |Latent-DMAP|200|27.2 / 0.764 / 0.328 / 101|24.7 / 0.672 / 0.425 / 137|
> |Latent-DAPS|200|25.9 / 0.694 / 0.396 / 104|23.7 / 0.585 / 0.456 / 139|
> |Latent-SITCOM|200|27.0 / 0.753 / 0.374 / 126|23.6 / 0.627 / 0.466 / 184|
> |LMAP-RPS (0)|200|**28.1 / 0.799 / 0.277 / 61.4**|**25.0 / 0.700** / 0.389 / 108|
>
> **3. Comparison with pixel space method.**
>
> We thank the reviewer for the suggestion. The table below compares MAP-RPS with seven pixel-space methods on FFHQ ($\sigma_\mathbf{y}=0.05$). MAP-RPS achieves highly competitive performance.
> ||Inp|Deblur|SR$4\times$|
> |-|-|-|-|
> |DDNM|33.6 / 0.917 / 0.151 / 58.1|29.8 / 0.843 / 0.219 / 80.5|28.8 / 0.824 / 0.243 / 88.5|
> |DPS|33.1 / 0.907 / 0.138 / 39.8|27.1 / 0.763 / 0.234 / 69.8|24.5 / 0.687 / 0.307 / 90.6|
> |RED-diff|28.5 / 0.765 / 0.316 / 103|27.7 / 0.776 / 0.326 / 98.3|26.9 / 0.668 / 0.443 / 124|
> |DMPS|27.9 / 0.834 / 0.229 / 89.9|28.1 / 0.799 / 0.265 / 97.1|26.0 / 0.745 / 0.308 / 112|
> |DiffPIR|30.7 / 0.859 / 0.236 / 97.5|27.6 / 0.777 / 0.304 / 114 |25.5 / 0.719 / 0.347 / 120|
> |DAPS|30.5 / 0.779 / 0.239 / 61.5|28.7 / 0.783 / 0.277 / 77.4 | 27.7 / 0.765 / 0.294 / 86.7|
> |SITCOM|30.0 / 0.768 / 0.254 / 78.8 |27.1 / 0.659 / 0.358 / 109 |24.7 / 0.543 / 0.505 / 156|
> |MAP-RPS (0)|34.8 / 0.903 / 0.163 / 47.7|**31.6** / 0.857 / 0.239 / 68.6 | **29.1** / 0.822 / 0.289 / 93.9|
> |MAP-RPS (300)|**35.8 / 0.943 / 0.110 / 36.5**|30.7 / **0.865 / 0.178 / 56.9**|28.8 / **0.826 / 0.232 / 75.4**|
>
> **4. More posterior sampling methods.**
>
> We thank the reviewer for the suggestion. We further compare MAP-RPS on denoising ($\sigma_\mathbf{y}=0.3$) using DPS, $\Pi$GDM, and DMPS. We kindly refer to Point 4 of our response to Reviewer gEN5 for metrics, and to the D-P curves in Fig. R.1 in the anonymous link. We apologize for any inconvenience caused by the rebuttal length constraint.
>
> **5. LMAP-RPS with $t_0=600$ shows degradation in both distortion and perception metrics on inpainting and HDR.**
>
> We thank the reviewer for pointing this out. We emphasize that this behavior is task-dependent, not a limitation of the algorithm. In Table 1, the observation noise is $\sigma_\mathbf{y}=0.05$, which is generally smaller than the noise used in our D-P traversal experiments. For random inpainting and HDR, this leads to a more concentrated posterior, aligning distortion and perception metrics. Consequently, at $t_0=600$, PSNR and FID both degrade because the error is dominated by score estimation in Latent-DPS (see Theorem 3.5 and Theorem C.3). A similar effect is observed for MAP-RPS as presented in our response to point 3 above. A difference is that MAP-RPS with $t_0=300$ achieves both better distortion and perception metrics in inpainting. This is because we apply DDNM in the RPS stage, which introduces smaller estimation errors at small $t$s.
>
> ---
> We thank the reviewers for their valuable feedback. All additional experiments and discussions will be included in the final version. We welcome further questions and discussion if needed.

---

> > ### Author Rebuttal · Reviewer_J7Js · 2026-04-02
> >
> > The authors have addressed all my concerns and therefore I decide to raise my score and recommend the paper as "4, weak accept"

---

> > > ### Author Response · Authors · 2026-04-02
> > >
> > > We sincerely thank the reviewer for the positive feedback and for raising the score. We are greatly encouraged to hear that our responses have addressed all your concerns.

---

### Official Review · Reviewer_gEN5 · 2026-03-13

**Soundness:** 2
**Presentation:** 4
**Significance:** 1
**Originality:** 2
**Overall Recommendation:** 2
**Confidence:** 4

**Summary:**

The paper introduces MAP-RPS, a two-stage framework that first approximates the MMSE with a MAP estimate, allowing for a low-distortion initialization, which is renoised at intermediate diffusion noise levels to follow a posterior sampling second stage, improving perceptual quality. This allows for a principal distortion-perceptual quality traversal than ad hoc related methods. The latent diffusion variant LMAP-RPS was also introduced, and extensive experimental results show strong performance of MAP-RPS and LMAP-RPS as inverse problem solvers.

**Compliance With Llm Reviewing Policy:**

Affirmed.

**Final Justification:**

While I appreciate the authors' responses and experiments conducted during the rebuttal, I find the lack of representative experiments, the arguments, method design, and conclusions drawn to be very conflated, as raised in my original review and during the rebuttal. Therefore, I'm inclined towards maintaining my original rating.

**Key Questions For Authors:**

Please see the weaknesses above.

**Limitations:**

Yes

**Strengths And Weaknesses:**

**Strengths:**

>The motivation of the work is novel, and the paper is very well written and organized. The related works, the methodology, and experimental design were well explained, making it easy to follow the paper. Under the strong log-concavity assumption, the methodology appears sound, though it does not hold in general.

**Weaknesses:**

>The strong log-concavity assumption is severely restrictive. As the authors already mentioned, it leads to a unimodal posterior. This clearly fails for several inverse problems with moderate-high degradation operators, and only holds for cases where the measurement is highly correlated with the data sample (e.g., for minor perturbations or degradations that preserve much information).

>Only for an unimodal posterior, the first stage in LMAP-RPS makes sense, as the MMSE and the MAP estimate converge. Also, the proposed MAP estimation feels ad hoc, and the question of why the authors did not use existing MAP methods remains.

>MMSE estimator, i.e., $E[x\_0|y]$ can be approximated without explicit MAP solving since it turns out that $E[x\_0|y]$ = $E[x\_t|y]$ = $x\_t + \\sigma^2\_t \\nabla_{x\_t} \\log p(x\_t|y)$, assuming variance-exploding formulation. One can directly approximate $\nabla_{x_t} \log p(x_t|y)$ with the DPS-style approximation. So the necessity of the proposed approximation of MMSE with the MAP estimate is in question.

>Also, note that the DMAP paper shows that DPS encourages MAP solutions more than behaving like a posterior sampler. So the potential consequence of using Latent-DPS in the second stage as a posterior sampler, while it is known to promote MAP solutions, is unclear and needs careful analysis.

>The experiments also considered easy inverse problems to enforce a unimodal posterior, which do not reflect challenging real-world settings.

>The contribution and the scope of the method are limited to strongly log-concave distributions, which is a very simplistic case far from any real-world use cases. So I feel that the proposed principled traversal of the D-P tradeoff cannot be generalized any further beyond the simple case considered in the paper.

---

> ### Author Rebuttal · Authors · 2026-03-31
>
> We thank the reviewer for recognizing the novelty of our method. Below, we clarify the misunderstandings and questions. All numerical results are included in this rebuttal, with corresponding visualizations available at: https://anonymous.4open.science/r/anonymous_map_rps_rebuttal_vis.
>
> ---
> **1. "The strong log-concavity is severely restrictive."**
>
> We respectfully disagree.
>
> First, the log-concave posterior assumption aligns with practical scenarios and is well-supported theoretically. In numerous image inverse problems, especially **restoration** tasks, the posterior is typically concentrated around the ground truth and unimodal. Examples include super-resolution, deblurring, and denoising, which are widely studied in the literature and of significant practical importance. Moreover, our reference [Bohr & Nickl, 2024] theoretically shows that, in high-dimensional inverse problems, log-concave approximations closely match the true posterior with high probability, supporting our assumption.
>
> Second, while Theorem 3.2 relies on log-concavity for the upper bound, our method generalizes beyond strictly log-concave settings. We present additional quantitative results of LMAP-RPS below: on $8\times$ super-resolution, $25$% compressed sensing, and $128\times 128$ box inpainting, we observe consistently favorable D-P tradeoff (also see Fig. R.2 and R.3 in the anonymous link), demonstrating robustness and broad applicability.
>
> |||$t_0=0$|$t_0=100$|$t_0=200$|$t_0=300$|$t_0=400$|$t_0=500$|
> |-|-|-|-|-|-|-|-|
> |SR $8\times$|RMSE|0.0739|0.0749|0.0758|0.0761|0.0778|0.0804|
> ||FID|176.4|171.7|167.4|150.0|133.7|113.6|
> |CS $25$%|RMSE|0.0577|0.0624|0.0654|0.0687|0.0748|0.0841|
> ||FID|165.5|160.6|149.6|138.4|133.2|127.1|
> |Inp ($128$ box)|RMSE|0.0776|0.0780|0.0799|0.0830|0.0831|0.0835|
> ||FID|136.2|116.2|100.6|100.0|101.4|103.7|
>
> **2. "Why not use existing MAP methods?"**
>
> Our proposed MAP solver is both more efficient and empirically accurate, which constitutes one of our core contributions. Existing diffusion-based zero-shot MAP estimators (e.g., RED-diff) typically require long annealing schedules. Moreover, trained MAP estimators are not applicable in the zero-shot inverse problems, as they rely on paired training data and require task-specific training.
>
> **3. Directly calculating MMSE estimation $\mathbb{E}[\mathbf{x}_0\mid\mathbf{y}]$.**
>
> We believe the reviewer is referring to Tweedie’s formula:
> $$
> \mathbb{E}[\mathbf{x}\_0| \mathbf{y}, \mathbf{x}\_t]=\mathbf{x}\_t+\sigma\_t^2\nabla\_{\mathbf{x}\_t}\log p(\mathbf{x}\_t|\mathbf{y}).
> $$
> Note that this corresponds to the posterior expectation conditioned on $\mathbf{x}_t$, and therefore does not avoid the computational cost when integrating over $p(\mathbf{x}_t\mid\mathbf{y})$. We also refer to Appendix E.4, where we discuss the applicability of such Monte Carlo-based estimators and provide a detailed analysis of their practical computational complexity compared to our method.
>
> **4. "DPS is more aligned with the MAP solution."**
>
> DMAP shows that DPS is closer to a *local* MAP solution, i.e.,
>
> $$
> X\_{t-1}\sim\delta(X-\arg\max p(X\_{t-1}\mid X_t, \mathbf{y})),
> $$
>
> which is not equivalent to the global MAP $\arg\max p(X_0\mid \mathbf{y})$. DPS is fundamentally a posterior sampling method with theoretical guarantees, where the approximation error is bounded via Jensen’s gap (Theorem 1 in the DPS paper). Empirically, DPS also achieves significantly better FID but only marginal or slightly lower PSNR (Tables 1, 2, 6, 7 in the DPS paper).
>
> Moreover, Stage 2 is not restricted to DPS, and we can adopt any reasonable posterior sampler. Here we further evaluate MAP-RPS on denoising ($\sigma_\mathbf{y}=0.3$) using DPS, $\Pi$GDM, and DMPS. The resulting D-P curves are consistently reasonable across methods, with accuracy depending on the sampler's quality.
>
> |||$t_0=0$|$t_0=25$|$t_0=50$|$t_0=75$|$t_0=100$|
> |-|-|-|-|-|-|-|
> |MAP-RPS (DPS)|RMSE|0.0350|0.0352|0.0356|0.0362|0.0367|
> ||FID|96.64|67.99|57.87|55.10|53.49|
> |MAP-RPS ($\Pi$GDM)|RMSE|0.0350|0.0352|0.0365|0.0374|0.0384|
> ||FID|96.64|84.14|74.06|70.93|69.65|
> |MAP-RPS (DMPS)|RMSE|0.0350|0.0358|0.0385|0.0413|0.0446|
> ||FID|96.64|86.22|75.10|68.75|66.27|
>
> **5. "The experiments do not reflect real-world settings."**
>
> We respectfully disagree. Most of our settings are standard and practically relevant, such as $4\times$ SR and denoising with $\sigma_\mathbf{y}=0.3$ (corresponding to $\sigma_\mathbf{y}=75$ for data range [0, 255]). These tasks are widely studied in prior work. For example, MambaIR evaluates super-resolution at scales $2, 3, 4$ and denoising with $\sigma=15, 25, 50$; our settings are in fact more challenging. Moreover, our method generalizes to more difficult scenarios. We refer to our response to Point 1.
>
> ---
> We hope these clarifications and new results better demonstrate the validity, practical relevance, and novelty of our work. All additions will be included in the final version, and we welcome further discussion if needed.

---

> > ### Author Rebuttal · Reviewer_gEN5 · 2026-04-03
> >
> > I appreciate the authors' rebuttal and their address of some of my concerns. However, I do not find the method's main arguments for its applicability beyond simplified unimodal posteriors convincing, as this limitation is quite straightforward from theory.
> >
> > > Q1. My question is not regarding real-world applicability (it may be applicable if the tasks have a unimodal posterior, obviously). The arguments you mentioned again are based on the assumption that the posterior is mostly unimodal. Your method's first stage itself is entirely based on this assumption. The severity of tasks you presented (e.g., image restoration) depends on how multimodal the posterior can be, and it depends on how complex the data distribution is in the first place. Which dataset are the presented experiments on? Considering large-hole inpainting (e.g., huge blob masks) on ImageNet would simply prove this point.
> >
> > > Q2. I do not find this a fair argument, especially when the latest zero-shot diffusion MAP solvers are not even considered in experiments.
> >
> > > Q3. Indeed, there was a typo in my earlier question. In summary, my concern was that since you want to aim for the posterior mean $\mathbb{E}[x_0|y]$, why not derive an approximation that directly aims at estimating this? The MAP estimation method you proposed will coincide only in the case of a unimodal posterior, which is the biggest assumption in my concern.
> >
> > > Q4: Isn't the global and local MAP the same in the unimodal posterior setting? It is hard to understand the effects of stages 1 and 2 since the algorithms like piGDM or DPS, etc., inherently rely on approximations that are ideal to facilitate MAP estimation.
> >
> > > Q5: To clarify, the real-world setting where posteriors are multimodal and challenging. As also mentioned, for cases where the posterior is sufficiently unimodal, e.g., a dominant peak, that method may be applicable. Again, a proper dataset like ImageNet with severe degradation operators would prove this point.

---

> > > ### Author Response · Authors · 2026-04-07
> > >
> > > Thank you for the follow-up. Anonymous link: https://anonymous.4open.science/r/anonymous_map_rps_rebuttal_vis.
> > > ## Q1 & Q5
> > >
> > > The table below reports the D–P tradeoff of LMAP-RPS on **ImageNet** $128\times 128$ **box inpainting** ($\sigma_\mathbf{y}=0.1$), where our method continues to achieve a favorable D–P curve. The corresponding curves and visualizations are provided in Fig. R.Reply.1 and Fig. R.Reply.2 of the anonymous link.
> > >
> > > Our experiments included in the previous rebuttal, i.e., SR $8\times$, CS $4\times$, and box inpainting, are conducted on FFHQ.
> > >
> > > We believe an important point of agreement has been established: our proposed method is clearly applicable in scenarios where the log-concavity assumption approximately holds. We would like to again emphasize that numerous real-world applications fall into this regime. This strongly supports the wide applicability of our approach.
> > >
> > > Furthermore, the additional experiments provided during the rebuttal and this reply demonstrate that our method can adapt to more challenging settings, such as the large-hole inpainting on ImageNet mentioned by the reviewer.
> > > |LMAP-RPS|$t_0=0$|$t_0=200$|$t_0=400$|$t_0=600$|$t_0=800$|
> > > |-|-|-|-|-|-|
> > > |RMSE|0.1124|0.1128|0.1153|0.1296|0.1355|
> > > |FID|279.71|267.19|227.15|143.93|125.99|
> > > ## Q2
> > >
> > > Our claim regarding the superior empirical accuracy and efficiency of the proposed MAP solver is directly supported by the experimental results already presented in the paper (see Table 2 and Table 3).
> > >
> > > To further substantiate this claim, we include an additional comparison on FFHQ (denoising with $\sigma_\mathbf{y}=0.3$). We incorporate RED-diff as an alternative MAP solver. To clarify, RED-diff is an MAP estimator because it effectively employs a variational distribution of the form $\delta(\mathbf{x}-\mathbf{x}_0)$, which reduces to MAP estimation. The results are shown below. The corresponding D-P curves are shown in Fig. R.Reply.3 in the anonymous link, where MAP-RPS achieves the most favorable trade-off.
> > >
> > > In addition, we consider DMAP as an initialization. We again emphasize that the solution obtained by **DMAP is not a MAP estimate**. See our response to Q4 for further discussion.
> > >
> > > For computational overhead, MAP-RPS requires clearly fewer NFEs.
> > > |Algo|Base NFEs||$t_0=0$|$t_0=25$|$t_0=50$|$t_0=75$|$t_0=100$|
> > > |-|-|-|-|-|-|-|-|
> > > |MAP-RPS|60|RMSE|0.0350|0.0352|0.0356|0.0362|0.0367|
> > > |||FID|96.64|67.99|57.87|55.10|53.49|
> > > |RED-diff-RPS|1000|RMSE|0.0413|0.0368|0.0367|0.0369|0.0372|
> > > |||FID|82.09|59.83|54.37|56.35|54.79|
> > > |DMAP-RPS|400|RMSE|0.0364|0.0366|0.0363|0.0372|0.0383|
> > > |||FID|140.83|108.83|64.08|61.28|60.59|
> > > |RMP-RPS|1000|RMSE|0.0363|0.0353|0.0355|0.0358|0.0363|
> > > |||FID|109.36|66.15|58.21|57.49|55.25|
> > > ## Q3
> > >
> > > To the best of our knowledge, Monte Carlo estimation is the most effective general approach for approximating the MMSE estimator. However, it is computationally expensive in diffusion-based inverse problems due to the need for repeated sampling. We again kindly refer the reviewer to Appendix A (lines 619–623) and Appendix E.4 for further discussion. A primary motivation of our Stage 1 is precisely to avoid this sampling overhead.
> > >
> > > The reviewer may also be aware of RMP [R.1], which attempts to ''propagate'' the MMSE solution to obtain the MMSE of $\mathbf{x}_0$. A closer examination of the method reveals that its practical approximation is valid only under Gaussian posteriors. We also evaluate RMP as Stage 1, shown in the table above.
> > >
> > > We emphasize that our Stage 1 is **a valid approximation of MMSE under reasonable assumptions**. Notably, beyond Monte Carlo, alternative MMSE approximation strategy (e.g., RMP) requires some form of assumption on the posterior distribution, and our approach falls into this category. Using MAP as a proxy for MMSE yields competitive PSNR, as already shown in our experiments (see Table 2 and all additional experiments).
> > > ## Q4
> > >
> > > Here, the term *''local MAP''* refers to
> > > $$
> > > \arg\max_{X_{t-1}} p(X_{t-1}\mid X_t,\mathbf{y}),
> > > $$
> > > i.e., a conditional MAP estimate given the previous step. This is fundamentally different from ''local minimum'' or ''global minimum'' in optimization. Importantly, such a formulation does not guarantee convergence to the global MAP $\arg\max p(\mathbf{x}_0\mid \mathbf{y})$. We kindly refer to [R.2] for a more detailed discussion on this point.
> > >
> > > DPS-type methods, which aim to approximate the posterior score through various techniques, fall into the category of **Diffusion-based Posterior Sampling** approaches. These are valid **posterior sampling methods**. We again kindly note that such methods empirically tend to achieve better FID (i.e., perception quality), while offering comparatively fewer improvements in PSNR (i.e., distortion). **This facilitates the objective of Stage 2, i.e., traversing from low distortion to better perception.**
> > >
> > > [R.1] Xue et. al. Score-Based Reverse Mean Propagation for Solving Inverse Problems. TSP.
> > >
> > > [R.2] Zhang et. al. Local MAP Sampling for Diffusion Models. Arxiv.

---

### Decision · Program_Chairs · 2026-04-30

**Decision:**

Accept (regular)

**Comment:**

This paper addresses an important problem and makes a clear practical contribution: a simple two-stage framework for inference-time traversal of the distortion–perception tradeoff in zero-shot inverse problems using a single diffusion model. The paper is well written, and the rebuttal materially strengthened the empirical case through matched-NFE comparisons, broader pixel-space baselines, alternative samplers and initializers, and harder evaluation settings.

The main remaining weakness is scope. The Stage-1 theory justifies MAP as an MMSE surrogate only under strongly log-concave or approximately unimodal posteriors, and the paper does not fully establish that the same principled interpretation extends to broader multimodal settings. In addition, the Stage-2 interpretation remains somewhat under-justified when instantiated mainly with DPS-like samplers, since the distinction between posterior-sampling behavior and MAP-seeking behavior is not fully clarified.

Overall, I do not view these limitations as fatal to the paper’s core contribution. The work provides a useful and reasonably supported framework in an important practical regime, and the rebuttal addressed several key evaluation concerns. I therefore recommend acceptance.

For the camera-ready version, the authors should narrow the scope of their claims so that the theorem-supported regime is clearly separated from broader empirical observations, and sharpen the conceptual explanation of Stage 2, especially when using DPS-like samplers that may exhibit MAP-seeking behavior. It would also help to acknowledge explicitly that the evidence is strongest in approximately unimodal restoration regimes, with more limited support in strongly multimodal settings.